# Confidence Calibration of Classifiers with Many Classes

**Adrien Le Coz**[1,2,3]    **Stéphane Herbin**[2,3]    **Faouzi Adjed**[1]
[1]IRT SystemX    [2]ONERA - DTIS    [3] Paris-Saclay University
adrien.2mrvb@passinbox.com    stephane.herbin@onera.fr    faouzi.adjed@irt-systemx.fr

## Abstract

For classification models based on neural networks, the maximum predicted class probability is often used as a confidence score. This score rarely predicts well the probability of making a correct prediction and requires a post-processing calibration step. However, many confidence calibration methods fail for problems with many classes. To address this issue, we transform the problem of calibrating a multiclass classifier into calibrating a single surrogate binary classifier. This approach allows for more efficient use of standard calibration methods. We evaluate our approach on numerous neural networks used for image or text classification and show that it significantly enhances existing calibration methods. Our code can be accessed at the following link: `https://github.com/allglc/tva-calibration`.

## 1  Introduction

The considerable performance increase of modern deep neural networks (DNNs) and their potential deployment in real-world applications has made reliably estimating the probability of wrong decisions a key concern. When such components are expected to be embedded in safety-critical systems (e.g., medical or transportation), estimating this probability is crucial to mitigate catastrophic behavior. One way to address this question is to treat it as an uncertainty quantification problem [2, 12], where the uncertainty value computed for each prediction is considered as a confidence. This confidence can be used to reject uncertain decisions proposed by the DNN [13], for out-of-distribution detection [22], or to control active learning [34] or reinforcement learning based systems [76]. When confidence values reliably reflect the true probability of correct decisions, i.e., their accuracy, a predictive system is said to be *calibrated*. In this case, confidence values can be used as a reliable control for decision-making.

We are interested in producing an uncertainty indicator for decision problems where the input is high dimensional and the decision space large, typically classifiers with tens to thousands of classes. For this kind of problem, DNNs are common predictors, and their outputs can be used to provide an uncertainty value at no cost, i.e., without necessitating heavy estimation such as Bayesian sampling [15] or ensemble methods [33]. Indeed, most neural architectures for classification instantiate their decision as a softmax layer, where the maximum value can be interpreted as the maximum of the posterior probability and, therefore, as a confidence. Unfortunately, uncertainty values computed in this way are often miscalibrated. DNNs have been shown to be over-confident [17], meaning their confidence is higher than their accuracy: predictions with 90% confidence might be correct only 80% of the time. A later study [44] suggests that model architecture impacts calibration more than model size, pre-training, and accuracy. For ImageNet classifiers, the accuracy and the number of model parameters are not correlated to calibration, but model families are [11].

These studies show that it is difficult to anticipate the calibration level of confidence values computed directly from DNNs and exhibit the benefits of a complementary post-processing calibration. This calibration process can be seen as a learning step that exploits data from a calibration set, distinct from the training set, and is used to learn a function that maps classifier outputs into better-calibrated values. This process is typically lightweight and decoupled from the issue of improving model

38th Conference on Neural Information Processing Systems (NeurIPS 2024).

performance. A standard baseline for post-processing calibration is Temperature Scaling [17], where the penultimate logit layer is scaled by a coefficient optimized on the calibration set.

Many post-processing calibration methods have been developed for binary classification models [50, 69, 70]. Applying these methods to multiclass classifiers requires some adaptation. One standard approach reformulates the multiclass setting into many One-versus-All binary problems (one per class) [70]. One limitation of this approach is that it does not scale well. When the number of classes is large, the calibration data is divided into highly unbalanced subsets that do not contain enough positive examples to solve the One-versus-All binary problems. Other methods based on Platt scaling [50] involve learning a set of parameters whose size grows with the number of classes. For problems with many classes, they tend to overfit, as we demonstrate in this work.

The main idea of our work is to reformulate the multiclass confidence estimation into a *single* binary problem. This problem can be phrased as the unique question: "Is the prediction correct?". In this formulation, the confidence score is defined as the maximum class probability of the binary problem that outputs $1$ if the predicted class is correct and $0$ otherwise. The intent is that the confidence score accurately describes whether the prediction is correct, regardless of the class. We show that this novel approach, which we call *Top-versus-All* (TvA), significantly improves the performance of standard calibration methods: Temperature and Vector Scaling [17], Dirichlet Calibration [29], Histogram Binning [69], Isotonic Regression [70], Beta Calibration [28], and Bayesian Binning into Quantiles [46]. We also introduce a simple regularization for Vector Scaling or Dirichlet Calibration that mitigates overfitting when the number of classes is high relative to the calibration data size. We conduct experiments on multiple image and text classification datasets and many pre-trained models.

Our main contributions are the following:

- We discuss four issues of the standard approach to confidence calibration.
- To solve these issues, we develop the Top-versus-All approach to confidence calibration of multiclass classifiers, transforming the problem into a single binary classifier's calibration. This straightforward reformulation enables more efficient use of existing calibration methods, achieved with minimal modifications to the methods' original algorithms.
- Applied to scaling methods for calibration (such as Temperature Scaling), TvA allows the use of the binary cross-entropy loss, which is more efficient in decreasing the confidence of wrong predictions and leads to stronger gradients in the case of Temperature Scaling.
  Applied to binary methods for calibration (such as Histogram Binning), TvA significantly improves their performance and makes them accuracy-preserving.
- We demonstrate our approach's scalability and generality with extensive experiments on image classification with state-of-the-art models for complex datasets and on text classification with Pre-trained Language Models (PLMs) and Large Language Models (LLMs).

## 2   Related work

**Calibration**   There are various notions of multiclass calibration. One can consider confidence [17], class-wise [28], top-$r$ [19], top-label [18], decision [75], projection smooth [16], or strong [60, 65] calibration. For recent surveys, we refer to [10] and [63]. In this work, we focus on confidence calibration and not on the calibration of the full probability vector. Indeed, confidence calibration is useful for many applications that only require a single confidence value: selective classification [13], out-of-distribution detection [22], or active learning [34]. For these applications, stronger notions of calibration are both difficult and useless. Also, class-wise calibration metrics do not appropriately scale to large numbers of classes, a setting we consider in this work, as explained in Appendix E.

**Metrics**   Several metrics have been proposed to quantify calibration error. The most common is the Expected Calibration Error (ECE) [46] (see Equation 2). ECE has flaws: the estimation quality is influenced by the binning scheme, and it is not a proper scoring rule [14, 60, 48]. Despite its flaws, it remains the standard comparison metric for confidence calibration. Variants of ECE have also been developed: classwise-ECE [29], ECE with equal mass bins [48, 44], or top-label-ECE, which adds a conditioning on the predicted class [18]. The Brier score [4] is also used to measure calibration. The proximity-informed expected calibration error (PIECE) evaluates the miscalibration due to proximity bias [68]. We mainly use the standard ECE in this work, and the Appendix contains more metrics.

**Training calibrated networks**   Several solutions have been proposed in the literature to improve calibration by training neural networks in specific ways, generally by making use of a new loss term [31, 58, 26, 6, 5]. While these methods directly optimize calibration during the training phase of the networks, they require a high development time, often compromise accuracy, and are not adapted to pre-trained foundation models. That is why we prefer to focus on calibrating already-trained models.

**Post-processing (or post-hoc) calibration methods**   Another approach is to calibrate already-trained models. This lowers the development time by decoupling accuracy optimization and calibration. In this paper, we divide post-hoc calibration methods into two categories: scaling and binary. Scaling methods are derived from Platt scaling [50] and optimize some parameters to scale the logits. Temperature Scaling [17] is a popular simple post-processing calibration method. The logits vector is scaled by a coefficient, which modifies the probability vector. Vector Scaling [17] is more expressive and has good performance in many cases [17, 48, 29]. Matrix Scaling can also be considered for more expressiveness but is difficult to apply without overfitting [17]. Dirichlet Calibration [29] proposes a regularization strategy for Matrix Scaling. [72] developed Ensemble Temperature Scaling. Scaling can be combined with binning [30]. Besides logits or probabilities, features can also be used [35]. Another family of methods tackles binary classification. We designate them as binary methods. Histogram Binning [69] divides the prediction into $B$ bins according to the predicted probability. For each bin, a calibrated probability is computed from the calibration data. The probability becomes discrete: it can only take $B$ values. With some modifications, it outperforms scaling methods [18, 49]. Isotonic Regression [70] learns a piecewise constant function to remap probabilities. Bayesian Binning into Quantiles [46] brings Bayesian model averaging to Histogram Binning. Beta Calibration [28] uses a beta distribution to obtain a calibration mapping.
Our work reformulates the multiclass calibration problem and allows more efficient use of all these calibration methods, with little to no change in their algorithms.

**Multiclass to Binary**   Using binary calibration methods for a multiclass classifier requires adapting the multiclass setting. This is usually done with a One-versus-All approach [70, 17]. The multiclass setting is decomposed into $L$ One-versus-All independent problems: one binary problem for each class. [18] introduce the notion of top-label calibration, i.e., confidence calibration with additional conditioning on the predicted class (top-label). They describe a general multiclass-to-binary framework to develop top-label calibrators. [6] derive $L(L-1)/2$ pairwise binary problems. The approach requires training the classifier from scratch, and its performance decreases with the number of classes. Our work tackles this multiclass-to-binary research problem. Contrary to [6], the One-versus-All approach [70] and top-label calibrators [18], our approach works well for problems with many classes. The methods I-Max [49] and IRM [72] use a shared class-wise strategy to compute a single calibrator. The calibrator is applied to all class probabilities separately, so the class probabilities ranking and prediction might change. In contrast, TvA applied to binary methods rescales the confidence after the class prediction is made. I-Max and IRM consider the full class probabilities vectors, while TvA only considers confidence values. Also, they build on top of Histogram Binning and Isotonic Regression, respectively, while we apply our approach to many calibration methods. A concurrent work building on an intuition similar to ours derives a calibration method based on a Correctness-Aware Loss [38]. Appendix C discusses the differences between our approach and others.

# 3   Problem setting

## 3.1   Background

**Confidence calibration of a classifier**   We consider the classification problem where an input $x$ is associated with a class label $y \in \mathcal{Y} = \{1, 2, ..., L\}$. The neural network classifier $f$ provides a class prediction from a final softmax layer $\sigma$ that transforms intermediate logits $z$ into probabilities. The classifier prediction is the most probable class $\hat{y} = \arg\max_{k \in \mathcal{Y}} f_k(x)$ with $f_k(x)$ referring to the probability of class $k$, and the confidence score defined as $s = \max_{k \in \mathcal{Y}} f_k(x)$. Note that we use the term *confidence* to denote the maximum class probability. With $y$ the real label, we consider the confidence calibration definition from [17] that says that the classifier $f$ is calibrated if:

$$P(\hat{y} = y | s = p) = p, \quad \forall p \in [0, 1] \tag{1}$$

where the probability is over the data distribution. Equation (1) expresses that the probability of being correct when the confidence is around $p$ is indeed $p$. For instance, if we consider the set of predictions

with a confidence of $90\%$, they should be correct $90\%$ of the time. The conditional probability of (1) is not rigorously defined mathematically (the event $\{s = p\}$ has zero probability), and interval-based empirical estimators are often used to define metrics capable of evaluating how well (1) is satisfied. This is the case of ECE, which approximates the calibration error by partitioning the confidence distribution into $B$ bins. The absolute difference between the accuracy and confidence is computed for each subset of data in the bins. The final value is a weighted sum of the differences of each bin.

$$\text{ECE} = \sum_{b=1}^{B} \frac{n_b}{N} |\text{acc}(b) - \text{conf}(b)| \qquad (2)$$

Where $n_b$ is the number of samples in bin $b$, $N$ is the total number of samples, $\text{acc}(b)$ is the accuracy in bin $b$, and $\text{conf}(b)$ is the average confidence in bin $b$. ECE can be interpreted visually by looking at diagrams such as those of Figure 1: ECE computes the sum of the red bars (difference between bin accuracy and average confidence) weighted by the proportion of samples in the bin.

**Post-processing calibration methods**   We are considering the scenario where a classifier has already been trained, and the objective is to enhance its calibration. Post-processing calibration methods aim to remap the classifier probabilities to better-calibrated values without modifying the classifier. They typically use a calibration set different from the training set to optimize parameters or learn a function. We note the calibration data $D_{cal} = \{(x_i, y_i)\}_{i=1}^{N}$. We focus on post-processing calibration because it enables better utilization of off-the-shelf models and separates model training (optimized for accuracy) from calibration. These advantages significantly reduce the development cost of obtaining a well-performing and well-calibrated model, contrary to optimizing calibration during training. We categorize the post-processing calibration techniques considered in this paper into two groups: scaling methods and binary methods.

## 3.2   Issues related to current approaches

**Behavior of current scaling methods**   Scaling methods for calibration optimize one or more coefficients that scale the logits vector to minimize on calibration data the cross-entropy loss defined as $l_{CE} = -\sum_{k=1}^{L} 1_{k=y} \cdot \log(f_k(x)) = -\log(f_y(x))$. Minimizing $l_{CE}$ therefore increases the probability of the true class. We can distinguish two cases to understand what happens during the optimization: whether the prediction $\hat{y}$ is correct or not. In the first case, the confidence score is $s = f_y(x)$: minimizing $l_{CE}$ increases the confidence $f_y(x)$. In the second case, the prediction is incorrect, which implies that $f_y(x) < s$. Minimizing $l_{CE}$ increases the probability of the true class $f_y(x)$ but does not directly change the confidence (because $s \neq f_y(x)$). Instead, the confidence (which was attributed to a wrong class) is *indirectly* lowered through the softmax normalization.

↪ ***Issue 1:*** *Cross-entropy loss only indirecly lowers confidence in wrong predictions.*

We identified another issue of some scaling methods. By design, the number of parameters optimized by Vector Scaling and Dirichlet Calibration grows with the number of classes. When the number of classes is high, these methods overfit the calibration set as shown in Figure 2.

↪ ***Issue 2:*** *Vector Scaling and Dirichlet Calibration overfit calibration sets with many classes.*

**One-versus-All approach for binary methods**   The One-versus-All (OvA) calibration approach [70] allows adapting calibration methods for binary classifiers to multiclass classifiers. To do so, it decomposes the calibration of multiclass classifiers into sets of $L$ binary calibration problems: one for each class $k$. For each problem, the considered probability is $f_k(x)$, and the associated label $1_{y=k} \in \{0, 1\}$. When calibrating a classifier from data, each binary problem is highly imbalanced with a ratio between positive and negative examples equal to $\frac{1}{L-1}$ if the classes are equally sampled. For instance, for ImageNet, the ratio is 1/999: out of 25000 examples, only 25 have a positive label.

↪***Issue 3:*** *OvA approach leads to highly imbalanced binary problems.*

At test time, each of the $L$ class probabilities is calibrated by a separate calibration model. The resulting probability vector can be normalized to ensure a unit norm. Because each probability is calibrated independently, their ranking can change, thus modifying the predicted class. In Table 9, we see that accuracy is often negatively impacted in practice.

↪***Issue 4:*** *OvA approach can change the predicted class and negatively impact the accuracy.*

# 4 Top-versus-All approach to confidence calibration

---

**Algorithm 1** Top-versus-All approach to confidence calibration

---

**Input**:
$D_{cal}$: $\{(x_i, y_i)\}_{i=1}^N$ the calibration data
$f$: the multiclass classifier
$g$: a calibration function $\qquad\qquad\qquad\qquad\qquad$ ▷ *e.g., Temperature Scaling*

**Preprocessing**:
$\hat{y}_i \leftarrow \arg\max_{k \in \mathcal{Y}} f_k(x_i)$ $\qquad\qquad\qquad\qquad$ ▷ *Compute class predictions*
$y_i^b \leftarrow 1_{\hat{y}_i = y_i}$ $\qquad\qquad\qquad\qquad\qquad$ ▷ *Compute predictions correctness*
$f^b \leftarrow \max_{k \in \mathcal{Y}} f_k$ $\qquad\qquad\qquad\qquad$ ▷ *Create surrogate binary classifier*
$D_{cal}^{\mathrm{TvA}} \leftarrow \{(x_i, y_i^b)\}_{i=1}^N$ $\qquad\qquad\qquad\qquad$ ▷ *Build binary calibration set*

**Learn calibration function**:
Learn $g$ to calibrate the surrogate binary classifier $f^b$ on $D_{cal}^{\mathrm{TvA}}$

**Inference**:
Use $g$ to calibrate the confidences of the original multiclass classifier $f$

---

## 4.1 General presentation

In the calibration definition (1) and the standard ECE metrics, only the confidence, i.e., the maximal probability, reflects the likelihood of making an accurate prediction. The probabilities of other classes are not taken into account. However, the standard approach to calibration uses the entire set of probabilities, not just confidence, which introduces unnecessary complexity. We aim to simplify the process by reformulating the problem of calibrating multiclass classifiers into a *single* binary problem. This problem can be phrased as: "Is the prediction correct?". In this setting, we do not calibrate the predicted probabilities vector but only a scalar: the confidence. The remaining probabilities are discarded. This is equivalent to calibrating a *surrogate* binary classifier that predicts whether the class prediction is correct. Since this correctness classifier only considers the maximal probability versus all others, we call our approach *Top-versus-All* (TvA).

Replacing the standard approach by TvA is straightforward. Given the standard calibration data $D_{cal} = \{(x_i, y_i)\}_{i=1}^N$, we add a few data preprocessing steps. First, compute the class predictions $\hat{y}$ and their correctness: $y^b = 1_{\hat{y}=y}$. Second, create the surrogate binary classifier $f^b(x) = \max_{k \in \mathcal{Y}} f_k(x)$. Finally, build the calibration set for the surrogate binary classifier:

$$D_{cal}^{\mathrm{TvA}} = \{(x_i, y_i^b)\}_{i=1}^N \tag{3}$$

After this preprocessing, we choose a standard calibration function $g$, e.g., Temperature Scaling, to calibrate the surrogate binary classifier. The learning of the calibration function follows its original underlying algorithm but uses the modified calibration data $D_{cal}^{\mathrm{TvA}}$. The learned calibration function is then applied to the confidences of the original multiclass classifier. Algorithm 1 describes our approach. In the Appendix, Algorithm 3 provides more details and highlights differences with the standard approach of Algorithm 2.

After this general presentation, we explain how TvA impacts the two categories of calibration methods, scaling and binary, in Subsections 4.2 and 4.3, respectively. We also justify its behavior.

## 4.2 Top-versus-All approach for scaling methods

Because our Top-versus-All setting reformulates the calibration of multiclass classifiers into a binary problem, the natural loss is the binary cross-entropy:

$$l_{BCE} = -\big(y^b \cdot \log s + (1 - y^b) \cdot \log(1 - s)\big) \tag{4}$$

Minimizing this loss results in confidence estimates that more accurately describe the probability of being correct, regardless of the $L - 1$ less likely class predictions. Using the binary cross-entropy as a calibration loss makes an important difference compared to the usual multiclass cross-entropy. The cross-entropy loss takes into account the probability of the *correct* class, while with TvA the binary cross-entropy takes into account the probability of the *predicted* class (i.e., the confidence).

As for the standard approach, only two cases are possible. When the prediction is correct, $l_{BCE} = -\log(s) = -\log(f_y) = l_{CE}$. We get the same result as the cross-entropy loss: minimizing it directly increases the confidence. But when the prediction is incorrect, $l_{BCE} = -\log(1-s) \neq l_{CE}$. Minimizing the loss now *directly* decreases the confidence. This is a key difference compared to using the multiclass cross-entropy loss.

The impact of the reformulation can be seen for Temperature Scaling, which optimizes a coefficient $T$ that scales the logits $z_k$. The reformulation generates stronger gradients when the prediction is incorrect:

$$\left| \frac{\partial l_{BCE}}{\partial T} \right| > \left| \frac{\partial l_{CE}}{\partial T} \right| \text{ for } s > 0.5 \tag{5}$$

with $\frac{\partial l_{BCE}}{\partial T} = \frac{1}{T^2} \cdot \frac{1}{s-1} \cdot (\max_k z_k - \sum_k z_k \cdot f_k)$ and $\frac{\partial l_{CE}}{\partial T} = \frac{1}{T^2}(z_y - \sum_k z_k \cdot f_k)$. See Appendix D for the mathematical calculations. Because for interesting problems, the confidence verifies $s > 0.5$ most of the time (as shown in Figure 1), our approach strengthens the gradients. The optimization of the temperature $T$ is more efficient as confident incorrect predictions are more heavily penalized. This effect is not mitigated by the choice of learning rate, which does not vary with $s$. Applying standard Temperature Scaling usually results in overconfident probabilities, but our approach limits this overconfidence. This is verified experimentally in Table 8, which displays the average confidences for TS without and with TvA.

↪**Solution for Issue 1:** *Use the binary cross-entropy loss resulting from TvA approach.*

**Regularization of scaling methods**  Overfitting of Vector Scaling and Dirichlet Calibration can be reduced with a simple L2 regularization that penalizes the coefficients of the vector $v$ that are far from the reference value 1.

$$l_{reg}(v) = \frac{1}{L} \sum_{i=1}^{L} (v_i - 1)^2 \tag{6}$$

This regularization allows these methods to take advantage of their additional expressiveness without being subject to overfitting. The loss for Vector Scaling becomes $l_{BCE} + \lambda l_{reg}(v)$ where $\lambda$ is a hyperparameter. The loss for Dirichlet Calibration uses additional matrix regularization terms. $\lambda$ is the only additional hyperparameter introduced by our method, and it applies only to Vector Scaling and Dirichlet Calibration.

↪**Solution for Issue 2:** *Use L2 regularization.*

## 4.3   Top-versus-All approach for binary methods

Our TvA approach replaces the One-versus-All approach to apply binary methods to the multiclass setting. TvA transforms the multiclass setting into a *single* binary problem that uses the binary calibration dataset (3). In this dataset, the proportion of positive labels equals the classifier's accuracy $a$. The ratio between negative and positive examples is $\frac{(1-a)N}{aN} = \frac{1}{a} - 1$. For a classifier with 80% accuracy on ImageNet and a calibration dataset of 25000 examples, there are 5000 negative and 20000 positive examples (ratio of 1/4). This is still a bit imbalanced but orders of magnitude smaller than the class-wise binary calibration datasets of the One-versus-All approach (ratio of 1/999).

↪**Solution for Issue 3:** *By not dividing the calibration data into class-wise datasets, the TvA approach yields a much better balanced binary calibration problem.*

The Top-versus-All approach operates on confidence alone, not the full class probabilities vector. This means that the class prediction is already done, and the ranking of the class probabilities does not change, unlike with One-versus-All. The classifier's prediction and accuracy are unaffected. This scheme allows decoupling accuracy improvements (during training time) and calibration (during post-processing calibration), thus avoiding compromises and reducing development time.

↪**Solution for Issue 4:** *By operating on confidence alone, the Top-versus-All approach does not impact the classifier's prediction or accuracy for binary methods applied to the multiclass scenario.*

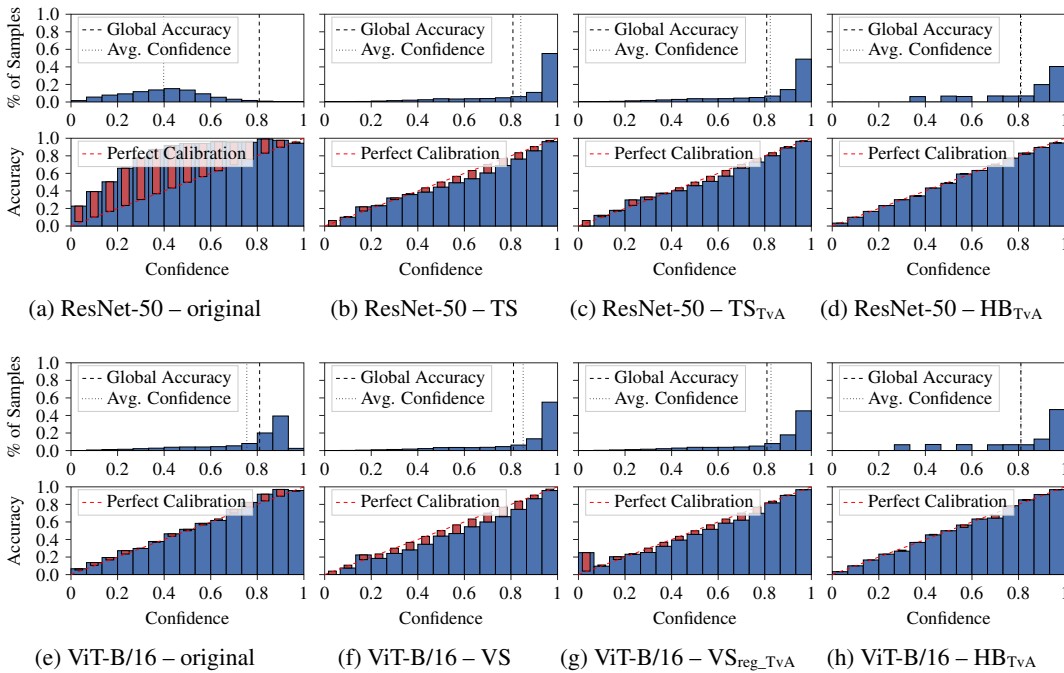

Figure 1: Reliability diagrams for ResNet-50 and ViT-B/16 when using Temperature Scaling (TS), Vector Scaling (VS), and Histogram Binning (HB) on ImageNet. The subscript $_{TvA}$ signifies that the TvA reformulation was used, and $_{reg}$ means our regularization (6) was applied. Red bars show the differences between bin accuracy (blue bar) and accuracy for perfect calibration (dashed red line). As the methods improve the calibration, these differences are reduced and the average confidence (vertical dotted line) will get closer to the global accuracy (vertical dashed line).

## 5 Experiments

### 5.1 Setting

**Datasets and models**    For image classification, we used the datasets *CIFAR-10 (C10)* and *CIFAR-100 (C100)* [27] with 10 and 100 classes respectively, *ImageNet (IN)* [7] with 1000 classes, and *ImageNet-21K (IN21K)* [54] with 10450 classes. For text classification, we used *Amazon Fine Foods (AFF)* [43] and *DynaSent (DF)* [51] for sentiment analysis with 3 classes, *MNLI* [66] for natural language inference with 3 classes, and *Yahoo Answers (YA)* [73] for topic classification on 10 classes. Experiment results are averaged over five random seeds that randomly split the concatenation of the original validation and test sets into calibration and test sets.

We used the following models for image classification: *ResNet* [21], *Wide-ResNet-26-10 (WRN)* [71], *DenseNet-121* [24], *MobileNetV3 (MN3)* [23], *ViT* [9], *ConvNeXt* [41], *EfficientNet* [56, 57], *Swin* [40, 39], and *CLIP* [52] which matches input images to text descriptions in a shared embedding space, assigning labels based on the highest similarity score. For text classification, we used the PLMs *RoBERTa* [37] and *T5* [53].

More details about datasets, calibration sets sizes, and model weights can be seen in Appendix F.

**Baselines**    Our Top-versus-All ($_{TvA}$) reformulation and regularization ($_{reg}$) can be applied to different calibration methods. We have tested the following scaling methods: *Temperature Scaling (TS)* and *Vector Scaling (VS)* [17], and *Dirichlet Calibration (DC)* [29] with the best-performing variant Dir-ODIR, which regularizes off-diagonal and bias coefficients. We also tested the following binary methods: *Histogram Binning (HB)* [69] using for each case the best-performing variant between equal-mass or equal-size bins, *Isotonic Regression (Iso)* [70], *Beta Calibration (Beta)* [28], and *Bayesian Binning into Quantiles (BBQ)* [46]. For comparison, we include methods with state-of-the-art results on problems with many classes: *I-Max* [49] and *IRM* [72]. See Appendix C for more details on these methods. More details on code implementations can be seen in Appendix F.

Table 1: ECE in % (lower is better). The subscript $_{TvA}$ denotes that our reformulation was applied to the calibration method. IRM and I-Max are competing methods. The best method for a given model is in bold. Methods in purple impact the model prediction, potentially degrading accuracy; methods in teal do not. Values are averaged over five random seeds. Results are averaged over models of the same family. Detailed results for all models can be seen in Tables 5 and 6 of the Appendix.

| Dataset | Models | Uncal. | IRM | I-Max | scaling methods | | | | | | binary methods | | | | | |
|---|---|---|---|---|---|---|---|---|---|---|---|---|---|---|---|---|
| | | | | | TS | TS$_{TvA}$ | VS | VS$_{reg\_TvA}$ | DC | DC$_{reg\_TvA}$ | Iso | Iso$_{TvA}$ | BBQ | BBQ$_{TvA}$ | HB | HB$_{TvA}$ |
| C10 | ConvNets | 1.77 | 0.67 | 0.61 | 1.20 | 1.25 | 1.17 | 1.17 | 1.16 | 1.17 | 1.12 | 0.68 | 1.17 | 0.77 | 1.06 | **0.43** |
| | CLIP | 5.03 | 1.03 | 0.94 | 0.78 | **0.73** | 2.56 | 1.85 | 2.56 | 1.84 | 1.05 | 0.86 | 1.39 | 0.86 | 1.82 | **0.73** |
| C100 | ConvNets | 6.04 | 1.30 | 1.15 | 4.65 | 2.96 | 4.91 | 2.35 | 4.89 | 2.35 | 5.33 | 1.38 | 9.63 | 1.35 | 9.56 | **1.02** |
| | CLIP | 10.37 | 2.90 | 2.57 | 2.54 | 2.51 | 7.78 | 2.86 | 7.55 | 1.84 | 2.53 | 1.61 | 7.48 | 1.48 | 7.23 | **1.39** |
| IN | ResNet | 15.26 | 1.31 | 1.07 | 2.65 | 1.89 | 2.77 | 1.67 | 3.59 | 2.23 | 3.05 | 0.79 | 8.41 | 0.76 | 7.49 | **0.55** |
| | EffNet | 15.72 | 0.68 | 0.48 | 3.48 | 2.59 | 3.67 | 1.26 | 3.65 | 1.23 | 2.83 | 0.68 | 6.55 | 0.64 | 4.39 | **0.43** |
| | ConvNeXt | 16.46 | 0.82 | 0.58 | 3.67 | 2.25 | 4.05 | 1.37 | 4.04 | 1.35 | 2.97 | 0.75 | 7.41 | 0.68 | 5.13 | **0.52** |
| | ViT | 4.40 | 0.81 | 0.61 | 4.09 | 2.96 | 4.31 | 2.02 | 4.31 | 1.99 | 3.60 | 0.77 | 6.64 | 0.73 | 6.59 | **0.52** |
| | Swin | 5.85 | 0.75 | 0.49 | 3.63 | 2.91 | 4.04 | 1.70 | 4.03 | 1.67 | 3.19 | 0.74 | 7.09 | 0.71 | 5.39 | **0.48** |
| | CLIP | 1.96 | 1.08 | **0.72** | 1.89 | 1.82 | 1.63 | 1.05 | 32.03 | 67.65 | 2.35 | 0.92 | 8.31 | 0.93 | 7.16 | 0.80 |
| IN21K | MN3 | 12.34 | err. | err. | 8.69 | 4.39 | 2.52 | 2.40 | 58.84 | 81.16 | 2.00 | 0.21 | err. | 0.20 | 5.50 | **0.17** |
| | ViT-B/16 | 6.27 | err. | err. | 8.92 | 6.55 | 2.38 | 1.54 | 8.22 | 3.20 | 2.14 | 0.22 | err. | 0.24 | 7.89 | **0.12** |
| AFF | T5 | 5.47 | 0.27 | 0.26 | 1.10 | 1.15 | 1.52 | 1.42 | 1.18 | 1.31 | 0.37 | 0.27 | 0.39 | 0.28 | 2.87 | **0.17** |
| | RoBERTa | 7.37 | 0.30 | 0.28 | 2.40 | 2.33 | 1.41 | 1.85 | 1.38 | 1.68 | 0.52 | 0.27 | 0.75 | 0.35 | 4.02 | **0.20** |
| DS | T5 | 8.86 | 1.39 | 1.38 | 2.19 | 2.17 | 6.13 | 2.00 | 5.91 | 2.02 | 1.50 | 1.55 | 1.38 | 1.58 | 1.90 | **1.12** |
| | RoBERTa | 16.12 | 1.56 | 1.50 | 12.07 | 12.07 | 14.66 | 6.80 | 13.90 | 5.57 | 1.71 | 1.53 | 1.64 | 1.14 | 1.05 | **0.91** |
| MNLI | T5 | 7.04 | 0.72 | 0.70 | 2.81 | 2.80 | 4.46 | 1.79 | 4.31 | 1.82 | 0.80 | 0.74 | 1.38 | 0.69 | 2.09 | **0.43** |
| | RoBERTa | 9.22 | 0.89 | 0.71 | 5.72 | 5.72 | 6.99 | 1.92 | 6.59 | 1.99 | 1.00 | 0.92 | 1.67 | 0.84 | 1.02 | **0.60** |
| YA | T5 | 7.84 | 0.80 | 0.81 | 1.07 | 1.35 | 3.70 | 1.16 | 3.75 | 1.15 | 1.73 | 0.82 | 2.81 | 0.96 | 3.65 | **0.69** |
| | RoBERTa | 19.59 | 0.97 | 0.79 | 12.39 | 12.38 | 16.47 | 2.52 | 16.07 | 2.21 | 1.92 | 0.99 | 5.00 | 0.75 | 3.41 | **0.58** |

## 5.2 Top-versus-All

For visual qualitative results, Figure 1 displays reliability diagrams [47]. We observe that initially, ResNet-50 is highly underconfident and ViT-B/16 a bit underconfident. Applying TS and VS solves the underconfidence and makes the models slightly overconfident. TvA improves these methods, and the average confidence gets closer to the accuracy. HB$_{TvA}$ makes the calibration almost perfect.

Table 1 shows the results of applying the Top-versus-All reformulation to several calibration methods. For clarity, results are averaged over families of models (models based on the same architecture) and the full results are available in Tables 5 and 6 of the Appendix. In most cases, the TvA reformulation significantly lowers the ECE by dozens of percent. Without TvA, binary methods often perturb the prediction and degrade the classifier's accuracy (see Table 9), making them inapplicable in a practical setting. TvA solves the issue as it only scales the confidence (after the prediction is made) and makes binary methods outperform scaling methods.

Improvements due to TvA are consistent across models. However, exceptions are observed for CLIP: it is the model family with the lowest ECE pre-calibration, but the highest ECE post-calibration for ImageNet. CLIP's multimodal training regime, zero-shot adaptation as a classifier, and very large training dataset might cause this different behavior. CLIP's low ECE was also observed in [44, 11]. [64] specifically tackles the calibration of fine-tuned CLIP, a setting not considered here.

We also found that DC is sensitive to hyperparameter tuning, and its performance is usually not much better than VS, which is consistent with [29]. In some cases, the optimization diverges, leading to very poor results, e.g., for CLIP on ImageNet.

Improvements due to TvA are also consistent across datasets, although they tend to increase with the number of classes. Improvements on ImageNet are usually better than on CIFAR-100, whose improvements are usually better than on CIFAR-10. This is notable with e.g., TS or HB. The magnitude of improvement is usually higher for binary methods, e.g., HB, than scaling methods, e.g., TS, especially for many classes. This indicates that Issue 3 is more serious than Issue 1.

For text datasets with only three classes (AFF, DS, and MNLI), TS does not benefit from TvA, but other methods do, despite the small number of classes. According to [5], TS is among the best calibration methods for the text classification tasks considered here, even compared to ones that retrain the model. Even so, our method HB$_{TvA}$ significantly outperforms it.

Some methods' current implementations could not handle the large scale of ImageNet-21K, resulting in out-of-memory errors written as "err." in the Table. For I-Max and IRM, this is because they consider the full probability vectors while TvA efficiently uses data by considering only confidence values. Indeed, TvA handles this scale without difficulty.

Additional results are included in Appendix H. Tables 5 and 6 contain the full results for ECE, with the standard deviations in Table 7. Table 8 reveals that ImageNet networks are mostly underconfident. This is aligned with [11] and goes against previous knowledge on overconfidence, which was initially believed to be linked to network size [17]. Table 9 provides the accuracies after calibration. Table 10 exhibits that ECE with equal-mass bins has similar values as standard ECE. Table 11 shows that TvA mostly lowers the Brier score, except for Iso, which has the lowest score overall.

Calibration methods can also be applied to Large Languages Models (LLMs) using In-Context Learning (ICL) to tackle text classification tasks [77, 20, 25, 78, 1]. The primary goal of these methods is to improve model accuracy. TvA was not designed for this objective, but it can still be applied on top of an existing method that improves the accuracy. TvA then lowers the calibration error while keeping the accuracy gain. Results for GPT-J [62] and Llama-2 [59] are in Table 12.

To summarize the results for practical use, our experiments show that Histogram Binning (within the TvA or I-Max setting) is the best calibration method overall, providing ECE values mostly below 1%. This is the method we advise using. However, suppose the underlying application requires a confidence with continuous values, e.g., to rank the predictions for selective classification. In that case, we advise using a method that improves the AUROC, shown in Appendix G, such as TS or Iso.

### 5.3 Solving overfitting with regularization and TvA

On ImageNet, VS and DC overfit the calibration set, degrading the calibration on the test set. The lower performance of VS relative to TS indicates this overfitting. As visualized in Figure 2, combining the binary cross-entropy loss used in the TvA reformulation and an additional regularization term prevents overfitting. We fixed the value $\lambda = 0.01$ as it works well across models. Initializing the vector coefficients to $\frac{1}{T}$ with $T$ obtained by $TS_{TvA}$ helps further improve performance.

### 5.4 Influence of the calibration set size

The size of the calibration set influences the performance of the different methods, as seen in Figure 3. TS and $TS_{TvA}$ do not benefit from more data due to their low expressiveness. VS does not improve the ECE because of the overfitting problem. In contrast, $VS_{reg\_TvA}$ benefits from more calibration data. With enough data ($\approx 15000$), it outperforms $TS_{TvA}$. Binary methods using the standard One-versus-All approach have poor performance and need a large amount of data to be competitive. Using TvA, they get excellent performance with little data.

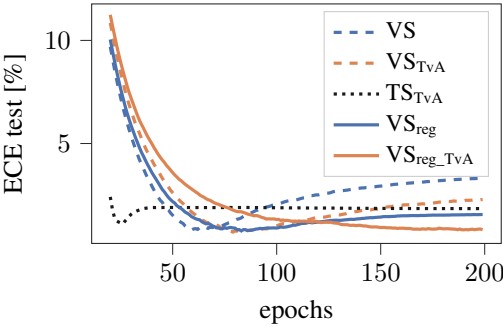

Figure 2: Test ECE evolution during training with ResNet-50 on ImageNet. The combination of regularization and TvA prevents overfitting of Vector Scaling. Temperature Scaling with TvA is shown for reference.

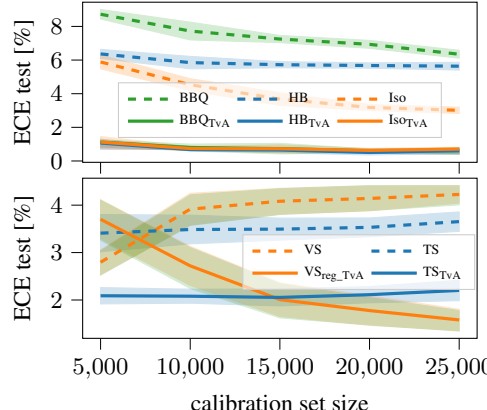

Figure 3: Influence of the calibration set size for ResNet-101 on ImageNet. Binary methods at the top and scaling methods at the bottom.

# 6  Limitations

Our approach tackles confidence calibration and is unlikely to improve performance for stronger notions of calibration, such as class-wise calibration. However, confidence calibration is useful for many practical cases, such as selective classification [13], out-of-distribution detection [22], or active learning [34]. Also, calibration improvements are less significant for problems with few classes ($\leq 10$) than for problems with many classes, but our approach still provides the best results.

# 7  Conclusion

Reducing the miscalibration of neural networks is essential to improve trust in their predictions. This can be done after the model training with an optimization using calibration data. However, many current calibration methods do not scale well to complex datasets: binary methods under the One-versus-All setting do not have enough per-class calibration data, and scaling methods are inefficient. We demonstrate that reformulating the confidence calibration of multiclass classifiers as a single binary problem significantly improves the performance of baseline calibration techniques. The competitiveness of scaling methods is increased, and binary methods use per-class calibration data more efficiently without altering the model's accuracy. In short, our TvA reformulation enhances many existing calibration methods with little to no change in their algorithm. Extensive experiments with state-of-the-art image classification models on complex datasets and with text classification demonstrate our approach's scalability and generality.

## Acknowledgments and Disclosure of Funding

We thank Tahar Nabil, Houssem Ouertatani, and Pol Labarbarie for their constructive feedback on the paper draft.
This work has been supported by the French government under the "France 2030" program, as part of the SystemX Technological Research Institute.
This work was granted access to the HPC/AI resources of IDRIS under the allocation 2024-AD011013372R1 made by GENCI.

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

# A    Appendix contents

# B    Broader impacts

Our reformulation of the confidence calibration of multiclass classifiers as a binary problem is both simple and general. It has several benefits. On the theoretical side, it might lead to new perspectives on the confidence calibration problem and the development of new calibration methods. On the practical side, existing calibration methods can be adapted to our problem reformulation by adding just a few lines of code. This is an easy and quick way to improve the calibration of classification models. Better-calibrated models are more trustworthy: potential incorrect predictions are more easily identifiable and preventable. However, this also comes with potential risks. The knowledge that a model is well-calibrated might lead to undue trust in the system and the tendency to overlook prediction errors. Even well-calibrated models are not entirely reliable, and developers and users must remember this. Post-processing calibration requires data not included in the training set, which leaves less data available for a thorough evaluation of the model. Calibration does not fix biases in the data. Finally, we tested the calibration improvement only on in-distribution data, but real systems might receive out-of-distribution data (e.g., an image of a new class) or adversarial examples. For such inputs, the classifier predicted probabilities (and thus the confidence) are unreliable, even for well-calibrated models, and a pipeline to filter such data is necessary.

# C    Details on the method

---

**Algorithm 2** Standard approach

---

**Input**:
$D_{cal}$: $\{(x_i, y_i)\}_{i=1}^N$ the calibration data
$f$: the multiclass classifier
$g$: a calibration function ▷ *e.g., Temperature Scaling*

**Learn calibration function**:
**if** $g$ **is** scaling method **then**
    loss $l :=$ Cross-Entropy
    Learn $g$ to calibrate $f$ by minimizing $l$ on $D_{cal}$
**else if** $g$ **is** binary method **then**
    **for** $k = 1$ to $L$ **do**        ▷ *One-versus-All approach*
        $D_{cal}^k \leftarrow \{(x_i, y_i) \mid y_i = k\}_{i=1}^N$
        Learn $g_k$ to calibrate $f$ on $D_{cal}^k$
    **end for**
    $g \leftarrow (g_1, g_2, \ldots, g_L)$
**end if**

**Inference**:
Use $g$ to calibrate confidences from $f$

---

---

**Algorithm 3** Top-versus-all approach

---

**Input**:
$D_{cal}$: $\{(x_i, y_i)\}_{i=1}^N$ the calibration data
$f$: the multiclass classifier
$g$: a calibration function        ▷ *e.g., Temperature Scaling*
**Preprocessing**:
$\hat{y}_i \leftarrow \arg\max_{k \in \mathcal{Y}} f_k(x_i)$    ▷ *Compute class predictions*
$y_i^b \leftarrow \mathbb{1}_{\hat{y}_i = y_i}$        ▷ *Compute predictions correctness*
$f^b \leftarrow \max_{k \in \mathcal{Y}} f_k$    ▷ *Create surrogate binary classifier*
$D_{cal}^{\text{TvA}} \leftarrow \{(x_i, y_i^b)\}_{i=1}^N$        ▷ *Build binary calib. set*
**Learn calibration function**:
**if** $g$ **is** scaling method **then**
    loss $l :=$ Binary Cross-Entropy
    **if** $g$ **is** vector or Dirichlet scaling
        loss $l \leftarrow l + \lambda l_{reg}$        ▷ *Add regularization*
    **end if**
    Learn $g$ to calibrate $f^b$ by minimizing $l$ on $D_{cal}^{\text{TvA}}$
**else if** $g$ **is** binary method **then**
    Learn $g$ to calibrate $f^b$ on $D_{cal}^{\text{TvA}}$
**end if**

**Inference**:
Use $g$ to calibrate confidences from $f$

---

**Comparison with the standard approach** Algorithm 2 describes the standard approach to post-processing calibration, and Algorithm 3 describes our approach in more details and shows in blue the differences with the standard approach. Our approach adds a preprocessing step to keep only the confidences instead of the full probabilities vector. It can be seen as creating a surrogate "correctness" classifier and its associated calibration data. The calibrator is learned for the surrogate classifier and applied to the original classifier at inference time. Also, we add regularization for some scaling methods and we have only one binary calibrator instead of one per class.

**Comparison with IRM and I-Max** IRM [72] and I-Max [49] are, like TvA, multiclass-to-binary reductions. This is why TvA cannot be applied on top of them: they already transform the multiclass problem into a binary one using a different strategy.

The shared class-wise strategy of [49] and the data ensemble strategy of [72] are described very briefly in subsections 3.2 and 3.3.2 of their respective papers and not rigorously justified. Our understanding is that these two strategies do exactly the same thing. To build the calibration set, they concatenate all the class probability vectors so that we get a big probability vector of size $N.L$ ($N$ samples and $L$ classes) as predictions and similarly concatenate the one-hot embedding of the target class (a big vector with $N$ ones and $N.(L-1)$ zeros) as targets. Then, they learn a single calibrator. For each example, this calibrator aims to simultaneously increase the probabilities for the target class (target is 1) and decrease all the other class probabilities (target is 0). The single calibrator is applied to each class probability separately, meaning that the ranking of class probabilities can change, modifying the classifier prediction.

Our strategy derives from transforming the multiclass calibration into a single binary problem. The intuition is to learn the calibrator on a surrogate binary classifier and apply this calibrator to the original classifier. This binary classifier is built on top of the original classifier (by applying the max function to the class probabilities vector). They thus share their confidence. However, the binary classifier aims to solve a different task: predicting the correctness of the original classifier. To build the calibration set, we concatenate all the confidences (a vector of size N) as predictions and concatenate all the correctnesses as targets (also a vector of size N). The correctness value of a given example is 1 if the class prediction is correct; otherwise, it is 0. Then, we learn a single calibrator, similar to the strategy above. However, there is a key difference: this calibrator aims to increase the probabilities for correct predictions and decrease them for incorrect predictions. Note that our probabilities are all confidences (the maximum class probabilities), meaning we only consider the confidences, which the calibrator directly increases or decreases. In the strategy from [49] and [72], the calibrator has to manage all class probabilities (L times more), even the ones that do not matter, including the lowest class probabilities close to 0. This is less efficient (actually, while this can surely be fixed, the original implementations of IRM and I-Max could not run on ImageNet-21K). This point is closely linked to the analysis of the binary cross entropy loss for scaling methods in Subsection 4.2: when the prediction is incorrect, increasing the probability of the correct class indirectly decreases the confidence (strategy from [49] and [72]) while our strategy directly decreases the confidence.

I-Max is more complex because it modifies the Histogram Binning algorithm, while our approach does not. Additionally, [35] found that I-Max produces unusable probability vectors. Indeed, they do not sum up to 1, and normalizing them degrades the method's performance.

We wrote our paper with practicality and generality in mind. Contrary to [49] and [72], we demonstrate the generality of our strategy by applying it on top of existing calibration baselines of different natures (scaling and binary). One of our main goals is that practitioners can easily and quickly try our TvA approach, using just a few lines of code, which can significantly improve the calibration performance of their existing calibration pipeline while having no impact on the predicted class by design (except for VS and DC).

**Comparison with ProCal** Another recent calibration method is ProCal [68]. However, its primary objective differs from ours: it "focuses on the problem of proximity bias in model calibration, a phenomenon wherein deep models tend to be more overconfident on data of low proximity". Its goal is to lower the difference in the confidence score values between regions of low and high density, i.e., to make the confidence score independent of a local density indicator called "proximity." There is no theoretical guarantee, however, that minimizing the proximity bias improves the confidence calibration, the focus of our work. Theorem 4.2 about the PIECE metric is a direct consequence of Jensen's inequality and is true for any random variable D, not necessarily a proximity score.

Theorem 5.1 is an interesting bias/variance decomposition of the Brier score. However, as this type of decomposition usually states, the error may come from bias (here, a wrong initial calibration) or high estimation variance (which can be related to low density but is not expressed as such in the decomposition). We experimentally compare our approach to the ProCal algorithm using the code provided by its authors and observe in Table 2 that our approach gives much better ECE confidence calibration and, for half of the models, also better PIECE values.

ProCal aims to achieve three goals: mitigate proximity bias, improve confidence calibration, and provide a plug-and-play method. We share the last two goals. Concerning improving confidence calibration, our approach has better results, as shown in Table 2. Both approaches are plug-and-play, but they apply very differently. ProCal is applied *after* existing calibration methods to further improve calibration. It thus does not solve any of the four issues we identified (e.g., cross-entropy loss is still inefficient, and One-versus-All still leads to highly imbalanced problems). Our Top-versus-All approach is a reformulation of the calibration problem that uses a surrogate binary classifier. Existing approaches are applied to this surrogate classifier, which is how the four issues are solved. We do not propose a new method but a new way of applying existing methods. Our approach does not introduce new hyperparameters (except in the particular case of regularizing scaling methods). ProCal introduces several new hyperparameters, such as the choice of the distance, the number of KNN neighbors, or a shrinkage coefficient.

**Comparison with Correctness-Aware Loss**    A concurrent work [38] builds a calibration method on top of an intuition similar to ours: binarize the calibration problem. However, what the authors do with this intuition differs vastly from our approach. They derive a Correctness-Aware Loss (Eq. 7 of their paper), which is almost the standard binary cross-entropy loss we use for scaling methods but without a logarithm. They use this loss to learn a separate model that predicts a sample-wise temperature coefficient. This is a new calibration method, which is not straightforward to implement due to the numerous hyperparameters (network architecture, image transformations...). It also requires multiple inferences at test-time, which can be problematic in some production models. Our approach is, again, not a calibration method but a general reformulation of the calibration problem that enhances existing methods. By looking at their Table 1, they get an ECE of 2.22 on ImageNet (in-distribution), while our approach achieves values around 0.5 for most models in our paper's Table 1. Their method, contrary to TvA, improves the AUROC, but in our understanding, it seems mostly due to the use of image transformations, not from their proposed loss. Their method seems to work best in out-of-distribution scenarios, which is not the main objective of our paper. However, these good results for AUROC and out-of-distribution scenarios make this method complementary to our approach, and combining the two in some way could be promising.

Table 2: ECE, MCE, ACE, and PIECE in %. The experimental setting is the one used for Table 4 of [68]. The baselines are no calibration (conf), Temperature Scaling (TS), Multi Isotonic Regression (MIR), and Histogram Binning (HB). The ProCal calibration method [68] is applied *after* one of the baselines, as symbolized by the "+" symbol. Our approach, Top-versus-All, changes what the baselines optimize and is symbolized by "$_{\text{TvA}}$". We apply it for TS, Isotonic Regression (Iso), and HB. The best values for each model and metric are in bold.

Overall, $\text{HB}_{\text{TvA}}$ is the best calibration method as it always gets the lowest ECE and ACE. Our TvA approach lowers PIECE and even achieves the lowest value for half of the models.

| Model | Method | ECE ($\downarrow$) | MCE ($\downarrow$) | ACE ($\downarrow$) | PIECE ($\downarrow$) |
|---|---|---|---|---|---|
| BeiT | conf | 3.60 | 1.52 | 3.58 | 4.26 |
| | TS | 2.99 | 0.76 | 3.08 | 3.56 |
| | MIR | 0.59 | **0.14** | 0.64 | 1.88 |
| | HB | 4.96 | 1.83 | 6.13 | 7.20 |
| | conf+ProCal | 1.02 | 0.33 | 0.94 | 1.69 |
| | TS+ProCal | 1.52 | 0.76 | 1.45 | 2.05 |
| | MIR+ProCal | 0.61 | 0.15 | 0.71 | **1.41** |
| | HB+ProCal | 5.53 | 4.13 | 5.81 | 6.39 |
| | $\text{TS}_{\text{TvA}}$ | 2.10 | 0.49 | 2.20 | 2.88 |
| | $\text{Iso}_{\text{TvA}}$ | 0.65 | **0.14** | 0.68 | 1.92 |
| | $\text{HB}_{\text{TvA}}$ | **0.44** | 0.19 | **0.58** | 1.70 |
| Mixer | conf | 10.78 | 5.00 | 10.78 | 10.86 |
| | TS | 4.81 | 1.92 | 4.72 | 5.57 |
| | MIR | 1.14 | 0.25 | 1.29 | 3.41 |
| | HB | 9.65 | 4.00 | 9.97 | 13.09 |
| | conf+ProCal | 2.64 | 0.94 | 2.55 | 3.23 |
| | TS+ProCal | 1.32 | 0.38 | 1.22 | 2.13 |
| | MIR+ProCal | 0.83 | **0.14** | 0.88 | **2.08** |
| | HB+ProCal | 6.57 | 4.43 | 7.32 | 7.83 |
| | $\text{TS}_{\text{TvA}}$ | 2.51 | 0.74 | 2.50 | 4.32 |
| | $\text{Iso}_{\text{TvA}}$ | 0.86 | 0.15 | 0.86 | 3.18 |
| | $\text{HB}_{\text{TvA}}$ | **0.64** | 0.19 | **0.75** | 3.17 |
| ResNet50 | conf | 8.59 | 4.58 | 8.50 | 8.74 |
| | TS | 5.03 | 2.48 | 5.01 | 5.34 |
| | MIR | 0.75 | 0.18 | 0.82 | 1.79 |
| | HB | 7.63 | 2.61 | 9.32 | 10.17 |
| | conf+ProCal | 2.63 | 1.31 | 2.61 | 3.26 |
| | TS+ProCal | 1.66 | 0.66 | 1.53 | 2.50 |
| | MIR+ProCal | 0.76 | **0.17** | 0.74 | 1.78 |
| | HB+ProCal | 6.32 | 4.38 | 7.52 | 7.65 |
| | $\text{TS}_{\text{TvA}}$ | 6.89 | 1.17 | 6.95 | 7.13 |
| | $\text{Iso}_{\text{TvA}}$ | 0.76 | 0.19 | 0.73 | 1.79 |
| | $\text{HB}_{\text{TvA}}$ | **0.55** | 0.21 | **0.59** | **1.35** |
| ViT | conf | 1.14 | 0.33 | 1.09 | 1.83 |
| | TS | 1.46 | 0.46 | 1.41 | 2.03 |
| | MIR | 0.64 | 0.13 | 0.75 | 1.54 |
| | HB | 4.59 | 2.32 | 7.07 | 7.20 |
| | conf+ProCal | 0.81 | 0.22 | 0.81 | 1.71 |
| | TS+ProCal | 0.83 | 0.25 | 0.84 | 1.74 |
| | MIR+ProCal | 0.78 | 0.14 | 0.76 | 1.59 |
| | HB+ProCal | 6.80 | 4.72 | 7.42 | 7.55 |
| | $\text{TS}_{\text{TvA}}$ | 1.08 | 0.25 | 1.05 | 1.86 |
| | $\text{Iso}_{\text{TvA}}$ | 0.51 | **0.11** | 0.62 | 1.46 |
| | $\text{HB}_{\text{TvA}}$ | **0.37** | 0.17 | **0.51** | **1.20** |

# D  Theoretical justification for Top-versus-All in the case of Temperature Scaling

We define $L$ the number of classes, $f_k(x)$ the classifier estimated probability for class $k$ and data sample $x$, $y$ the correct class, and the confidence $s(x) := \max_k f_k(x)$. The cross-entropy loss is $l_{CE}(x, y) = -\sum_{k=1}^{L} 1\{k = y\} \cdot \log(f_k(x)) = -\log(f_y(x))$. Because the last layer of the classifier is a softmax function, $f_y(x) = \frac{e^{z_y}}{\sum_k e^{z_k}}$ with $z$ the logits vector. Note that we omit the writing variable $x$ in the following for clarity.

Temperature scaling optimizes a coefficient $T > 0$ that scales the logits vector. Predicted probabilities become $f_y(x) = \frac{e^{z_y/T}}{\sum_k e^{z_k/T}}$.

Let us first develop the standard cross-entropy loss when temperature scaling is applied:

$$l_{CE} = -\log(f_y) = -\log\left(\frac{e^{z_y/T}}{\sum_k e^{z_k/T}}\right) = -\left(\log(e^{z_y/T}) - \log\left(\sum_k e^{z_k/T}\right)\right) = -\frac{z_y}{T} + \log\left(\sum_k e^{z_k/T}\right)$$

Let us compute its gradient:

$$\frac{\partial l_{CE}}{\partial T} = \frac{z_y}{T^2} + \frac{\partial \log(\sum_k e^{z_k/T})}{\partial \sum_k e^{z_k/T}} \cdot \frac{\partial \sum_k e^{z_k/T}}{\partial T} \text{ by application of the chain rule on the second term.}$$

$$= \frac{z_y}{T^2} + \frac{1}{\sum_k e^{z_k/T}} \cdot \sum_k \frac{\partial e^{z_k/T}}{\partial T}$$

$$= \frac{z_y}{T^2} + \frac{1}{\sum_k e^{z_k/T}} \cdot \sum_k \frac{\partial (e^{z_k})^{1/T}}{\partial T}$$

$$= \frac{z_y}{T^2} + \frac{1}{\sum_k e^{z_k/T}} \cdot \sum_k \frac{\log(e^{z_k})}{-T^2}(e^{z_k})^{1/T}$$

$$= \frac{z_y}{T^2} + \frac{1}{\sum_k e^{z_k/T}} \cdot \sum_k \frac{z_k \cdot e^{z_k/T}}{-T^2}$$

$$= \frac{1}{T^2}\left(z_y - \frac{\sum_k z_k \cdot e^{z_k/T}}{\sum_k e^{z_k/T}}\right)$$

$$= \frac{1}{T^2}\left(z_y - \sum_k \frac{z_k \cdot e^{z_k/T}}{\sum_j e^{z_j/T}}\right)$$

$$= \frac{1}{T^2}\left(z_y - \sum_k z_k \cdot f_k\right) \qquad (7)$$

For our TvA approach, the problem becomes binary. The classification output becomes the confidence $s(x) = \max_{j \in \mathcal{Y}} f_j(x)$ and the ground truth label becomes a binary representation of the prediction correctness: $y^b = 1_{\hat{y}=y}$ with $\hat{y}(x) = \arg\max_{k \in \mathcal{Y}} f_k(x)$ and $1$ the indicator function. The loss we use is the binary cross entropy $l_{BCE}(x, y) = -\left(y^b \cdot \log s(x) + (1 - y^b) \cdot \log(1 - s(x))\right)$

Let us compute the gradient:

$$\frac{\partial l_{BCE}}{\partial T} = \frac{\partial l_{BCE}}{\partial s} \cdot \frac{\partial s}{\partial T}$$

$$= -\left(y^b \frac{1}{s} + (1-y^b)\frac{-1}{1-s}\right) \cdot \frac{\partial s}{\partial T}$$

$$= -\left(\frac{y^b(1-s)}{s(1-s)} - \frac{s(1-y^b)}{s(1-s)}\right) \cdot \frac{\partial s}{\partial T}$$

$$= \frac{s-y^b}{s(1-s)} \cdot \frac{\partial s}{\partial T}$$

$$= \frac{s-y^b}{s(1-s)} \cdot \frac{\partial \frac{e^{z_m/T}}{\sum_k e^{z_k/T}}}{\partial T} \text{ because } s = \max_j \frac{e^{z_j/T}}{\sum_k e^{z_k/T}} = \frac{e^{z_m/T}}{\sum_k e^{z_k/T}} \text{ with } z_m = \max_k z_k$$

$$= \frac{s-y^b}{s(1-s)} \cdot \frac{\frac{\partial e^{z_m/T}}{\partial T}\sum_k e^{z_k/T} - e^{z_m/T}\frac{\partial \sum_k e^{z_k/T}}{\partial T}}{(\sum_k e^{z_k/T})^2}$$

$$= \frac{s-y^b}{s(1-s)} \cdot \frac{\frac{\log(e^{z_m})}{-T^2}(e^{z_m})^{1/T}\sum_k e^{z_k/T} - e^{z_m/T}\sum_k \frac{\log(e^{z_k})}{-T^2}(e^{z_k})^{1/T}}{(\sum_k e^{z_k/T})^2}$$

$$= \frac{s-y^b}{s(1-s)} \cdot \frac{e^{z_m/T}}{T^2} \cdot \frac{-\log(e^{z_m})\sum_k e^{z_k/T} + \sum_k \log(e^{z_k})(e^{z_k})^{1/T}}{(\sum_k e^{z_k/T})^2}$$

$$= \frac{s-y^b}{s(1-s)} \cdot \frac{e^{z_m/T}}{T^2} \cdot \frac{-z_m\sum_k e^{z_k/T} + \sum_k z_k \cdot e^{z_k/T}}{(\sum_k e^{z_k/T})^2}$$

$$= \frac{s-y^b}{s(1-s)} \cdot \frac{1}{T^2} \cdot s \cdot \frac{-z_m\sum_k e^{z_k/T} + \sum_k z_k \cdot e^{z_k/T}}{\sum_k e^{z_k/T}}$$

$$= \frac{1}{T^2} \cdot \frac{s-y^b}{1-s} \cdot \frac{-z_m\sum_k e^{z_k/T} + \sum_k z_k \cdot e^{z_k/T}}{\sum_k e^{z_k/T}}$$

$$= \frac{1}{T^2} \cdot \frac{s-y^b}{1-s} \cdot \left(\frac{\sum_k z_k \cdot e^{z_k/T}}{\sum_k e^{z_k/T}} - z_m\right)$$

$$= \frac{1}{T^2} \cdot \frac{y^b-s}{1-s} \cdot \left(z_m - \frac{\sum_k z_k \cdot e^{z_k/T}}{\sum_k e^{z_k/T}}\right)$$

$$= \frac{1}{T^2} \cdot \frac{y^b-s}{1-s} \cdot \left(z_m - \sum_k \frac{z_k \cdot e^{z_k/T}}{\sum_j e^{z_j/T}}\right)$$

$$= \frac{1}{T^2} \cdot \frac{y^b-s}{1-s} \cdot \left(z_m - \sum_k z_k \cdot f_k\right) \tag{8}$$

- First case, the prediction is correct: $y^b = 1$ and $z_m = z_y$. Let us inject these in (8): $\frac{\partial l_{BCE}}{\partial T} = \frac{1}{T^2} \cdot (z_y - \sum_k z_k \cdot f_k) = \frac{\partial l_{CE}}{\partial T}$. We thus get the same gradient as the standard cross-entropy loss.

- Second case, the prediction is incorrect: $y^b = 0$ and $z_m > z_y$. (8) becomes: $\frac{\partial l_{BCE}}{\partial T} = \frac{1}{T^2} \cdot \frac{s}{s-1} \cdot (z_m - \sum_k z_k \cdot f_k)$. By comparing to (7), we have the term $\frac{1}{T^2} \cdot (z_m - \sum_k z_k \cdot f_k) > \frac{1}{T^2}(z_y - \sum_k z_k \cdot f_k) = \frac{\partial l_{CE}}{\partial T}$ and the remaining part of (8) $|\frac{s}{s-1}| > 1$ when $s > 0.5$.

So to recapitulate, $|\frac{\partial l_{BCE}}{\partial T}| > |\frac{\partial l_{CE}}{\partial T}|$ when $s > 0.5$, which corresponds to the vast majority of data points as the classifier gets better calibrated. This is shown in Figure 1.

We also have $\lim_{s\to 1}|\frac{s}{s-1}| = \infty$. In practice, $s$ is not close enough to 1 to generate exploding gradients, so it just means that as confidences for wrong predictions gets higher, so does the gradient to reduce the confidences.

The conclusion is that for correct predictions, our approach does not change the optimization, but for incorrect predictions, the gradient is stronger and penalizes more heavily confident predictions that are wrong. This is also proven experimentally by looking at Table 8 where we see that average

confidences of temperature scaling with the TvA approach ($\text{TS}_{\text{TvA}}$) are lower than the ones using the standard approach (TS), for almost all networks. This makes the average confidences closer to the accuracy, showing reduced overconfidence.

# E    Limits of classwise-ECE and top-label-ECE for a high number of classes

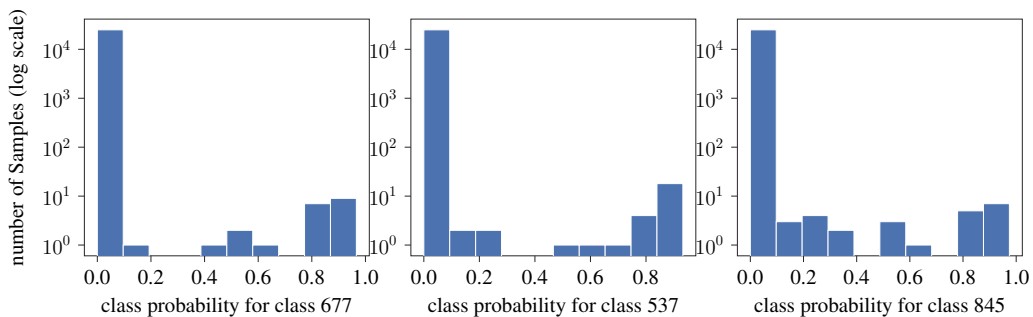

Figure 4: Histogram of class probabilities for 3 random classes, for ViT-16/B on ImageNet.

Let us define the ECE for class $j$:

$$\text{ECE}_j = \sum_{b=1}^{B} \frac{n_b}{N} |\text{acc}(b, j) - \text{conf}(b, j)|$$

The difference compared to (2) is that now $\text{acc}(b, j)$ corresponds to the proportion of class $j$ in the bin. Also, $\text{conf}(b, j)$ now is the average probability given to class $j$ for all samples in the bin. Then, classwise-ECE [29] takes the average for all classes:

$$\text{ECE}_{\text{cw}} = \sum_{j=1}^{L} \text{ECE}_j$$

Classwise-ECE considers the full probabilities vectors: all the class probabilities for each prediction. However, this metric does not scale to large numbers of classes. Let us see why with an example.

Let us use a test set of $N$ samples, $N/L$ for each of the $L$ classes (the dataset is balanced), and a high-accuracy classifier fairly calibrated. The classifier predicts $N$ probability vectors of length $L$. Predicted probabilities for class $j$ are all the values of the vector at dimension $j$. Because the classifier has a high accuracy and is fairly calibrated, around $N/L$ values are close to 1 (corresponding to mostly correct predictions), and the remaining ones, around $N - N/L$, are close to 0 (because the predicted class is not class $j$, and the predicted probability is high for another class).

To compute $\text{ECE}_j$ with equal size 15 bins, the predicted probabilities for class $j$ are partitioned into 15 bins. The first bin (with probabilities close to 0), contains $n_1 \approx N - N/L$ samples while the last one (with probabilities close to 1) contains $n_B \approx N/L$ samples. The remaining bins are usually even more empty. That means that the calibration error in the first bin is weighted $n_1/n_B = L - 1$ times more than the last one. For the 1000 classes of ImageNet, $L - 1 = 999$. Figure 4 shows the number of samples ($n_b$) in each bin for an ImageNet classifier.

Because the impact of the calibration error in the bin with the high probabilities is negligible relative to the bin with the low probabilities, the classwise-ECE mostly measures whether probabilities close to 0 are calibrated. We argue this is not what we are interested in: what matters more is the calibration of higher values of the probabilities.

Top-label ECE [18] is another interesting metric that does not scale to large numbers of classes either. Top-label-ECE divides data into subsets according to the predicted class, computes the ECEs of these subsets, and averages them. For an ImageNet test set of 25000 samples (25 per class), data is divided into 1000 subsets of $\approx 25$ samples each (the classifier is high-accuracy, most of the time the predicted class is equal to the true class). The ECE is computed for each subset containing only 25 samples. To compute the ECE, samples are typically partitioned into 15 bins. The number of samples per bin does not allow a correct estimation of the average confidence or accuracy.

# F  Implementation details

## F.1  Models weights

- Model weights for CIFAR are from [45].
- Model weights for ImageNet come from torchvision [42].
- Model weights for ImageNet-21K are from [54].
- CLIP weights are from OpenAI's Hugging Face.
- Original weights for T5 and RoBERTa come from the Transformers library [67]. The models are fine-tuned for each task using prompt-based learning [36]. For more details, see [5].
- We used the model GPT-J `https://huggingface.co/EleutherAI/gpt-j-6b` (Apache-2.0 license) and Llama-2 `https://huggingface.co/meta-llama/Llama-2-13b` (license).

## F.2  Datasets

- *CIFAR-10 (C10)* and *CIFAR-100 (C100)* [27] contain 60000 32x32 images corresponding to 10 and 100 classes, respectively. Data is split into subsets of 45000/5000/10000 images for train/validation/test. We concatenate the original validation and test sets, and randomly split that into a calibration set of size 5000, and a test set of size 10000.
- *ImageNet (IN)* [7] contains 1.3 million images from 1000 classes. Following [17], we randomly split the original validation test of size 50000 into a calibration set and a test set, both of size 25000.
- *ImageNet-21K (IN21K)* [54], in its winter21 version, contains 11 million images in the train set, and 522500 in the test set (50 for each of the 10450 classes). We randomly split the test set into equal-sized calibration and test set (261250 samples each, 25 per class).
- *Amazon Fine Foods* [43] is a collection of customer reviews for fine foods sold on Amazon. Reviews are categorized into bad, neutral, and good. The original validation set size is 78741 and test size 91606. We randomly split them into 78741 samples for calibration and 91606 for test.
- *DynaSent* [51] is a dynamic benchmark for sentiment analysis consisting of sentences annotated as positive, neutral, and negative. The original validation set size is 11160 and test size 4320. We randomly split them into 11160 samples for calibration and 4320 for test.
- *MNLI* [66] contains pairs of sentences labeled as contradiction, neutral, and entailment. The original validation set size is 19635 and test size 9815. We randomly split them into 19635 samples for calibration and 9815 for test.
- *Yahoo Answers (YA)* [73] contains question-answers pairs corresponding to 10 different topics. The original validation set size is 14000 and test size 60000. We randomly split them into 14000 samples for calibration and 60000 for test.
- TREC [61] contains questions categorized into 6 classes. The training set contains 5500 labeled questions, and the test set contains another 500.
- SST-5 [55] contains 11855 sentences corresponding to 5 sentiments (from very negative to very positive).
- DBpedia [74] contains text for topic classification with 14 classes. The training set contains 560000 samples, and the test set 5000.

## F.3  Code

- We used the library netcal [32] (Apache-2.0 license) for their implementation of binary methods for calibration and adapted their reliability diagrams code. For HB, we tested equal-size and equal-mass bins, and chose the best variant for each case. All hyperparameters were kept at their default values (10 bins for HB).
- We took inspiration from the official implementation of temperature scaling: `https://github.com/gpleiss/temperature_scaling` (MIT license).

- We took inspiration from the official implementation of Dirichlet calibration: `https://github.com/dirichletcal/experiments_dnn` (MIT license).
- We used the official implementation of I-Max: `https://github.com/boschresearch/imax-calibration` (AGPL-3.0 license).
- We used the official implementation of IRM: `https://github.com/zhang64-llnl/Mix-n-Match-Calibration` (MIT license).
- For evaluation, we used codes from `https://github.com/JeremyNixon/uncertainty-metrics-1` (Apache-2.0 license) and `https://github.com/IdoGalil/benchmarking-uncertainty-estimation-performance` (MIT license).
- We used PyTorch 2.0.0 [3] (BSD-style license).
- We used CIFAR models from [45] `https://github.com/torrvision/focal_calibration` (MIT license).
- We used ImageNet-21K models [54] `https://github.com/Alibaba-MIIL/ImageNet21K` (MIT license).
- We used CLIP models from HuggingFace's Transformers library [67] (Apache-2.0 license).
- We used pretrained language models from Transformers [67] and calibration codes from [5] `https://github.com/lifan-yuan/PLMCalibration` (MIT license).
- We used code from `https://github.com/mominabbass/LinC` for the calibration of LLMs using ICL, itself built upon `https://github.com/tonyzhaozh/few-shot-learning` (Apache-2.0 license).

Table 3: Computing time (in seconds) of the calibration on ImageNet, using one NVIDIA V100 GPU. The first column denotes the data preprocessing time, which includes computing the model logits for all calibration examples. Post-hoc calibration methods do not usually require much computing power compared to classifier training.

| Model | Preproc. | IRM | I-Max | TS | $TS_{TvA}$ | VS | $VS_{reg\_TvA}$ | DC | $DC_{reg\_TvA}$ | HB | $HB_{TvA}$ | Iso | $Iso_{TvA}$ | Beta | $Beta_{TvA}$ | BBQ | $BBQ_{TvA}$ |
|---|---|---|---|---|---|---|---|---|---|---|---|---|---|---|---|---|---|
| ResNet-50 | 141 | 2021 | 543 | 215 | 218 | 214 | 217 | 226 | 226 | 129 | 1 | 66 | 1 | 873 | 22 | 1156 | 2 |
| ViT-B/16 | 151 | 7119 | 524 | 225 | 226 | 217 | 222 | 232 | 235 | 127 | 1 | 61 | 1 | 917 | 23 | 1169 | 2 |

# G   Impact on selective classification

Selective classification aims to improve a model's prediction performance by trading-off coverage: a reject option allows to discard data that might result in wrong predictions, thus improving the accuracy on the remaining data. A strong standard baseline uses thresholding on the maximum softmax probability outputted by the classifier [13]. Improving confidence calibration means uncertainty is better quantified and should result in better selective classification.

Results in Table 1 show the superiority of Histogram Binning (applied with the right framework) in reducing the calibration error ECE. Unfortunately, it does not translate into improvements in selective classification. AUROC is a standard metric for selective classification [11]. Table 4 shows that Histogram Binning actually degrades the AUROC, while the best method is Isotonic Regression. Our TvA framework does not significantly impact the AUROC.

This paper addresses confidence calibration, usually measured by ECE. AUROC is a global rank-based metric for selective classification: it relies on the relative values of the scores, not their absolute values. Even though calibration and selective classification are related, improvement in calibration does not directly translate to better selective classification. This has been clearly demonstrated experimentally by [11].
A good example of that difference is the behavior of $HB_{TvA}$: it is the best calibration method overall but actually degrades the AUROC in most cases. Such a difference can be explained by the fact that selective classification benefits from a continuous score able to discriminate between certain and uncertain examples finely, but HB quantizes the confidences into, e.g., 10 different values.

Table 4: AUROC in % (higher is better). Methods in purple impact the model prediction, potentially degrading accuracy; methods in teal do not. Improvements from the uncalibrated model are colored in blue and degradations in orange.

(a) CIFAR-10

| Model | Uncal. | IRM | I-Max | TS | $TS_{TvA}$ | VS | $VS_{reg\_TvA}$ | DC | $DC_{reg\_TvA}$ | Beta | $Beta_{TvA}$ | Iso | $Iso_{TvA}$ | BBQ | $BBQ_{TvA}$ | HB | $HB_{TvA}$ |
|---|---|---|---|---|---|---|---|---|---|---|---|---|---|---|---|---|---|
| ResNet-50 | 92.09 | 91.79 | 91.30 | 92.01 | 91.98 | 92.71 | 92.76 | 92.71 | 92.76 | 90.94 | 92.09 | 92.29 | 91.87 | 75.63 | 85.53 | 75.78 | 84.69 |
| ResNet-110 | 92.26 | 92.25 | 91.53 | 92.19 | 92.18 | 92.18 | 92.31 | 92.16 | 92.30 | 91.33 | 92.26 | 92.40 | 92.20 | 73.44 | 85.01 | 74.46 | 84.56 |
| WRN | 91.17 | 91.17 | 90.36 | 91.21 | 91.19 | 91.80 | 92.32 | 91.82 | 92.31 | 91.45 | 91.17 | 92.28 | 91.18 | 75.99 | 86.62 | 73.91 | 85.38 |
| DenseNet | 90.46 | 90.15 | 89.68 | 90.48 | 90.45 | 90.84 | 91.40 | 90.84 | 91.39 | 89.98 | 90.46 | 91.46 | 90.12 | 77.07 | 87.07 | 74.61 | 83.07 |
| CLIP (ViT-B/32) | 89.85 | 89.72 | 89.73 | 89.97 | 89.97 | 90.73 | 91.07 | 90.95 | 91.14 | 90.58 | 89.85 | 90.28 | 89.66 | 88.86 | 89.41 | 88.01 | 88.76 |
| CLIP (ViT-B/16) | 91.00 | 90.96 | 90.72 | 91.10 | 91.10 | 91.79 | 91.99 | 91.88 | 92.06 | 91.75 | 91.00 | 90.69 | 90.83 | 89.36 | 90.57 | 89.12 | 89.97 |
| CLIP (ViT-L/14) | 93.22 | 93.15 | 94.32 | 93.32 | 93.31 | 93.69 | 93.78 | 93.79 | 93.87 | 93.17 | 93.22 | 92.78 | 93.05 | 87.29 | 91.73 | 88.65 | 91.33 |

(b) CIFAR-100

| Model | Uncal. | IRM | I-Max | TS | $TS_{TvA}$ | VS | $VS_{reg\_TvA}$ | DC | $DC_{reg\_TvA}$ | Beta | $Beta_{TvA}$ | Iso | $Iso_{TvA}$ | BBQ | $BBQ_{TvA}$ | HB | $HB_{TvA}$ |
|---|---|---|---|---|---|---|---|---|---|---|---|---|---|---|---|---|---|
| ResNet-50 | 85.80 | 85.78 | 85.03 | 85.73 | 85.62 | 85.92 | 86.50 | 85.91 | 86.51 | 85.49 | 85.80 | 87.19 | 85.69 | 81.21 | 85.31 | 81.90 | 83.90 |
| ResNet-110 | 85.03 | 84.99 | 83.76 | 84.94 | 84.85 | 84.82 | 85.34 | 84.84 | 85.33 | 84.57 | 85.03 | 86.32 | 84.97 | 80.54 | 84.47 | 81.40 | 82.93 |
| WRN | 87.59 | 87.52 | 87.02 | 87.59 | 87.48 | 87.83 | 88.17 | 87.85 | 88.14 | 87.45 | 87.59 | 88.65 | 87.46 | 83.59 | 86.46 | 83.77 | 85.69 |
| DenseNet | 86.17 | 86.02 | 85.64 | 86.12 | 86.00 | 86.60 | 86.79 | 86.59 | 86.79 | 86.01 | 86.17 | 87.09 | 86.11 | 83.02 | 85.42 | 83.47 | 84.86 |
| CLIP (ViT-B/32) | 83.06 | 83.14 | 82.99 | 83.74 | 83.70 | 83.95 | 85.16 | 83.99 | 85.02 | 84.17 | 83.06 | 84.61 | 82.95 | 85.59 | 83.00 | 85.61 | 82.82 |
| CLIP (ViT-B/16) | 82.57 | 82.55 | 82.51 | 83.65 | 83.66 | 83.95 | 85.29 | 84.04 | 85.76 | 84.18 | 82.57 | 84.45 | 82.51 | 85.76 | 82.46 | 85.62 | 82.24 |
| CLIP (ViT-L/14) | 84.26 | 84.22 | 84.14 | 85.51 | 85.50 | 86.56 | 87.57 | 86.62 | 87.50 | 85.64 | 84.26 | 86.94 | 84.11 | 86.92 | 84.12 | 86.67 | 84.07 |

(c) ImageNet

| Model | Uncal. | IRM | I-Max | TS | $TS_{TvA}$ | VS | $VS_{reg\_TvA}$ | DC | $DC_{reg\_TvA}$ | Beta | $Beta_{TvA}$ | Iso | $Iso_{TvA}$ | BBQ | $BBQ_{TvA}$ | HB | $HB_{TvA}$ |
|---|---|---|---|---|---|---|---|---|---|---|---|---|---|---|---|---|---|
| ResNet-18 | 85.73 | 85.70 | 85.37 | 85.64 | 85.65 | 85.63 | 85.88 | 85.30 | 85.39 | 84.39 | 85.73 | 86.12 | 85.69 | 83.19 | 85.40 | 83.73 | 85.03 |
| ResNet-34 | 86.18 | 86.19 | 85.78 | 86.11 | 86.10 | 86.25 | 86.38 | 85.86 | 85.99 | 84.64 | 86.18 | 86.41 | 86.14 | 82.26 | 85.77 | 83.03 | 85.77 |
| ResNet-50 | 80.53 | 80.54 | 80.12 | 85.93 | 85.69 | 85.60 | 85.57 | 85.58 | 85.57 | 82.34 | 80.53 | 86.91 | 80.49 | 85.27 | 80.52 | 83.63 | 79.96 |
| ResNet-101 | 84.18 | 84.20 | 83.57 | 85.96 | 85.71 | 85.38 | 85.56 | 85.34 | 85.53 | 82.38 | 84.18 | 87.09 | 84.18 | 82.48 | 84.11 | 81.16 | 83.80 |
| EffNet-B7 | 84.92 | 84.84 | 84.10 | 86.61 | 86.34 | 85.18 | 85.51 | 85.18 | 85.52 | 81.91 | 84.92 | 87.14 | 84.87 | 81.57 | 84.94 | 80.38 | 84.26 |
| EffNetV2-S | 85.77 | 85.87 | 85.29 | 87.02 | 86.86 | 85.30 | 85.67 | 85.28 | 85.68 | 82.55 | 85.77 | 87.42 | 85.72 | 82.44 | 85.75 | 80.86 | 84.80 |
| EffNetV2-M | 82.36 | 82.36 | 81.58 | 85.26 | 84.92 | 83.66 | 84.19 | 83.64 | 84.23 | 80.51 | 82.36 | 86.51 | 82.32 | 81.24 | 82.23 | 79.45 | 81.71 |
| EffNetV2-L | 84.63 | 84.58 | 83.96 | 86.33 | 86.05 | 85.77 | 85.94 | 85.75 | 85.87 | 82.27 | 84.63 | 86.70 | 84.55 | 81.78 | 84.56 | 80.56 | 84.08 |
| ConvNeXt-T | 82.35 | 82.31 | 81.84 | 85.47 | 85.18 | 85.60 | 85.57 | 85.59 | 85.59 | 81.93 | 82.35 | 86.97 | 82.29 | 82.58 | 82.30 | 81.44 | 81.88 |
| ConvNeXt-S | 82.29 | 82.36 | 81.87 | 85.27 | 84.88 | 84.81 | 85.01 | 84.81 | 85.01 | 81.22 | 82.29 | 86.98 | 82.26 | 81.29 | 82.20 | 79.78 | 81.86 |
| ConvNeXt-B | 82.27 | 82.27 | 81.70 | 85.13 | 84.75 | 84.40 | 84.88 | 84.40 | 84.90 | 80.87 | 82.27 | 87.01 | 82.22 | 81.79 | 82.25 | 79.89 | 81.69 |
| ConvNeXt-L | 82.35 | 82.35 | 81.42 | 84.81 | 84.38 | 84.04 | 84.58 | 84.04 | 84.54 | 80.24 | 82.35 | 86.79 | 82.32 | 80.49 | 82.23 | 79.15 | 81.75 |
| ViT-B/32 | 85.57 | 85.60 | 85.10 | 86.31 | 86.13 | 85.95 | 85.98 | 85.93 | 85.98 | 83.56 | 85.57 | 87.16 | 85.54 | 83.39 | 85.55 | 82.69 | 85.11 |
| ViT-B/16 | 85.52 | 85.55 | 84.92 | 86.32 | 86.12 | 85.36 | 85.53 | 85.34 | 85.56 | 81.82 | 85.52 | 87.19 | 85.50 | 81.56 | 85.39 | 81.21 | 85.09 |
| ViT-L/32 | 85.42 | 85.45 | 84.78 | 85.93 | 85.73 | 85.19 | 85.29 | 85.20 | 85.30 | 82.07 | 85.42 | 87.25 | 85.41 | 81.51 | 85.40 | 81.42 | 85.09 |
| ViT-L/16 | 85.85 | 85.83 | 84.63 | 86.16 | 86.00 | 84.33 | 84.65 | 84.32 | 84.64 | 80.96 | 85.85 | 86.97 | 85.83 | 79.76 | 85.66 | 80.08 | 84.85 |
| ViT-H/14 | 87.28 | 87.24 | 86.60 | 87.53 | 87.34 | 86.74 | 86.71 | 86.77 | 86.78 | 82.15 | 87.28 | 86.65 | 87.22 | 79.33 | 86.54 | 80.08 | 85.32 |
| Swin-T | 85.68 | 85.71 | 85.04 | 86.50 | 86.34 | 85.79 | 85.86 | 85.80 | 85.86 | 83.14 | 85.68 | 87.10 | 85.66 | 82.15 | 85.64 | 81.39 | 85.16 |
| Swin-S | 85.37 | 85.36 | 84.75 | 85.99 | 85.78 | 84.99 | 85.20 | 85.00 | 85.20 | 80.92 | 85.37 | 86.92 | 85.35 | 80.25 | 85.32 | 80.05 | 84.80 |
| Swin-B | 84.11 | 84.26 | 83.38 | 85.26 | 84.91 | 83.93 | 84.18 | 83.92 | 84.20 | 79.78 | 84.11 | 86.55 | 84.19 | 78.94 | 84.24 | 79.09 | 83.09 |
| SwinV2-T | 85.80 | 85.79 | 85.10 | 86.74 | 86.56 | 85.82 | 86.04 | 85.80 | 86.02 | 83.06 | 85.80 | 87.29 | 85.78 | 82.58 | 85.76 | 81.47 | 85.26 |
| SwinV2-S | 85.75 | 85.75 | 85.03 | 86.61 | 86.39 | 85.19 | 85.51 | 85.19 | 85.50 | 81.77 | 85.75 | 87.18 | 85.74 | 80.54 | 85.75 | 80.41 | 84.73 |
| SwinV2-B | 85.15 | 85.13 | 84.19 | 86.07 | 85.82 | 84.43 | 84.70 | 84.40 | 84.65 | 80.31 | 85.15 | 86.99 | 85.16 | 79.57 | 84.98 | 79.42 | 83.86 |
| CLIP (ViT-B/32) | 80.56 | 80.56 | 80.24 | 80.59 | 80.57 | 81.73 | 83.21 | 77.72 | 77.92 | 81.14 | 80.56 | 81.43 | 80.52 | 81.85 | 80.53 | 82.14 | 80.18 |
| CLIP (ViT-B/16) | 81.12 | 81.08 | 80.96 | 81.16 | 81.15 | 82.28 | 83.64 | 78.10 | 84.57 | 81.52 | 81.12 | 82.17 | 81.10 | 82.33 | 81.07 | 82.42 | 80.76 |
| CLIP (ViT-L/14) | 82.98 | 82.95 | 82.50 | 82.90 | 82.86 | 83.66 | 85.07 | 80.09 | 79.86 | 82.75 | 82.98 | 83.29 | 82.93 | 82.85 | 82.80 | 82.87 | 82.36 |

(d) ImageNet-21K

| Model | Uncal. | IRM | I-Max | TS | $TS_{TvA}$ | VS | $VS_{reg\_TvA}$ | DC | $DC_{reg\_TvA}$ | Beta | $Beta_{TvA}$ | Iso | $Iso_{TvA}$ | BBQ | $BBQ_{TvA}$ | HB | $HB_{TvA}$ |
|---|---|---|---|---|---|---|---|---|---|---|---|---|---|---|---|---|---|
| MN3 | 68.79 | err. | err. | 67.80 | 65.89 | 80.00 | 81.00 | 61.24 | 51.94 | err. | 68.79 | 79.62 | 68.77 | err. | 68.77 | 90.86 | 68.40 |
| ViT-B/16 | 72.99 | err. | err. | 74.36 | 73.17 | 79.78 | 81.42 | 76.29 | 78.92 | err. | 72.99 | 79.66 | 73.10 | err. | 73.20 | 90.27 | 71.95 |

(e) Amazon Fine Foods

| Model | Uncal. | IRM | I-Max | TS | $TS_{TvA}$ | VS | $VS_{reg\_TvA}$ | DC | $DC_{reg\_TvA}$ | Beta | $Beta_{TvA}$ | Iso | $Iso_{TvA}$ | BBQ | $BBQ_{TvA}$ | HB | $HB_{TvA}$ |
|---|---|---|---|---|---|---|---|---|---|---|---|---|---|---|---|---|---|
| T5 | 85.04 | 84.99 | 84.32 | 85.11 | 85.11 | 87.84 | 87.99 | 88.09 | 88.02 | 87.46 | 85.04 | 87.03 | 85.04 | 85.53 | 83.47 | 80.53 | 79.41 |
| T5-large | 81.33 | 81.34 | 80.52 | 80.75 | 80.74 | 85.84 | 87.94 | 87.47 | 87.80 | 87.27 | 81.33 | 85.15 | 81.38 | 83.35 | 78.27 | 77.53 | 74.25 |
| RoBERTa | 83.52 | 83.50 | 80.50 | 83.34 | 83.34 | 84.96 | 86.75 | 86.44 | 86.56 | 86.27 | 83.51 | 83.93 | 83.51 | 81.62 | 75.17 | 75.80 | 73.35 |
| RoBERTa-large | 87.88 | 87.67 | 82.18 | 87.99 | 87.99 | 88.04 | 88.25 | 88.70 | 88.33 | 88.40 | 87.88 | 87.34 | 87.88 | 81.52 | 74.92 | 75.63 | 75.30 |

(f) DynaSent

| Model | Uncal. | IRM | I-Max | TS | $TS_{TvA}$ | VS | $VS_{reg\_TvA}$ | DC | $DC_{reg\_TvA}$ | Beta | $Beta_{TvA}$ | Iso | $Iso_{TvA}$ | BBQ | $BBQ_{TvA}$ | HB | $HB_{TvA}$ |
|---|---|---|---|---|---|---|---|---|---|---|---|---|---|---|---|---|---|
| T5 | 78.01 | 77.80 | 77.51 | 78.12 | 78.12 | 78.11 | 78.34 | 78.31 | 78.31 | 78.20 | 78.01 | 78.57 | 77.91 | 76.88 | 76.81 | 74.14 | 76.04 |
| T5-large | 77.61 | 77.69 | 76.63 | 77.81 | 77.81 | 77.82 | 79.18 | 78.87 | 79.35 | 78.83 | 77.61 | 79.20 | 77.61 | 74.72 | 74.34 | 59.63 | 73.32 |
| RoBERTa | 75.16 | 75.15 | 72.77 | 75.30 | 75.30 | 75.23 | 75.47 | 75.60 | 76.05 | 75.31 | 75.16 | 76.10 | 75.06 | 64.47 | 66.97 | 59.47 | 68.49 |
| RoBERTa-large | 76.18 | 75.83 | 72.80 | 76.34 | 76.34 | 76.25 | 76.52 | 76.05 | 76.12 | 75.67 | 76.18 | 76.13 | 76.06 | 61.33 | 66.46 | 59.53 | 68.27 |

(g) MNLI

| Model | Uncal. | IRM | I-Max | TS | $TS_{TvA}$ | VS | $VS_{reg\_TvA}$ | DC | $DC_{reg\_TvA}$ | Beta | $Beta_{TvA}$ | Iso | $Iso_{TvA}$ | BBQ | $BBQ_{TvA}$ | HB | $HB_{TvA}$ |
|---|---|---|---|---|---|---|---|---|---|---|---|---|---|---|---|---|---|
| T5 | 84.44 | 84.39 | 83.31 | 84.51 | 84.51 | 84.55 | 85.01 | 85.07 | 85.15 | 84.59 | 84.44 | 84.89 | 84.37 | 79.85 | 80.50 | 70.97 | 78.39 |
| T5-large | 82.01 | 81.83 | 79.68 | 82.10 | 82.10 | 82.08 | 81.99 | 82.26 | 82.30 | 82.14 | 82.01 | 82.11 | 81.92 | 66.48 | 69.68 | 59.63 | 73.32 |
| RoBERTa | 82.36 | 82.33 | 79.46 | 82.50 | 82.50 | 82.48 | 82.83 | 82.82 | 82.89 | 82.78 | 82.36 | 82.99 | 82.32 | 67.99 | 72.67 | 60.25 | 73.19 |
| RoBERTa-large | 83.58 | 83.49 | 79.43 | 83.63 | 83.63 | 83.58 | 83.78 | 83.63 | 83.75 | 83.49 | 83.58 | 83.58 | 83.50 | 68.68 | 72.21 | 61.39 | 74.25 |

(h) Yahoo Anwsers

| Model | Uncal. | IRM | I-Max | TS | $TS_{TvA}$ | VS | $VS_{reg\_TvA}$ | DC | $DC_{reg\_TvA}$ | Beta | $Beta_{TvA}$ | Iso | $Iso_{TvA}$ | BBQ | $BBQ_{TvA}$ | HB | $HB_{TvA}$ |
|---|---|---|---|---|---|---|---|---|---|---|---|---|---|---|---|---|---|
| T5 | 81.60 | 81.53 | 81.10 | 81.61 | 81.61 | 81.64 | 81.65 | 81.63 | 81.65 | 81.59 | 81.60 | 81.70 | 81.53 | 80.46 | 80.81 | 78.99 | 80.03 |
| T5-large | 81.09 | 81.00 | 80.45 | 81.08 | 81.07 | 81.16 | 81.17 | 81.22 | 81.29 | 81.07 | 81.09 | 81.15 | 81.05 | 79.24 | 79.89 | 77.41 | 79.06 |
| RoBERTa | 78.63 | 78.59 | 76.52 | 78.80 | 78.80 | 78.73 | 78.70 | 78.90 | 79.07 | 78.68 | 78.63 | 78.86 | 78.57 | 74.45 | 73.17 | 73.20 | 67.91 |
| RoBERTa-large | 79.13 | 79.08 | 76.92 | 79.41 | 79.41 | 79.30 | 79.46 | 79.37 | 79.81 | 79.34 | 79.13 | 79.35 | 79.04 | 75.00 | 72.69 | 73.67 | 72.96 |

# H Additional results

Table 5: ECE in % (lower is better, best in bold) – full results for image classification datasets. Averages on 5 seeds. Mean relative improvements from TvA are shown (negative values for reductions of ECE). Methods in purple impact the model prediction, potentially degrading accuracy; methods in teal do not. Values are averaged over five random seeds.

## (a) CIFAR-10

| Model | Uncal. | IRM | I-Max | TS | TS$_\text{TvA}$ | VS | VS$_\text{reg\_TvA}$ | DC | DC$_\text{reg\_TvA}$ | Beta | Beta$_\text{TvA}$ | Iso | Iso$_\text{TvA}$ | BBQ | BBQ$_\text{TvA}$ | HB | HB$_\text{TvA}$ |
|---|---|---|---|---|---|---|---|---|---|---|---|---|---|---|---|---|---|
| ResNet-50 | 1.80 | 0.77 | 0.68 | 1.07 | 1.09 | 0.91 | 0.90 | 0.91 | 0.89 | 2.16 | 1.50 | 1.13 | 0.74 | 1.27 | 0.94 | 1.02 | **0.53** |
| ResNet-110 | 2.57 | 0.53 | 0.54 | 1.32 | 1.36 | 1.35 | 1.33 | 1.35 | 1.34 | 2.97 | 1.40 | 1.20 | 0.56 | 1.45 | 0.67 | 1.42 | **0.37** |
| WRN | 1.21 | 0.78 | 0.64 | 1.10 | 0.92 | 1.19 | 0.97 | 1.19 | 0.97 | 1.75 | 1.46 | 1.16 | 0.82 | 0.88 | 0.78 | 0.79 | **0.52** |
| DenseNet | 1.52 | 0.60 | 0.59 | 1.32 | 1.61 | 1.21 | 1.47 | 1.19 | 1.47 | 2.04 | 2.05 | 0.98 | 0.58 | 1.07 | 0.71 | 1.00 | **0.30** |
| Mean improvement ConvNets | | | | 3% | | 0% | | 1% | | -25% | | -39% | | -31% | | -57% | |
| CLIP (ViT-B/32) | 4.77 | 1.39 | 1.35 | 1.02 | 1.02 | 2.79 | 1.88 | 2.82 | 1.90 | 1.64 | 1.46 | 1.21 | 1.16 | 1.79 | 1.05 | 2.32 | **0.98** |
| CLIP (ViT-B/16) | 5.39 | 1.04 | 0.95 | 0.64 | **0.57** | 2.91 | 1.91 | 2.92 | 1.89 | 1.15 | 1.83 | 1.29 | 0.79 | 1.58 | 0.91 | 2.14 | 0.75 |
| CLIP (ViT-L/14) | 4.93 | 0.65 | 0.52 | 0.68 | 0.60 | 1.98 | 1.76 | 1.94 | 1.74 | 0.96 | 0.77 | 0.65 | 0.62 | 0.81 | 0.62 | 1.01 | **0.46** |
| Mean improvement CLIP | | | | -8% | | -26% | | -26% | | 9% | | -16% | | -36% | | -59% | |

## (b) CIFAR-100

| Model | Uncal. | IRM | I-Max | TS | TS$_\text{TvA}$ | VS | VS$_\text{reg\_TvA}$ | DC | DC$_\text{reg\_TvA}$ | Beta | Beta$_\text{TvA}$ | Iso | Iso$_\text{TvA}$ | BBQ | BBQ$_\text{TvA}$ | HB | HB$_\text{TvA}$ |
|---|---|---|---|---|---|---|---|---|---|---|---|---|---|---|---|---|---|
| ResNet-50 | 6.56 | 1.37 | 1.35 | 4.93 | 2.97 | 5.23 | 2.24 | 5.22 | 2.24 | 5.59 | 3.39 | 5.70 | 1.40 | 10.07 | 1.47 | 9.62 | **1.17** |
| ResNet-110 | 7.95 | 1.40 | 1.31 | 5.05 | 4.04 | 5.32 | 2.65 | 5.30 | 2.70 | 6.10 | 4.74 | 8.53 | 1.44 | 10.04 | 1.44 | 10.04 | **1.23** |
| WRN | 4.41 | 1.24 | 0.95 | 4.42 | 2.70 | 4.57 | 2.28 | 4.55 | 2.27 | 4.50 | 2.75 | 4.42 | 1.41 | 10.02 | 1.26 | 8.45 | **0.94** |
| DenseNet | 5.23 | 1.20 | 0.97 | 4.19 | 2.12 | 4.53 | 2.23 | 4.51 | 2.20 | 4.99 | 2.84 | 4.61 | 1.35 | 9.91 | 1.24 | 10.12 | **0.76** |
| Mean improvement ConvNets | | | | -37% | | -52% | | -52% | | -36% | | -73% | | -86% | | -89% | |
| CLIP (ViT-B/32) | 9.51 | 2.22 | 1.80 | 2.22 | 2.12 | 8.74 | 3.49 | 7.97 | 1.98 | 6.52 | 2.71 | 2.35 | 1.47 | 8.11 | **1.21** | 8.13 | 1.23 |
| CLIP (ViT-B/16) | 10.63 | 3.33 | 3.04 | 2.71 | 2.74 | 8.64 | 3.04 | 8.14 | 1.80 | 7.09 | 2.53 | 2.78 | 1.74 | 7.41 | 1.76 | 7.09 | **1.48** |
| CLIP (ViT-L/14) | 10.96 | 3.15 | 2.86 | 2.68 | 2.66 | 5.96 | 2.06 | 6.54 | 1.74 | 6.44 | 1.99 | 2.46 | 1.62 | 6.93 | 1.47 | 6.46 | **1.46** |
| Mean improvement CLIP | | | | -1% | | -63% | | -75% | | -64% | | -36% | | -80% | | -80% | |

## (c) ImageNet

| Model | Uncal. | IRM | I-Max | TS | TS$_\text{TvA}$ | VS | VS$_\text{reg\_TvA}$ | DC | DC$_\text{reg\_TvA}$ | Beta | Beta$_\text{TvA}$ | Iso | Iso$_\text{TvA}$ | BBQ | BBQ$_\text{TvA}$ | HB | HB$_\text{TvA}$ |
|---|---|---|---|---|---|---|---|---|---|---|---|---|---|---|---|---|---|
| ResNet-18 | 2.71 | 1.01 | **0.57** | 1.87 | 1.86 | 1.73 | 2.12 | 3.43 | 3.44 | 4.39 | 1.44 | 3.87 | 0.94 | 9.68 | 0.91 | 9.54 | **0.57** |
| ResNet-34 | 3.63 | 0.84 | **0.56** | 1.77 | 1.80 | 1.87 | 2.02 | 3.46 | 2.98 | 4.97 | 1.11 | 4.08 | 0.83 | 9.17 | 0.85 | 8.42 | 0.60 |
| ResNet-50 | 41.15 | 2.56 | 2.59 | 3.25 | 1.66 | 3.26 | 0.90 | 3.27 | 0.94 | 11.30 | 2.20 | 1.23 | 0.68 | 8.44 | 0.66 | 5.80 | **0.50** |
| ResNet-101 | 13.56 | 0.82 | 0.58 | 3.72 | 2.22 | 4.22 | 1.62 | 4.20 | 1.58 | 9.21 | 1.87 | 3.01 | 0.71 | 6.35 | 0.61 | 6.18 | **0.52** |
| Mean improvement ResNet | | | | -22% | | -26% | | -37% | | -76% | | -69% | | -91% | | -92% | |
| EffNet-B7 | 12.61 | 0.61 | **0.40** | 3.71 | 2.96 | 3.84 | 1.41 | 3.82 | 1.35 | 9.32 | 2.27 | 2.93 | 0.65 | 6.93 | 0.58 | 4.89 | **0.40** |
| EffNetV2-S | 16.92 | 0.68 | **0.44** | 3.60 | 3.34 | 3.91 | 1.43 | 3.90 | 1.45 | 8.03 | 2.57 | 2.97 | 0.67 | 7.66 | 0.68 | 5.33 | 0.47 |
| EffNetV2-M | 24.88 | 0.80 | 0.70 | 3.77 | 2.71 | 3.84 | 1.16 | 3.82 | 1.14 | 8.32 | 1.79 | 2.89 | 0.75 | 6.55 | 0.75 | 4.36 | **0.49** |
| EffNetV2-L | 8.48 | 0.63 | 0.39 | 2.86 | 1.34 | 3.08 | 1.05 | 3.06 | 0.98 | 9.45 | 0.99 | 2.51 | 0.64 | 5.06 | 0.54 | 2.99 | **0.37** |
| Mean improvement EffNet | | | | -27% | | -66% | | -66% | | -78% | | -76% | | -90% | | -90% | |
| ConvNeXt-T | 16.95 | 1.11 | 0.84 | 3.08 | 1.52 | 3.49 | 1.18 | 3.48 | 1.15 | 8.95 | 1.66 | 2.55 | 0.87 | 7.34 | 0.70 | 5.63 | **0.61** |
| ConvNeXt-S | 17.60 | 0.75 | 0.59 | 3.76 | 2.29 | 4.19 | 1.32 | 4.18 | 1.31 | 8.77 | 1.73 | 3.06 | 0.70 | 7.46 | 0.68 | 5.32 | **0.48** |
| ConvNeXt-B | 18.78 | 0.74 | **0.41** | 3.83 | 2.51 | 4.10 | 1.33 | 4.09 | 1.31 | 9.44 | 1.84 | 3.03 | 0.77 | 7.72 | 0.65 | 5.02 | 0.52 |
| ConvNeXt-L | 12.52 | 0.66 | 0.47 | 4.02 | 2.69 | 4.42 | 1.64 | 4.42 | 1.63 | 7.97 | 1.37 | 3.26 | 0.67 | 7.12 | 0.62 | 4.55 | **0.46** |
| Mean improvement ConvNeXt | | | | -39% | | -66% | | -67% | | -81% | | -74% | | -91% | | -90% | |
| ViT-B/32 | 6.37 | 0.77 | 0.60 | 4.02 | 2.17 | 4.67 | 1.82 | 4.66 | 1.77 | 6.58 | 1.68 | 3.58 | 0.84 | 9.51 | 0.73 | 7.76 | **0.53** |
| ViT-B/16 | 5.61 | 0.86 | 0.54 | 3.80 | 3.25 | 4.29 | 1.93 | 4.27 | 1.92 | 7.36 | 2.29 | 3.39 | 0.79 | 5.88 | 0.71 | 6.79 | **0.51** |
| ViT-L/32 | 4.27 | 0.83 | 0.75 | 5.00 | 3.89 | 5.37 | 2.53 | 5.37 | 2.49 | 6.33 | 2.57 | 4.43 | 0.76 | 9.31 | 0.79 | 7.39 | **0.61** |
| ViT-L/16 | 5.17 | 0.99 | 0.77 | 5.77 | 4.63 | 5.29 | 2.62 | 5.27 | 2.58 | 7.44 | 3.05 | 4.10 | 0.85 | 6.83 | 0.78 | 7.38 | **0.54** |
| ViT-H/14 | 0.60 | 0.60 | **0.40** | 1.84 | 0.88 | 1.95 | 1.22 | 2.00 | 1.17 | 7.84 | 0.75 | 2.48 | 0.62 | 1.67 | 0.63 | 3.62 | 0.42 |
| Mean improvement ViT | | | | -31% | | -51% | | -53% | | -70% | | -78% | | -85% | | -92% | |
| Swin-T | 6.82 | 0.76 | 0.45 | 3.08 | 1.85 | 3.45 | 1.38 | 3.44 | 1.37 | 7.72 | 1.61 | 2.94 | 0.72 | 6.72 | 0.67 | 6.31 | **0.43** |
| Swin-S | 3.65 | 0.78 | 0.54 | 3.63 | 2.95 | 4.17 | 1.77 | 4.17 | 1.76 | 7.91 | 2.31 | 3.29 | 0.77 | 7.20 | 0.68 | 5.72 | **0.44** |
| Swin-B | 4.77 | 0.72 | **0.45** | 3.88 | 3.43 | 4.22 | 1.98 | 4.21 | 1.95 | 7.98 | 2.27 | 3.33 | 0.75 | 6.83 | 0.68 | 4.70 | 0.52 |
| SwinV2-T | 8.31 | 0.80 | **0.46** | 3.61 | 2.25 | 3.92 | 1.51 | 3.91 | 1.49 | 8.68 | 1.76 | 3.08 | 0.81 | 7.81 | 0.79 | 6.20 | 0.52 |
| SwinV2-S | 6.07 | 0.75 | 0.46 | 3.79 | 3.32 | 4.24 | 1.74 | 4.23 | 1.71 | 8.51 | 2.26 | 3.16 | 0.74 | 7.18 | 0.67 | 5.02 | **0.41** |
| SwinV2-B | 5.50 | 0.69 | 0.59 | 3.82 | 3.68 | 4.25 | 1.80 | 4.22 | 1.73 | 7.53 | 2.68 | 3.34 | 0.67 | 6.78 | 0.63 | 4.42 | **0.55** |
| Mean improvement Swin | | | | -21% | | -58% | | -59% | | -73% | | -77% | | -90% | | -91% | |
| CLIP (ViT-B/32) | 1.50 | 0.96 | **0.75** | 1.70 | 1.58 | 1.38 | 0.92 | 36.01 | 70.52 | 3.57 | 0.82 | 2.22 | 0.84 | 8.12 | 0.88 | 6.63 | 0.82 |
| CLIP (ViT-B/16) | 1.80 | 1.31 | 0.75 | 1.92 | 1.89 | 1.61 | 0.87 | 34.02 | 66.04 | 4.53 | 1.08 | 2.35 | 1.01 | 8.48 | 0.91 | 6.98 | **0.74** |
| CLIP (ViT-L/14) | 2.57 | 0.97 | **0.67** | 2.04 | 1.99 | 1.89 | 1.36 | 26.06 | 66.40 | 5.95 | 1.35 | 2.49 | 0.92 | 8.33 | 1.01 | 7.86 | 0.84 |
| Mean improvement CLIP | | | | -4% | | -36% | | 115% | | -77% | | -61% | | -89% | | -89% | |

## (d) ImageNet-21K

| Model | Uncal. | IRM | I-Max | TS | TS$_\text{TvA}$ | VS | VS$_\text{reg\_TvA}$ | DC | DC$_\text{reg\_TvA}$ | Beta | Beta$_\text{TvA}$ | Iso | Iso$_\text{TvA}$ | BBQ | BBQ$_\text{TvA}$ | HB | HB$_\text{TvA}$ |
|---|---|---|---|---|---|---|---|---|---|---|---|---|---|---|---|---|---|
| MN3 | 12.34 | err. | err. | 8.69 | 4.39 | 2.52 | 2.40 | 58.84 | 81.16 | err. | 1.02 | 2.00 | 0.21 | err. | 0.20 | 5.50 | **0.17** |
| ViT-B/16 | 6.27 | err. | err. | 8.92 | 6.55 | 2.38 | 1.54 | 8.22 | 3.20 | err. | 3.72 | 2.14 | 0.22 | err. | 0.24 | 7.89 | **0.12** |
| Mean improvement | | | | -38% | | -20% | | -12% | | err. | | -90% | | err. | | -98% | |

Table 6: ECE in % (lower is better, best in bold) – full results for text classification datasets. Averages on 5 seeds. Mean relative improvements from TvA are shown (negative values for reductions of ECE). Methods in purple impact the model prediction, potentially degrading accuracy; methods in teal do not. Values are averaged over five random seeds.

(a) Amazon Fine Foods

| Model | Uncal. | IRM | I-Max | TS | TS$_{\text{TvA}}$ | VS | VS$_{\text{reg\_TvA}}$ | DC | DC$_{\text{reg\_TvA}}$ | Beta | Beta$_{\text{TvA}}$ | Iso | Iso$_{\text{TvA}}$ | BBQ | BBQ$_{\text{TvA}}$ | HB | HB$_{\text{TvA}}$ |
|---|---|---|---|---|---|---|---|---|---|---|---|---|---|---|---|---|---|
| T5 | 5.18 | 0.28 | 0.25 | 1.24 | 1.26 | 1.34 | 1.34 | 0.99 | 1.28 | 5.44 | 0.80 | 0.41 | 0.28 | 0.38 | 0.30 | 2.45 | **0.21** |
| T5-large | 5.76 | 0.26 | 0.26 | 0.97 | 1.04 | 1.70 | 1.49 | 1.36 | 1.34 | 5.71 | 1.63 | 0.33 | 0.26 | 0.40 | 0.26 | 3.29 | **0.14** |
| Mean improvement T5 | | | | 4% | | -6% | | 14% | | -78% | | -26% | | -28% | | -94% | |
| RoBERTa | 7.90 | 0.28 | 0.30 | 2.27 | 2.21 | 1.37 | 1.93 | 1.51 | 1.78 | 7.48 | 4.30 | 0.31 | 0.28 | 1.11 | 0.37 | 4.07 | **0.24** |
| RoBERTa-large | 6.83 | 0.32 | 0.25 | 2.52 | 2.44 | 1.45 | 1.78 | 1.24 | 1.57 | 6.36 | 4.45 | 0.72 | 0.26 | 0.38 | 0.34 | 3.96 | **0.16** |
| Mean improvement RoBERTa | | | | -3% | | 32% | | 22% | | -36% | | -37% | | -39% | | -95% | |

(b) DynaSent

| Model | Uncal. | IRM | I-Max | TS | TS$_{\text{TvA}}$ | VS | VS$_{\text{reg\_TvA}}$ | DC | DC$_{\text{reg\_TvA}}$ | Beta | Beta$_{\text{TvA}}$ | Iso | Iso$_{\text{TvA}}$ | BBQ | BBQ$_{\text{TvA}}$ | HB | HB$_{\text{TvA}}$ |
|---|---|---|---|---|---|---|---|---|---|---|---|---|---|---|---|---|---|
| T5 | 7.99 | 1.48 | 1.40 | 1.18 | **1.16** | 4.88 | 1.97 | 4.66 | 2.04 | 10.95 | 2.81 | 1.44 | 1.66 | 1.32 | 1.53 | 1.99 | 1.32 |
| T5-large | 9.73 | 1.30 | 1.36 | 3.20 | 3.19 | 7.38 | 2.03 | 7.15 | 2.00 | 11.81 | 4.72 | 1.56 | 1.45 | 1.44 | 1.62 | 1.80 | **0.92** |
| Mean improvement T5 | | | | -1% | | -66% | | -64% | | -67% | | 4% | | 14% | | -41% | |
| RoBERTa | 17.37 | 1.67 | 1.59 | 13.20 | 13.20 | 15.84 | 7.83 | 15.02 | 6.40 | 18.36 | 10.48 | 1.68 | 1.64 | 1.76 | 1.19 | 1.26 | **1.06** |
| RoBERTa-large | 14.88 | 1.46 | 1.42 | 10.94 | 10.94 | 13.49 | 5.77 | 12.78 | 4.73 | 15.69 | 9.30 | 1.74 | 1.43 | 1.52 | 1.08 | 0.85 | **0.75** |
| Mean improvement RoBERTa | | | | 0% | | -54% | | -60% | | -42% | | -10% | | -31% | | -14% | |

(c) MNLI

| Model | Uncal. | IRM | I-Max | TS | TS$_{\text{TvA}}$ | VS | VS$_{\text{reg\_TvA}}$ | DC | DC$_{\text{reg\_TvA}}$ | Beta | Beta$_{\text{TvA}}$ | Iso | Iso$_{\text{TvA}}$ | BBQ | BBQ$_{\text{TvA}}$ | HB | HB$_{\text{TvA}}$ |
|---|---|---|---|---|---|---|---|---|---|---|---|---|---|---|---|---|---|
| T5 | 6.48 | 0.71 | 0.67 | 1.17 | 1.15 | 3.25 | 1.88 | 3.21 | 2.01 | 7.74 | 2.12 | 0.84 | 0.73 | 0.98 | 0.80 | 1.91 | **0.47** |
| T5-large | 7.59 | 0.74 | 0.72 | 4.46 | 4.45 | 5.66 | 1.71 | 5.41 | 1.64 | 8.20 | 4.43 | 0.77 | 0.76 | 1.78 | 0.57 | 2.28 | **0.40** |
| Mean improvement T5 | | | | -1% | | -56% | | -54% | | -59% | | -7% | | -43% | | -79% | |
| RoBERTa | 10.26 | 0.90 | 0.81 | 6.52 | 6.52 | 7.83 | 2.03 | 7.38 | 2.12 | 11.06 | 6.16 | 0.87 | 0.94 | 1.25 | 0.93 | 1.20 | **0.60** |
| RoBERTa-large | 8.18 | 0.87 | 0.61 | 4.93 | 4.92 | 6.15 | 1.80 | 5.81 | 1.85 | 8.80 | 5.39 | 1.12 | 0.90 | 2.09 | 0.75 | 0.84 | **0.60** |
| Mean improvement RoBERTa | | | | -0% | | -72% | | -70% | | -42% | | -6% | | -45% | | -39% | |

(d) Yahoo Anwsers

| Model | Uncal. | IRM | I-Max | TS | TS$_{\text{TvA}}$ | VS | VS$_{\text{reg\_TvA}}$ | DC | DC$_{\text{reg\_TvA}}$ | Beta | Beta$_{\text{TvA}}$ | Iso | Iso$_{\text{TvA}}$ | BBQ | BBQ$_{\text{TvA}}$ | HB | HB$_{\text{TvA}}$ |
|---|---|---|---|---|---|---|---|---|---|---|---|---|---|---|---|---|---|
| T5 | 6.64 | 0.74 | 0.90 | **0.67** | 0.97 | 2.70 | 1.01 | 2.66 | 0.94 | 7.70 | 1.64 | 1.57 | 0.79 | 2.61 | 0.95 | 4.15 | 0.71 |
| T5-large | 9.04 | 0.87 | 0.72 | 1.47 | 1.73 | 4.70 | 1.31 | 4.84 | 1.36 | 10.34 | 2.39 | 1.90 | 0.85 | 3.01 | 0.98 | 3.15 | **0.67** |
| Mean improvement T5 | | | | 31% | | -67% | | -68% | | -78% | | -52% | | -66% | | -81% | |
| RoBERTa | 19.53 | 1.03 | 0.72 | 12.02 | 12.00 | 16.26 | 2.29 | 15.85 | 1.73 | 20.13 | 9.41 | 1.96 | 1.05 | 5.05 | 0.72 | 3.56 | **0.60** |
| RoBERTa-large | 19.65 | 0.90 | 0.86 | 12.77 | 12.75 | 16.67 | 2.75 | 16.30 | 2.70 | 20.18 | 10.19 | 1.87 | 0.94 | 4.96 | 0.78 | 3.26 | **0.57** |
| Mean improvement RoBERTa | | | | -0% | | -85% | | -86% | | -51% | | -48% | | -85% | | -83% | |

## Table 7: Standard deviations of ECE in % for 5 seeds.

### (a) CIFAR-10

| Model | Uncal. | IRM | I-Max | TS | TS$_{\text{TvA}}$ | VS | VS$_{\text{reg\_TvA}}$ | DC | DC$_{\text{reg\_TvA}}$ | Beta | Beta$_{\text{TvA}}$ | Iso | Iso$_{\text{TvA}}$ | BBQ | BBQ$_{\text{TvA}}$ | HB | HB$_{\text{TvA}}$ |
|---|---|---|---|---|---|---|---|---|---|---|---|---|---|---|---|---|---|
| ResNet-50 | 0.16 | 0.15 | 0.15 | 0.14 | 0.26 | 0.25 | 0.18 | 0.24 | 0.16 | 0.29 | 0.29 | 0.19 | 0.14 | 0.39 | 0.12 | 0.37 | 0.09 |
| ResNet-110 | 0.10 | 0.22 | 0.08 | 0.12 | 0.10 | 0.14 | 0.16 | 0.12 | 0.15 | 0.14 | 0.21 | 0.22 | 0.18 | 0.24 | 0.20 | 0.17 | 0.19 |
| WRN | 0.09 | 0.17 | 0.21 | 0.29 | 0.21 | 0.25 | 0.07 | 0.25 | 0.08 | 0.09 | 0.57 | 0.23 | 0.20 | 0.19 | 0.22 | 0.26 | 0.24 |
| DenseNet | 0.11 | 0.08 | 0.16 | 0.12 | 0.18 | 0.09 | 0.09 | 0.09 | 0.08 | 0.13 | 0.78 | 0.14 | 0.05 | 0.15 | 0.08 | 0.19 | 0.10 |
| CLIP (ViT-B/32) | 0.12 | 0.49 | 0.35 | 0.17 | 0.13 | 0.31 | 0.37 | 0.30 | 0.32 | 0.29 | 0.32 | 0.13 | 0.44 | 0.48 | 0.41 | 0.39 | 0.32 |
| CLIP (ViT-B/16) | 0.17 | 0.36 | 0.33 | 0.19 | 0.10 | 0.22 | 0.15 | 0.22 | 0.13 | 0.13 | 0.91 | 0.18 | 0.27 | 0.09 | 0.26 | 0.11 | 0.29 |
| CLIP (ViT-L/14) | 0.05 | 0.22 | 0.21 | 0.11 | 0.09 | 0.11 | 0.15 | 0.10 | 0.16 | 0.22 | 0.15 | 0.19 | 0.18 | 0.21 | 0.11 | 0.18 | 0.09 |

### (b) CIFAR-100

| Model | Uncal. | IRM | I-Max | TS | TS$_{\text{TvA}}$ | VS | VS$_{\text{reg\_TvA}}$ | DC | DC$_{\text{reg\_TvA}}$ | Beta | Beta$_{\text{TvA}}$ | Iso | Iso$_{\text{TvA}}$ | BBQ | BBQ$_{\text{TvA}}$ | HB | HB$_{\text{TvA}}$ |
|---|---|---|---|---|---|---|---|---|---|---|---|---|---|---|---|---|---|
| ResNet-50 | 0.22 | 0.50 | 0.60 | 0.53 | 0.47 | 0.52 | 0.46 | 0.50 | 0.44 | 0.47 | 0.48 | 0.32 | 0.50 | 0.37 | 0.33 | 0.49 | 0.25 |
| ResNet-110 | 0.28 | 0.23 | 0.25 | 0.38 | 0.39 | 0.35 | 0.31 | 0.33 | 0.31 | 0.33 | 0.59 | 0.28 | 0.34 | 0.60 | 0.21 | 0.56 | 0.11 |
| WRN | 0.19 | 0.24 | 0.38 | 0.27 | 0.14 | 0.25 | 0.30 | 0.24 | 0.33 | 0.16 | 0.28 | 0.23 | 0.46 | 0.71 | 0.57 | 0.33 | 0.32 |
| DenseNet | 0.10 | 0.24 | 0.17 | 0.26 | 0.13 | 0.30 | 0.20 | 0.35 | 0.22 | 0.60 | 0.79 | 0.24 | 0.45 | 0.68 | 0.28 | 0.31 | 0.22 |
| CLIP (ViT-B/32) | 0.14 | 0.25 | 0.20 | 0.05 | 0.14 | 0.13 | 0.27 | 0.48 | 0.45 | 0.32 | 1.17 | 0.26 | 0.25 | 0.49 | 0.23 | 0.40 | 0.46 |
| CLIP (ViT-B/16) | 0.21 | 0.35 | 0.42 | 0.30 | 0.37 | 0.40 | 0.37 | 0.42 | 0.48 | 0.74 | 1.00 | 0.50 | 0.42 | 0.54 | 0.42 | 0.37 | 0.51 |
| CLIP (ViT-L/14) | 0.17 | 0.33 | 0.53 | 0.16 | 0.19 | 0.23 | 0.24 | 0.19 | 0.32 | 0.34 | 1.14 | 0.25 | 0.24 | 0.39 | 0.30 | 0.27 | 0.09 |

### (c) ImageNet

| Model | Uncal. | IRM | I-Max | TS | TS$_{\text{TvA}}$ | VS | VS$_{\text{reg\_TvA}}$ | DC | DC$_{\text{reg\_TvA}}$ | Beta | Beta$_{\text{TvA}}$ | Iso | Iso$_{\text{TvA}}$ | BBQ | BBQ$_{\text{TvA}}$ | HB | HB$_{\text{TvA}}$ |
|---|---|---|---|---|---|---|---|---|---|---|---|---|---|---|---|---|---|
| ResNet-18 | 0.14 | 0.12 | 0.14 | 0.11 | 0.11 | 0.13 | 0.13 | 0.15 | 0.14 | 0.38 | 0.19 | 0.03 | 0.17 | 0.23 | 0.24 | 0.22 | 0.17 |
| ResNet-34 | 0.12 | 0.16 | 0.24 | 0.12 | 0.10 | 0.14 | 0.20 | 0.13 | 0.17 | 0.54 | 0.10 | 0.24 | 0.17 | 0.19 | 0.21 | 0.06 | 0.13 |
| ResNet-50 | 0.21 | 0.20 | 0.18 | 0.26 | 0.34 | 0.32 | 0.15 | 0.31 | 0.15 | 1.71 | 0.18 | 0.22 | 0.16 | 0.15 | 0.26 | 0.21 | 0.09 |
| ResNet-101 | 0.15 | 0.11 | 0.22 | 0.11 | 0.18 | 0.18 | 0.25 | 0.17 | 0.25 | 1.02 | 0.08 | 0.19 | 0.11 | 0.23 | 0.16 | 0.25 | 0.14 |
| EffNet-B7 | 0.07 | 0.13 | 0.11 | 0.10 | 0.12 | 0.15 | 0.18 | 0.14 | 0.22 | 1.91 | 0.17 | 0.13 | 0.10 | 0.16 | 0.16 | 0.11 | 0.06 |
| EffNetV2-S | 0.15 | 0.19 | 0.08 | 0.17 | 0.19 | 0.22 | 0.21 | 0.22 | 0.17 | 1.10 | 0.24 | 0.26 | 0.15 | 0.32 | 0.15 | 0.20 | 0.24 |
| EffNetV2-M | 0.18 | 0.12 | 0.13 | 0.17 | 0.15 | 0.16 | 0.26 | 0.18 | 0.22 | 1.10 | 0.47 | 0.13 | 0.10 | 0.24 | 0.15 | 0.26 | 0.06 |
| EffNetV2-L | 0.11 | 0.07 | 0.12 | 0.15 | 0.12 | 0.21 | 0.17 | 0.18 | 0.19 | 1.64 | 0.23 | 0.25 | 0.07 | 0.33 | 0.17 | 0.41 | 0.13 |
| ConvNeXt-T | 0.16 | 0.09 | 0.12 | 0.25 | 0.28 | 0.29 | 0.30 | 0.28 | 0.33 | 1.57 | 0.42 | 0.31 | 0.12 | 0.22 | 0.21 | 0.15 | 0.10 |
| ConvNeXt-S | 0.14 | 0.26 | 0.18 | 0.23 | 0.17 | 0.24 | 0.27 | 0.23 | 0.28 | 1.17 | 0.21 | 0.15 | 0.12 | 0.14 | 0.14 | 0.29 | 0.08 |
| ConvNeXt-B | 0.20 | 0.09 | 0.16 | 0.30 | 0.26 | 0.36 | 0.33 | 0.36 | 0.32 | 2.12 | 0.36 | 0.33 | 0.10 | 0.11 | 0.06 | 0.40 | 0.12 |
| ConvNeXt-L | 0.16 | 0.10 | 0.09 | 0.26 | 0.17 | 0.21 | 0.27 | 0.20 | 0.28 | 1.33 | 0.12 | 0.28 | 0.14 | 0.26 | 0.19 | 0.33 | 0.13 |
| ViT-B/32 | 0.20 | 0.17 | 0.19 | 0.29 | 0.17 | 0.34 | 0.34 | 0.37 | 0.30 | 0.99 | 0.12 | 0.31 | 0.15 | 0.21 | 0.16 | 0.28 | 0.14 |
| ViT-B/16 | 0.15 | 0.07 | 0.13 | 0.26 | 0.17 | 0.35 | 0.36 | 0.35 | 0.34 | 0.54 | 0.24 | 0.28 | 0.08 | 0.23 | 0.11 | 0.23 | 0.10 |
| ViT-L/32 | 0.10 | 0.15 | 0.11 | 0.19 | 0.10 | 0.28 | 0.22 | 0.28 | 0.22 | 0.31 | 0.17 | 0.29 | 0.11 | 0.20 | 0.16 | 0.23 | 0.14 |
| ViT-L/16 | 0.22 | 0.24 | 0.39 | 0.19 | 0.15 | 0.35 | 0.36 | 0.35 | 0.37 | 0.81 | 0.12 | 0.30 | 0.28 | 0.31 | 0.23 | 0.21 | 0.32 |
| ViT-H/14 | 0.15 | 0.18 | 0.09 | 0.21 | 0.28 | 0.19 | 0.20 | 0.20 | 0.22 | 0.75 | 0.15 | 0.20 | 0.19 | 0.34 | 0.16 | 0.29 | 0.18 |
| Swin-T | 0.17 | 0.13 | 0.19 | 0.19 | 0.12 | 0.20 | 0.16 | 0.20 | 0.16 | 0.70 | 0.10 | 0.23 | 0.15 | 0.34 | 0.20 | 0.12 | 0.17 |
| Swin-S | 0.10 | 0.18 | 0.18 | 0.19 | 0.19 | 0.21 | 0.25 | 0.22 | 0.20 | 0.48 | 0.27 | 0.24 | 0.16 | 0.22 | 0.27 | 0.30 | 0.14 |
| Swin-B | 0.07 | 0.14 | 0.16 | 0.21 | 0.25 | 0.24 | 0.30 | 0.21 | 0.32 | 0.80 | 0.11 | 0.28 | 0.21 | 0.24 | 0.15 | 0.34 | 0.13 |
| SwinV2-T | 0.10 | 0.09 | 0.14 | 0.15 | 0.10 | 0.22 | 0.20 | 0.20 | 0.20 | 1.63 | 0.07 | 0.29 | 0.13 | 0.32 | 0.22 | 0.11 | 0.16 |
| SwinV2-S | 0.14 | 0.06 | 0.17 | 0.18 | 0.16 | 0.26 | 0.21 | 0.25 | 0.27 | 1.29 | 0.08 | 0.31 | 0.09 | 0.29 | 0.14 | 0.19 | 0.21 |
| SwinV2-B | 0.10 | 0.10 | 0.18 | 0.14 | 0.13 | 0.24 | 0.17 | 0.24 | 0.20 | 0.57 | 0.09 | 0.16 | 0.13 | 0.11 | 0.11 | 0.21 | 0.04 |
| CLIP (ViT-B/32) | 0.20 | 0.33 | 0.34 | 0.25 | 0.20 | 0.18 | 0.17 | 0.73 | 1.83 | 0.51 | 0.18 | 0.36 | 0.35 | 0.23 | 0.24 | 0.17 | 0.37 |
| CLIP (ViT-B/16) | 0.11 | 0.27 | 0.12 | 0.12 | 0.10 | 0.17 | 0.10 | 1.08 | 3.69 | 0.71 | 0.62 | 0.18 | 0.17 | 0.22 | 0.13 | 0.10 | 0.15 |
| CLIP (ViT-L/14) | 0.16 | 0.10 | 0.11 | 0.07 | 0.16 | 0.10 | 0.05 | 0.96 | 18.76 | 1.01 | 0.45 | 0.15 | 0.15 | 0.23 | 0.15 | 0.29 | 0.21 |

### (d) ImageNet-21K

| Model | Uncal. | IRM | I-Max | TS | TS$_{\text{TvA}}$ | VS | VS$_{\text{reg\_TvA}}$ | DC | DC$_{\text{reg\_TvA}}$ | Beta | Beta$_{\text{TvA}}$ | Iso | Iso$_{\text{TvA}}$ | BBQ | BBQ$_{\text{TvA}}$ | HB | HB$_{\text{TvA}}$ |
|---|---|---|---|---|---|---|---|---|---|---|---|---|---|---|---|---|---|
| MN3 | 0.04 | err. | err. | 0.10 | 0.07 | 0.07 | 0.08 | 7.34 | 17.87 | err. | 0.23 | 0.08 | 0.06 | err. | 0.04 | 0.04 | 0.05 |
| ViT-B/16 | 0.08 | err. | err. | 0.09 | 0.16 | 0.08 | 0.04 | 0.23 | 0.18 | err. | 0.29 | 0.10 | 0.06 | err. | 0.06 | 0.05 | 0.05 |

### (e) Amazon Fine Foods

| Model | Uncal. | IRM | I-Max | TS | TS$_{\text{TvA}}$ | VS | VS$_{\text{reg\_TvA}}$ | DC | DC$_{\text{reg\_TvA}}$ | Beta | Beta$_{\text{TvA}}$ | Iso | Iso$_{\text{TvA}}$ | BBQ | BBQ$_{\text{TvA}}$ | HB | HB$_{\text{TvA}}$ |
|---|---|---|---|---|---|---|---|---|---|---|---|---|---|---|---|---|---|
| T5 | 0.09 | 0.06 | 0.10 | 0.05 | 0.05 | 0.07 | 0.15 | 0.08 | 0.14 | 0.08 | 0.09 | 0.11 | 0.06 | 0.06 | 0.04 | 0.14 | 0.06 |
| T5-large | 0.09 | 0.06 | 0.07 | 0.06 | 0.05 | 0.02 | 0.18 | 0.07 | 0.23 | 0.09 | 0.28 | 0.07 | 0.05 | 0.05 | 0.07 | 0.13 | 0.05 |
| RoBERTa | 0.12 | 0.08 | 0.06 | 0.14 | 0.14 | 0.07 | 0.05 | 0.06 | 0.12 | 0.15 | 0.04 | 0.07 | 0.07 | 0.06 | 0.09 | 0.12 | 0.08 |
| RoBERTa-large | 0.10 | 0.11 | 0.06 | 0.09 | 0.10 | 0.05 | 0.16 | 0.09 | 0.26 | 0.13 | 0.07 | 0.08 | 0.04 | 0.07 | 0.13 | 0.12 | 0.05 |

### (f) DynaSent

| Model | Uncal. | IRM | I-Max | TS | TS$_{\text{TvA}}$ | VS | VS$_{\text{reg\_TvA}}$ | DC | DC$_{\text{reg\_TvA}}$ | Beta | Beta$_{\text{TvA}}$ | Iso | Iso$_{\text{TvA}}$ | BBQ | BBQ$_{\text{TvA}}$ | HB | HB$_{\text{TvA}}$ |
|---|---|---|---|---|---|---|---|---|---|---|---|---|---|---|---|---|---|
| T5 | 0.51 | 0.21 | 0.39 | 0.35 | 0.28 | 0.60 | 0.54 | 0.52 | 0.47 | 0.69 | 0.55 | 0.22 | 0.20 | 0.42 | 0.33 | 0.49 | 0.54 |
| T5-large | 0.28 | 0.32 | 0.31 | 0.20 | 0.20 | 0.28 | 0.20 | 0.38 | 0.33 | 0.60 | 0.48 | 0.21 | 0.35 | 0.31 | 0.58 | 0.32 | 0.47 |
| RoBERTa | 0.67 | 0.41 | 0.58 | 0.65 | 0.65 | 0.63 | 0.49 | 0.60 | 0.63 | 0.54 | 0.28 | 0.57 | 0.45 | 0.33 | 0.30 | 0.16 | 0.17 |
| RoBERTa-large | 0.48 | 0.25 | 0.44 | 0.43 | 0.43 | 0.45 | 0.42 | 0.45 | 0.35 | 0.40 | 0.69 | 0.48 | 0.31 | 0.94 | 0.25 | 0.34 | 0.34 |

### (g) MNLI

| Model | Uncal. | IRM | I-Max | TS | TS$_{\text{TvA}}$ | VS | VS$_{\text{reg\_TvA}}$ | DC | DC$_{\text{reg\_TvA}}$ | Beta | Beta$_{\text{TvA}}$ | Iso | Iso$_{\text{TvA}}$ | BBQ | BBQ$_{\text{TvA}}$ | HB | HB$_{\text{TvA}}$ |
|---|---|---|---|---|---|---|---|---|---|---|---|---|---|---|---|---|---|
| T5 | 0.17 | 0.19 | 0.20 | 0.23 | 0.22 | 0.20 | 0.21 | 0.23 | 0.18 | 0.18 | 0.35 | 0.23 | 0.18 | 0.37 | 0.24 | 0.17 | 0.15 |
| T5-large | 0.27 | 0.16 | 0.13 | 0.35 | 0.34 | 0.30 | 0.40 | 0.34 | 0.38 | 0.21 | 0.21 | 0.14 | 0.12 | 0.23 | 0.23 | 0.23 | 0.26 |
| RoBERTa | 0.41 | 0.33 | 0.23 | 0.36 | 0.36 | 0.37 | 0.38 | 0.36 | 0.40 | 0.34 | 0.20 | 0.17 | 0.34 | 0.18 | 0.27 | 0.45 | 0.21 |
| RoBERTa-large | 0.13 | 0.13 | 0.18 | 0.12 | 0.12 | 0.13 | 0.16 | 0.13 | 0.17 | 0.19 | 0.21 | 0.28 | 0.16 | 0.19 | 0.26 | 0.27 | 0.16 |

### (h) Yahoo Answers

| Model | Uncal. | IRM | I-Max | TS | TS$_{\text{TvA}}$ | VS | VS$_{\text{reg\_TvA}}$ | DC | DC$_{\text{reg\_TvA}}$ | Beta | Beta$_{\text{TvA}}$ | Iso | Iso$_{\text{TvA}}$ | BBQ | BBQ$_{\text{TvA}}$ | HB | HB$_{\text{TvA}}$ |
|---|---|---|---|---|---|---|---|---|---|---|---|---|---|---|---|---|---|
| T5 | 0.04 | 0.20 | 0.36 | 0.07 | 0.24 | 0.18 | 0.25 | 0.17 | 0.29 | 0.22 | 0.74 | 0.18 | 0.23 | 0.22 | 0.31 | 0.76 | 0.23 |
| T5-large | 0.07 | 0.12 | 0.13 | 0.10 | 0.20 | 0.15 | 0.08 | 0.18 | 0.09 | 0.33 | 0.23 | 0.15 | 0.13 | 0.25 | 0.28 | 0.22 | 0.09 |
| RoBERTa | 0.09 | 0.19 | 0.26 | 0.09 | 0.09 | 0.09 | 0.25 | 0.11 | 0.29 | 0.11 | 0.44 | 0.21 | 0.11 | 0.30 | 0.26 | 0.36 | 0.30 |
| RoBERTa-large | 0.06 | 0.15 | 0.18 | 0.06 | 0.06 | 0.07 | 0.29 | 0.10 | 0.32 | 0.10 | 0.20 | 0.17 | 0.13 | 1.25 | 0.24 | 0.26 | 0.18 |

Table 8: Average confidence in %. Methods in purple impact the model prediction, potentially degrading accuracy; methods in teal do not. Overconfidence (average confidence > accuracy) is shown in violet and underconfidence (average confidence < accuracy) in brown.

### (a) CIFAR-10

| Model | Acc. | Uncal. | IRM | I-Max | TS | TS$_{TvA}$ | VS | VS$_{reg\_TvA}$ | DC | DC$_{reg\_TvA}$ | Beta | Beta$_{TvA}$ | Iso | Iso$_{TvA}$ | BBQ | BBQ$_{TvA}$ | HB | HB$_{TvA}$ |
|---|---|---|---|---|---|---|---|---|---|---|---|---|---|---|---|---|---|---|
| ResNet-50 | 94.9 | 96.6 | 94.9 | 94.8 | 95.6 | 95.2 | 95.6 | 95.5 | 95.6 | 95.5 | 96.8 | 95.1 | 95.3 | 94.9 | 94.9 | 94.9 | 94.9 | 95.0 |
| ResNet-110 | 94.6 | 97.1 | 94.9 | 94.8 | 95.7 | 95.3 | 95.7 | 95.7 | 95.7 | 95.7 | 97.5 | 95.0 | 95.4 | 94.8 | 94.8 | 94.8 | 94.8 | 94.8 |
| WRN | 95.8 | 95.9 | 96.1 | 96.0 | 96.8 | 96.5 | 96.8 | 96.3 | 96.8 | 96.3 | 97.4 | 96.0 | 96.5 | 96.0 | 96.2 | 96.0 | 96.2 | 96.0 |
| DenseNet | 95.0 | 95.7 | 95.0 | 94.9 | 95.9 | 95.6 | 96.0 | 95.6 | 95.9 | 95.6 | 96.9 | 95.2 | 95.5 | 95.0 | 95.1 | 95.0 | 95.1 | 94.9 |
| CLIP (ViT-B/32) | 88.2 | 83.4 | 87.5 | 87.3 | 87.7 | 87.8 | 87.5 | 88.5 | 87.6 | 88.5 | 89.8 | 88.4 | 89.7 | 87.9 | 89.2 | 87.9 | 88.8 | 87.9 |
| CLIP (ViT-B/16) | 90.2 | 84.8 | 89.8 | 89.5 | 89.8 | 90.1 | 89.4 | 90.4 | 89.5 | 90.5 | 91.7 | 90.9 | 91.9 | 90.2 | 91.3 | 90.2 | 90.8 | 90.2 |
| CLIP (ViT-L/14) | 95.3 | 90.4 | 95.0 | 88.7 | 94.9 | 95.0 | 94.5 | 94.8 | 94.6 | 94.8 | 95.7 | 96.0 | 96.2 | 95.2 | 95.9 | 95.2 | 95.7 | 95.2 |

### (b) CIFAR-100

| Model | Acc. | Uncal. | IRM | I-Max | TS | TS$_{TvA}$ | VS | VS$_{reg\_TvA}$ | DC | DC$_{reg\_TvA}$ | Beta | Beta$_{TvA}$ | Iso | Iso$_{TvA}$ | BBQ | BBQ$_{TvA}$ | HB | HB$_{TvA}$ |
|---|---|---|---|---|---|---|---|---|---|---|---|---|---|---|---|---|---|---|
| ResNet-50 | 76.7 | 82.9 | 77.2 | 77.0 | 80.9 | 76.9 | 81.4 | 77.5 | 81.4 | 77.4 | 80.3 | 78.0 | 81.5 | 76.9 | 74.0 | 76.9 | 75.6 | 76.8 |
| ResNet-110 | 75.0 | 82.8 | 75.5 | 75.3 | 79.6 | 75.5 | 80.0 | 76.2 | 80.0 | 76.2 | 79.3 | 76.9 | 80.6 | 75.2 | 71.4 | 75.2 | 73.0 | 75.3 |
| WRN | 79.6 | 83.0 | 79.5 | 79.3 | 83.0 | 79.3 | 83.4 | 79.7 | 83.3 | 79.7 | 81.1 | 79.6 | 83.2 | 79.3 | 75.5 | 79.3 | 77.3 | 79.3 |
| DenseNet | 76.3 | 81.0 | 76.2 | 76.0 | 79.6 | 76.0 | 80.1 | 76.6 | 80.1 | 76.6 | 78.7 | 75.6 | 79.9 | 75.8 | 72.6 | 75.8 | 74.0 | 76.0 |
| CLIP (ViT-B/32) | 62.3 | 52.8 | 60.7 | 60.1 | 63.3 | 62.3 | 59.0 | 63.9 | 58.3 | 63.5 | 60.7 | 62.9 | 65.2 | 62.3 | 57.0 | 62.3 | 57.8 | 62.3 |
| CLIP (ViT-B/16) | 66.7 | 56.0 | 64.5 | 63.9 | 66.9 | 67.1 | 62.7 | 68.4 | 62.0 | 67.9 | 64.0 | 66.5 | 68.5 | 66.9 | 61.0 | 66.9 | 62.1 | 67.0 |
| CLIP (ViT-L/14) | 76.0 | 65.0 | 73.9 | 73.4 | 76.1 | 76.0 | 74.1 | 78.6 | 73.4 | 78.4 | 73.5 | 74.6 | 78.5 | 76.0 | 72.8 | 76.0 | 73.4 | 75.9 |

### (c) ImageNet

| Model | Acc. | Uncal. | IRM | I-Max | TS | TS$_{TvA}$ | VS | VS$_{reg\_TvA}$ | DC | DC$_{reg\_TvA}$ | Beta | Beta$_{TvA}$ | Iso | Iso$_{TvA}$ | BBQ | BBQ$_{TvA}$ | HB | HB$_{TvA}$ |
|---|---|---|---|---|---|---|---|---|---|---|---|---|---|---|---|---|---|---|
| ResNet-18 | 69.8 | 72.0 | 69.6 | 69.2 | 69.4 | 69.7 | 70.3 | 70.8 | 71.4 | 70.7 | 65.5 | 69.4 | 73.1 | 69.8 | 65.3 | 69.8 | 67.8 | 69.8 |
| ResNet-34 | 73.2 | 76.8 | 73.4 | 72.9 | 73.5 | 73.1 | 74.4 | 74.3 | 75.2 | 74.2 | 68.3 | 73.2 | 76.8 | 73.3 | 70.3 | 73.3 | 72.4 | 73.3 |
| ResNet-50 | 80.8 | 39.7 | 78.5 | 77.9 | 84.0 | 82.2 | 84.2 | 80.2 | 84.2 | 80.2 | 69.3 | 81.2 | 84.5 | 82.0 | 71.1 | 80.9 | 74.7 | 80.9 |
| ResNet-101 | 81.9 | 68.3 | 81.7 | 81.4 | 85.5 | 83.4 | 86.0 | 83.1 | 86.0 | 83.1 | 72.5 | 82.2 | 84.5 | 82.0 | 78.2 | 82.0 | 80.6 | 82.1 |
| EffNet-B7 | 84.2 | 71.6 | 84.2 | 84.0 | 87.8 | 85.7 | 88.3 | 85.5 | 88.3 | 85.5 | 75.0 | 84.2 | 87.0 | 84.0 | 82.2 | 84.1 | 84.0 | 84.1 |
| EffNetV2-S | 84.3 | 67.3 | 84.2 | 84.0 | 87.9 | 85.6 | 88.2 | 85.3 | 88.2 | 85.2 | 76.3 | 84.7 | 86.8 | 84.2 | 81.2 | 84.2 | 83.5 | 84.2 |
| EffNetV2-M | 85.1 | 60.2 | 84.9 | 84.7 | 88.8 | 86.6 | 89.1 | 85.9 | 89.1 | 85.8 | 76.8 | 84.9 | 87.7 | 85.1 | 81.8 | 85.2 | 84.4 | 85.1 |
| EffNetV2-L | 85.8 | 77.3 | 85.7 | 85.4 | 88.6 | 86.9 | 88.9 | 86.7 | 88.9 | 86.6 | 76.4 | 85.8 | 88.1 | 85.6 | 84.2 | 85.6 | 85.9 | 85.7 |
| ConvNeXt-T | 82.5 | 65.6 | 82.0 | 81.7 | 85.6 | 84.0 | 85.9 | 83.3 | 85.9 | 83.2 | 73.5 | 81.9 | 84.7 | 82.5 | 78.9 | 82.5 | 81.2 | 82.5 |
| ConvNeXt-S | 83.6 | 66.0 | 83.5 | 83.2 | 87.4 | 85.3 | 87.8 | 84.7 | 87.8 | 84.7 | 74.9 | 83.1 | 86.3 | 83.0 | 80.7 | 83.6 | 82.9 | 83.6 |
| ConvNeXt-B | 84.0 | 65.3 | 83.9 | 83.7 | 87.8 | 85.6 | 88.2 | 85.0 | 88.2 | 85.0 | 74.6 | 84.0 | 86.7 | 84.0 | 81.2 | 84.1 | 83.4 | 84.1 |
| ConvNeXt-L | 84.4 | 71.9 | 84.5 | 84.3 | 88.4 | 86.2 | 88.8 | 85.8 | 88.8 | 85.8 | 76.6 | 84.5 | 87.4 | 84.5 | 82.6 | 84.5 | 84.3 | 84.5 |
| ViT-B/32 | 75.9 | 69.6 | 75.9 | 75.6 | 79.9 | 77.3 | 80.5 | 77.4 | 80.4 | 77.3 | 69.3 | 75.7 | 78.9 | 75.9 | 71.0 | 75.9 | 73.7 | 76.0 |
| ViT-B/16 | 81.0 | 75.5 | 81.2 | 81.0 | 84.8 | 82.6 | 85.3 | 82.8 | 85.3 | 82.7 | 73.8 | 81.2 | 84.0 | 81.0 | 78.8 | 81.0 | 80.5 | 81.1 |
| ViT-L/32 | 77.0 | 74.2 | 77.3 | 77.2 | 81.6 | 78.8 | 82.2 | 79.0 | 82.2 | 79.0 | 71.7 | 77.0 | 80.8 | 76.9 | 73.9 | 76.9 | 76.0 | 76.9 |
| ViT-L/16 | 79.6 | 78.8 | 80.1 | 80.0 | 84.5 | 81.7 | 85.1 | 82.1 | 85.1 | 82.1 | 72.9 | 80.1 | 83.6 | 79.7 | 78.3 | 79.7 | 79.8 | 79.7 |
| ViT-H/14 | 88.6 | 89.0 | 88.6 | 88.4 | 90.4 | 89.3 | 90.5 | 89.5 | 90.5 | 89.5 | 80.8 | 88.4 | 90.8 | 88.6 | 88.7 | 88.6 | 89.3 | 88.7 |
| Swin-T | 81.5 | 74.7 | 81.3 | 81.1 | 84.5 | 82.6 | 85.0 | 82.7 | 85.0 | 82.6 | 73.7 | 81.3 | 84.0 | 81.5 | 78.8 | 81.5 | 80.9 | 81.4 |
| Swin-S | 83.2 | 79.9 | 83.3 | 83.1 | 86.8 | 84.6 | 87.3 | 84.7 | 87.3 | 84.7 | 75.3 | 83.7 | 86.1 | 83.1 | 81.7 | 83.1 | 83.1 | 83.2 |
| Swin-B | 83.6 | 79.7 | 83.8 | 83.6 | 87.5 | 85.4 | 88.0 | 85.6 | 88.0 | 85.5 | 75.8 | 83.6 | 86.7 | 83.5 | 82.8 | 83.5 | 84.0 | 83.5 |
| SwinV2-T | 82.0 | 73.7 | 82.1 | 81.9 | 85.6 | 83.4 | 86.0 | 83.4 | 86.0 | 83.4 | 73.4 | 82.2 | 84.7 | 82.2 | 79.4 | 82.2 | 81.5 | 82.2 |
| SwinV2-S | 83.7 | 77.7 | 83.8 | 83.7 | 87.5 | 85.2 | 88.0 | 85.4 | 88.0 | 85.3 | 75.3 | 84.1 | 86.7 | 83.7 | 82.3 | 83.7 | 83.8 | 83.7 |
| SwinV2-B | 84.1 | 78.9 | 84.3 | 84.1 | 87.9 | 85.7 | 88.4 | 85.8 | 88.4 | 85.8 | 76.7 | 84.5 | 87.1 | 84.2 | 83.1 | 84.1 | 84.3 | 84.2 |
| CLIP (ViT-B/32) | 57.3 | 57.3 | 57.1 | 56.7 | 57.9 | 57.5 | 59.3 | 60.1 | 67.1 | 70.7 | 55.4 | 57.1 | 61.8 | 57.2 | 52.7 | 57.2 | 55.0 | 57.2 |
| CLIP (ViT-B/16) | 62.9 | 62.6 | 62.5 | 62.1 | 63.4 | 63.2 | 64.6 | 65.6 | 69.9 | 66.2 | 60.2 | 62.5 | 67.4 | 63.0 | 59.4 | 63.0 | 61.4 | 63.0 |
| CLIP (ViT-L/14) | 70.1 | 72.1 | 70.0 | 69.8 | 70.9 | 70.3 | 73.0 | 72.8 | 76.9 | 66.6 | 65.7 | 70.5 | 74.6 | 70.1 | 67.3 | 70.1 | 69.2 | 69.9 |

### (d) ImageNet-21K

| Model | Acc. | Uncal. | IRM | I-Max | TS | TS$_{TvA}$ | VS | VS$_{reg\_TvA}$ | DC | DC$_{reg\_TvA}$ | Beta | Beta$_{TvA}$ | Iso | Iso$_{TvA}$ | BBQ | BBQ$_{TvA}$ | HB | HB$_{TvA}$ |
|---|---|---|---|---|---|---|---|---|---|---|---|---|---|---|---|---|---|---|
| MN3 | 15.4 | 27.7 | err. | err. | 24.0 | 17.4 | 35.5 | 33.3 | 71.9 | 81.2 | err. | 15.1 | 35.1 | 15.3 | err. | 15.3 | 15.7 | 15.3 |
| ViT-B/16 | 19.2 | 21.4 | err. | err. | 25.5 | 21.9 | 43.4 | 41.3 | 48.7 | 41.5 | err. | 19.3 | 42.8 | 19.2 | err. | 19.1 | 20.6 | 19.2 |

### (e) Amazon Fine Foods

| Model | Acc. | Uncal. | IRM | I-Max | TS | TS$_{TvA}$ | VS | VS$_{reg\_TvA}$ | DC | DC$_{reg\_TvA}$ | Beta | Beta$_{TvA}$ | Iso | Iso$_{TvA}$ | BBQ | BBQ$_{TvA}$ | HB | HB$_{TvA}$ |
|---|---|---|---|---|---|---|---|---|---|---|---|---|---|---|---|---|---|---|
| T5 | 89.8 | 95.0 | 89.8 | 89.8 | 90.2 | 90.3 | 90.8 | 91.5 | 90.7 | 91.6 | 96.0 | 89.8 | 90.5 | 89.8 | 90.4 | 89.8 | 90.2 | 89.8 |
| T5-large | 91.6 | 97.3 | 91.6 | 91.6 | 92.0 | 91.9 | 92.3 | 93.3 | 92.3 | 93.3 | 97.8 | 91.6 | 92.1 | 91.6 | 92.1 | 91.6 | 91.9 | 91.6 |
| RoBERTa | 90.0 | 97.8 | 90.0 | 90.0 | 92.2 | 92.2 | 90.6 | 91.7 | 90.7 | 91.7 | 98.0 | 90.0 | 90.5 | 90.0 | 90.4 | 90.0 | 90.2 | 90.0 |
| RoBERTa-large | 91.4 | 98.2 | 91.5 | 91.5 | 93.9 | 93.9 | 92.3 | 93.3 | 92.3 | 93.3 | 98.3 | 91.5 | 92.1 | 91.5 | 91.9 | 91.5 | 91.7 | 91.4 |

### (f) DynaSent

| Model | Acc. | Uncal. | IRM | I-Max | TS | TS$_{TvA}$ | VS | VS$_{reg\_TvA}$ | DC | DC$_{reg\_TvA}$ | Beta | Beta$_{TvA}$ | Iso | Iso$_{TvA}$ | BBQ | BBQ$_{TvA}$ | HB | HB$_{TvA}$ |
|---|---|---|---|---|---|---|---|---|---|---|---|---|---|---|---|---|---|---|
| T5 | 78.8 | 86.8 | 78.5 | 78.5 | 78.9 | 78.8 | 83.6 | 80.5 | 83.4 | 80.6 | 89.7 | 78.5 | 78.9 | 78.3 | 78.7 | 78.3 | 78.4 | 78.2 |
| T5-large | 82.2 | 91.8 | 81.8 | 81.7 | 85.0 | 85.0 | 89.5 | 83.9 | 89.3 | 83.9 | 94.1 | 82.2 | 82.2 | 81.6 | 82.0 | 81.6 | 81.8 | 81.6 |
| RoBERTa | 77.7 | 95.1 | 77.9 | 77.8 | 90.9 | 90.9 | 93.5 | 85.4 | 92.7 | 84.1 | 96.1 | 78.3 | 78.4 | 77.8 | 78.0 | 77.7 | 78.0 | 77.8 |
| RoBERTa-large | 81.2 | 96.0 | 81.2 | 81.2 | 92.1 | 92.1 | 94.6 | 86.7 | 94.0 | 85.9 | 96.9 | 80.5 | 81.7 | 81.0 | 81.4 | 80.9 | 81.3 | 81.0 |

### (g) MNLI

| Model | Acc. | Uncal. | IRM | I-Max | TS | TS$_{TvA}$ | VS | VS$_{reg\_TvA}$ | DC | DC$_{reg\_TvA}$ | Beta | Beta$_{TvA}$ | Iso | Iso$_{TvA}$ | BBQ | BBQ$_{TvA}$ | HB | HB$_{TvA}$ |
|---|---|---|---|---|---|---|---|---|---|---|---|---|---|---|---|---|---|---|
| T5 | 88.4 | 94.8 | 88.5 | 88.5 | 89.4 | 89.4 | 91.6 | 90.2 | 91.6 | 90.3 | 96.1 | 89.0 | 88.7 | 88.5 | 88.6 | 88.5 | 88.4 | 88.5 |
| T5-large | 90.1 | 97.6 | 90.0 | 89.9 | 94.5 | 94.5 | 95.7 | 91.7 | 95.5 | 91.7 | 98.3 | 90.2 | 90.2 | 89.9 | 90.1 | 89.9 | 90.0 | 89.9 |
| RoBERTa | 86.4 | 96.7 | 86.1 | 86.1 | 92.9 | 92.9 | 94.2 | 88.4 | 93.8 | 88.5 | 97.5 | 86.1 | 86.5 | 86.1 | 86.2 | 86.0 | 86.1 | 86.1 |
| RoBERTa-large | 88.9 | 97.1 | 88.8 | 88.7 | 93.8 | 93.8 | 95.0 | 90.6 | 94.7 | 90.7 | 97.7 | 88.9 | 89.1 | 88.7 | 88.9 | 88.7 | 88.8 | 88.7 |

### (h) Yahoo Answers

| Model | Acc. | Uncal. | IRM | I-Max | TS | TS$_{TvA}$ | VS | VS$_{reg\_TvA}$ | DC | DC$_{reg\_TvA}$ | Beta | Beta$_{TvA}$ | Iso | Iso$_{TvA}$ | BBQ | BBQ$_{TvA}$ | HB | HB$_{TvA}$ |
|---|---|---|---|---|---|---|---|---|---|---|---|---|---|---|---|---|---|---|
| T5 | 75.3 | 82.0 | 75.6 | 75.4 | 75.1 | 74.6 | 78.0 | 76.1 | 78.0 | 76.0 | 82.9 | 74.8 | 76.5 | 75.4 | 75.4 | 75.4 | 75.0 | 75.4 |
| T5-large | 75.6 | 84.6 | 75.8 | 75.6 | 75.4 | 74.8 | 80.3 | 76.3 | 80.5 | 76.4 | 85.8 | 75.9 | 77.0 | 75.6 | 75.8 | 75.6 | 75.3 | 75.6 |
| RoBERTa | 72.4 | 91.9 | 72.6 | 72.5 | 84.4 | 84.4 | 88.6 | 74.6 | 88.6 | 74.7 | 92.8 | 72.0 | 74.4 | 72.3 | 72.7 | 72.3 | 72.4 | 72.3 |
| RoBERTa-large | 72.8 | 92.5 | 73.1 | 73.0 | 85.6 | 85.6 | 89.5 | 75.6 | 89.3 | 76.1 | 93.2 | 72.6 | 74.8 | 72.9 | 73.2 | 72.9 | 73.0 | 73.0 |

Table 9: Accuracy in % (higher is better). Methods in purple impact the model prediction, potentially degrading accuracy; methods in teal do not. Because classifiers can be well calibrated when not accurate (by having low accuracy and low confidence), it is important to monitor the accuracy. It is even better when the methods preserve the accuracy by design.

(a) CIFAR-10

| Model | Uncal. | IRM | I-Max | TS | TS$_{\text{TvA}}$ | VS | VS$_{\text{reg\_TvA}}$ | DC | DC$_{\text{reg\_TvA}}$ | Beta | Beta$_{\text{TvA}}$ | Iso | Iso$_{\text{TvA}}$ | BBQ | BBQ$_{\text{TvA}}$ | HB | HB$_{\text{TvA}}$ |
|---|---|---|---|---|---|---|---|---|---|---|---|---|---|---|---|---|---|
| ResNet-50 | 94.89 | 94.92 | 94.89 | 94.89 | 94.89 | 94.84 | 94.83 | 94.84 | 94.83 | 94.84 | 94.89 | 94.83 | 94.89 | 94.72 | 94.89 | 94.70 | 94.89 |
| ResNet-110 | 94.55 | 94.53 | 94.51 | 94.55 | 94.55 | 94.50 | 94.52 | 94.51 | 94.52 | 94.55 | 94.55 | 94.42 | 94.55 | 94.40 | 94.55 | 94.32 | 94.55 |
| WRN | 95.79 | 95.80 | 95.76 | 95.79 | 95.79 | 95.68 | 95.73 | 95.68 | 95.74 | 95.80 | 95.79 | 95.63 | 95.79 | 95.65 | 95.79 | 95.59 | 95.79 |
| DenseNet | 94.99 | 94.98 | 94.97 | 94.99 | 94.99 | 95.05 | 95.05 | 95.05 | 95.05 | 94.99 | 94.99 | 94.86 | 94.99 | 94.74 | 94.99 | 94.80 | 94.99 |
| CLIP (ViT-B/32) | 88.17 | 88.16 | 87.92 | 88.17 | 88.17 | 90.29 | 90.28 | 90.38 | 90.33 | 90.18 | 88.17 | 90.36 | 88.17 | 90.29 | 88.17 | 90.18 | 88.17 |
| CLIP (ViT-B/16) | 90.23 | 90.16 | 90.01 | 90.23 | 90.23 | 92.33 | 92.30 | 92.40 | 92.33 | 92.10 | 90.23 | 92.21 | 90.23 | 92.19 | 90.23 | 91.90 | 90.23 |
| CLIP (ViT-L/14) | 95.28 | 95.22 | 88.91 | 95.28 | 95.28 | 96.48 | 96.50 | 96.49 | 96.48 | 96.31 | 95.28 | 96.31 | 95.28 | 96.39 | 95.28 | 96.25 | 95.28 |

(b) CIFAR-100

| Model | Uncal. | IRM | I-Max | TS | TS$_{\text{TvA}}$ | VS | VS$_{\text{reg\_TvA}}$ | DC | DC$_{\text{reg\_TvA}}$ | Beta | Beta$_{\text{TvA}}$ | Iso | Iso$_{\text{TvA}}$ | BBQ | BBQ$_{\text{TvA}}$ | HB | HB$_{\text{TvA}}$ |
|---|---|---|---|---|---|---|---|---|---|---|---|---|---|---|---|---|---|
| ResNet-50 | 76.71 | 76.64 | 76.72 | 76.71 | 76.71 | 76.63 | 76.50 | 76.65 | 76.50 | 76.67 | 76.71 | 76.24 | 76.71 | 76.01 | 76.71 | 74.89 | 76.71 |
| ResNet-110 | 75.00 | 75.00 | 74.91 | 75.00 | 75.00 | 75.00 | 74.94 | 75.00 | 74.94 | 74.96 | 75.00 | 74.66 | 75.00 | 74.14 | 75.00 | 73.12 | 75.00 |
| WRN | 79.57 | 79.52 | 79.42 | 79.57 | 79.57 | 79.36 | 79.34 | 79.36 | 79.36 | 79.51 | 79.57 | 79.08 | 79.57 | 78.57 | 79.57 | 77.35 | 79.57 |
| DenseNet | 76.26 | 76.31 | 76.21 | 76.26 | 76.26 | 76.31 | 76.31 | 76.33 | 76.31 | 76.30 | 76.26 | 76.07 | 76.26 | 75.48 | 76.26 | 74.42 | 76.26 |
| CLIP (ViT-B/32) | 62.33 | 62.18 | 61.60 | 62.33 | 62.33 | 67.77 | 67.33 | 66.31 | 63.95 | 66.40 | 62.33 | 66.48 | 62.33 | 63.85 | 62.33 | 64.06 | 62.33 |
| CLIP (ViT-B/16) | 66.66 | 66.62 | 66.10 | 66.66 | 66.66 | 71.35 | 71.02 | 70.18 | 68.48 | 70.04 | 66.66 | 70.21 | 66.66 | 67.38 | 66.66 | 67.55 | 66.66 |
| CLIP (ViT-L/14) | 75.96 | 75.87 | 75.67 | 75.96 | 75.96 | 80.09 | 79.87 | 79.96 | 79.02 | 79.38 | 75.96 | 79.52 | 75.96 | 77.29 | 75.96 | 77.05 | 75.96 |

(c) ImageNet

| Model | Uncal. | IRM | I-Max | TS | TS$_{\text{TvA}}$ | VS | VS$_{\text{reg\_TvA}}$ | DC | DC$_{\text{reg\_TvA}}$ | Beta | Beta$_{\text{TvA}}$ | Iso | Iso$_{\text{TvA}}$ | BBQ | BBQ$_{\text{TvA}}$ | HB | HB$_{\text{TvA}}$ |
|---|---|---|---|---|---|---|---|---|---|---|---|---|---|---|---|---|---|
| ResNet-18 | 69.77 | 69.76 | 69.37 | 69.77 | 69.77 | 69.81 | 69.21 | 68.12 | 67.60 | 69.72 | 69.77 | 69.24 | 69.77 | 68.34 | 69.77 | 66.65 | 69.77 |
| ResNet-34 | 73.23 | 73.19 | 72.77 | 73.23 | 73.23 | 73.19 | 72.70 | 71.96 | 71.53 | 73.10 | 73.23 | 72.81 | 73.23 | 72.21 | 73.23 | 70.46 | 73.23 |
| ResNet-50 | 80.85 | 80.80 | 80.26 | 80.85 | 80.85 | 80.93 | 80.81 | 80.94 | 80.81 | 80.60 | 80.85 | 80.47 | 80.85 | 78.13 | 80.85 | 79.57 | 80.85 |
| ResNet-101 | 81.86 | 81.83 | 81.54 | 81.86 | 81.86 | 81.77 | 81.65 | 81.80 | 81.66 | 81.74 | 81.86 | 81.44 | 81.86 | 80.82 | 81.86 | 79.80 | 81.86 |
| EffNet-B7 | 84.18 | 84.18 | 84.06 | 84.16 | 84.16 | 84.45 | 84.31 | 84.45 | 84.30 | 84.27 | 84.16 | 84.09 | 84.16 | 83.71 | 84.16 | 82.72 | 84.16 |
| EffNetV2-S | 84.27 | 84.19 | 83.99 | 84.27 | 84.27 | 84.33 | 84.23 | 84.34 | 84.22 | 84.26 | 84.27 | 83.88 | 84.27 | 83.52 | 84.27 | 82.55 | 84.27 |
| EffNetV2-M | 85.06 | 85.05 | 84.90 | 85.06 | 85.06 | 85.28 | 85.19 | 85.28 | 85.17 | 85.10 | 85.06 | 84.87 | 85.06 | 84.19 | 85.07 | 83.72 | 85.06 |
| EffNetV2-L | 85.80 | 85.78 | 85.60 | 85.80 | 85.80 | 85.88 | 85.83 | 85.89 | 85.87 | 85.89 | 85.80 | 85.58 | 85.80 | 85.23 | 85.80 | 84.25 | 85.80 |
| ConvNeXt-T | 82.50 | 82.49 | 82.18 | 82.50 | 82.50 | 82.44 | 82.29 | 82.45 | 82.28 | 82.45 | 82.50 | 82.10 | 82.50 | 81.51 | 82.50 | 80.37 | 82.50 |
| ConvNeXt-S | 83.65 | 83.59 | 83.36 | 83.65 | 83.65 | 83.63 | 83.55 | 83.63 | 83.55 | 83.64 | 83.65 | 83.28 | 83.65 | 82.89 | 83.65 | 81.84 | 83.65 |
| ConvNeXt-B | 84.04 | 84.01 | 83.78 | 84.04 | 84.04 | 84.09 | 83.98 | 84.10 | 83.96 | 84.04 | 84.04 | 83.68 | 84.04 | 83.22 | 84.04 | 82.34 | 84.04 |
| ConvNeXt-L | 84.38 | 84.37 | 84.23 | 84.38 | 84.38 | 84.41 | 84.32 | 84.41 | 84.34 | 84.44 | 84.38 | 84.12 | 84.38 | 83.98 | 84.38 | 82.92 | 84.38 |
| ViT-B/32 | 75.95 | 75.91 | 75.69 | 75.95 | 75.95 | 75.81 | 75.65 | 75.83 | 75.66 | 75.85 | 75.95 | 75.36 | 75.95 | 74.59 | 75.95 | 73.13 | 75.95 |
| ViT-B/16 | 81.04 | 81.01 | 80.90 | 81.04 | 81.04 | 81.00 | 80.88 | 81.01 | 80.87 | 80.96 | 81.04 | 80.63 | 81.04 | 80.38 | 81.04 | 79.06 | 81.04 |
| ViT-L/32 | 76.96 | 76.94 | 76.84 | 76.96 | 76.96 | 76.79 | 76.73 | 76.79 | 76.72 | 76.88 | 76.96 | 76.37 | 76.96 | 76.04 | 76.96 | 74.55 | 76.96 |
| ViT-L/16 | 79.64 | 79.64 | 79.59 | 79.64 | 79.64 | 79.80 | 79.67 | 79.81 | 79.68 | 79.66 | 79.64 | 79.47 | 79.64 | 79.20 | 79.64 | 77.82 | 79.64 |
| ViT-H/14 | 88.62 | 88.61 | 88.48 | 88.62 | 88.62 | 88.50 | 88.50 | 88.59 | 88.46 | 88.63 | 88.62 | 88.34 | 88.62 | 88.33 | 88.62 | 87.24 | 88.62 |
| Swin-T | 81.49 | 81.45 | 81.28 | 81.49 | 81.49 | 81.55 | 81.42 | 81.55 | 81.42 | 81.44 | 81.49 | 81.07 | 81.49 | 80.77 | 81.49 | 79.43 | 81.49 |
| Swin-S | 83.21 | 83.20 | 83.04 | 83.21 | 83.21 | 83.13 | 83.02 | 83.13 | 83.03 | 83.21 | 83.21 | 82.79 | 83.21 | 82.74 | 83.21 | 81.40 | 83.21 |
| Swin-B | 83.60 | 83.57 | 83.53 | 83.60 | 83.60 | 83.75 | 83.61 | 83.76 | 83.59 | 83.63 | 83.60 | 83.39 | 83.60 | 83.40 | 83.60 | 82.16 | 83.60 |
| SwinV2-T | 82.02 | 82.01 | 81.83 | 82.02 | 82.02 | 82.12 | 81.98 | 82.13 | 82.00 | 82.05 | 82.02 | 81.66 | 82.02 | 81.27 | 82.02 | 80.08 | 82.02 |
| SwinV2-S | 83.74 | 83.73 | 83.64 | 83.74 | 83.74 | 83.81 | 83.71 | 83.81 | 83.72 | 83.74 | 83.74 | 83.56 | 83.74 | 83.34 | 83.74 | 82.31 | 83.74 |
| SwinV2-B | 84.10 | 84.12 | 84.03 | 84.10 | 84.10 | 84.14 | 84.06 | 84.16 | 84.08 | 84.16 | 84.10 | 83.81 | 84.10 | 83.79 | 84.10 | 82.53 | 84.10 |
| CLIP (ViT-B/32) | 57.34 | 57.32 | 56.94 | 57.34 | 57.34 | 59.80 | 59.57 | 31.10 | 0.18 | 58.93 | 57.34 | 59.54 | 57.34 | 57.25 | 57.34 | 56.05 | 57.34 |
| CLIP (ViT-B/16) | 62.89 | 62.91 | 62.43 | 62.89 | 62.89 | 65.61 | 65.28 | 35.86 | 0.18 | 64.62 | 62.89 | 65.07 | 62.89 | 63.24 | 62.89 | 62.18 | 62.89 |
| CLIP (ViT-L/14) | 70.15 | 70.14 | 69.89 | 70.15 | 70.15 | 72.66 | 72.25 | 50.88 | 0.17 | 71.57 | 70.15 | 72.30 | 70.15 | 70.59 | 70.15 | 69.36 | 70.15 |

(d) ImageNet-21K

| Model | Uncal. | IRM | I-Max | TS | TS$_{\text{TvA}}$ | VS | VS$_{\text{reg\_TvA}}$ | DC | DC$_{\text{reg\_TvA}}$ | Beta | Beta$_{\text{TvA}}$ | Iso | Iso$_{\text{TvA}}$ | BBQ | BBQ$_{\text{TvA}}$ | HB | HB$_{\text{TvA}}$ |
|---|---|---|---|---|---|---|---|---|---|---|---|---|---|---|---|---|---|
| MN3 | 15.36 | err. | err. | 15.36 | 15.36 | 37.26 | 34.95 | 13.07 | 0.02 | err. | 15.36 | 35.62 | 15.36 | err. | 15.36 | 21.17 | 15.36 |
| ViT-B/16 | 19.18 | err. | err. | 19.18 | 19.18 | 45.42 | 42.70 | 40.54 | 40.70 | err. | 19.18 | 43.72 | 19.18 | err. | 19.18 | 28.46 | 19.18 |

(e) Amazon Fine Foods

| Model | Uncal. | IRM | I-Max | TS | TS$_{\text{TvA}}$ | VS | VS$_{\text{reg\_TvA}}$ | DC | DC$_{\text{reg\_TvA}}$ | Beta | Beta$_{\text{TvA}}$ | Iso | Iso$_{\text{TvA}}$ | BBQ | BBQ$_{\text{TvA}}$ | HB | HB$_{\text{TvA}}$ |
|---|---|---|---|---|---|---|---|---|---|---|---|---|---|---|---|---|---|
| T5 | 89.81 | 89.83 | 89.83 | 89.81 | 89.81 | 90.04 | 90.34 | 90.31 | 90.34 | 90.59 | 89.81 | 90.60 | 89.81 | 90.58 | 89.81 | 90.64 | 89.81 |
| T5-large | 91.57 | 91.58 | 91.61 | 91.57 | 91.57 | 91.67 | 91.82 | 92.04 | 91.98 | 92.10 | 91.57 | 92.14 | 91.57 | 92.14 | 91.57 | 92.11 | 91.57 |
| RoBERTa | 89.95 | 89.95 | 89.98 | 89.95 | 89.95 | 89.99 | 90.15 | 90.29 | 90.21 | 90.56 | 89.95 | 90.59 | 89.95 | 90.59 | 89.95 | 90.54 | 89.95 |
| RoBERTa-large | 91.42 | 91.50 | 91.48 | 91.42 | 91.42 | 91.42 | 91.64 | 91.74 | 91.69 | 91.94 | 91.42 | 91.95 | 91.42 | 91.90 | 91.39 | 91.91 | 91.42 |

(f) DynaSent

| Model | Uncal. | IRM | I-Max | TS | TS$_{\text{TvA}}$ | VS | VS$_{\text{reg\_TvA}}$ | DC | DC$_{\text{reg\_TvA}}$ | Beta | Beta$_{\text{TvA}}$ | Iso | Iso$_{\text{TvA}}$ | BBQ | BBQ$_{\text{TvA}}$ | HB | HB$_{\text{TvA}}$ |
|---|---|---|---|---|---|---|---|---|---|---|---|---|---|---|---|---|---|
| T5 | 78.83 | 78.88 | 78.82 | 78.83 | 78.83 | 78.84 | 78.83 | 78.82 | 78.86 | 78.81 | 78.83 | 78.70 | 78.83 | 78.71 | 78.83 | 78.73 | 78.83 |
| T5-large | 82.20 | 82.16 | 82.19 | 82.20 | 82.20 | 82.20 | 82.23 | 82.32 | 82.25 | 82.35 | 82.20 | 82.31 | 82.20 | 82.19 | 82.20 | 82.29 | 82.20 |
| RoBERTa | 77.72 | 77.67 | 77.71 | 77.72 | 77.72 | 77.72 | 77.74 | 77.76 | 77.70 | 77.77 | 77.72 | 77.75 | 77.72 | 77.86 | 77.72 | 77.97 | 77.72 |
| RoBERTa-large | 81.19 | 81.29 | 81.33 | 81.19 | 81.19 | 81.19 | 81.19 | 81.24 | 81.44 | 81.28 | 81.19 | 81.50 | 81.19 | 81.51 | 81.19 | 81.66 | 81.19 |

(g) MNLI

| Model | Uncal. | IRM | I-Max | TS | TS$_{\text{TvA}}$ | VS | VS$_{\text{reg\_TvA}}$ | DC | DC$_{\text{reg\_TvA}}$ | Beta | Beta$_{\text{TvA}}$ | Iso | Iso$_{\text{TvA}}$ | BBQ | BBQ$_{\text{TvA}}$ | HB | HB$_{\text{TvA}}$ |
|---|---|---|---|---|---|---|---|---|---|---|---|---|---|---|---|---|---|
| T5 | 88.35 | 88.35 | 88.38 | 88.35 | 88.35 | 88.36 | 88.39 | 88.39 | 88.39 | 88.41 | 88.35 | 88.34 | 88.35 | 88.28 | 88.35 | 88.37 | 88.35 |
| T5-large | 90.07 | 90.10 | 90.14 | 90.07 | 90.07 | 90.07 | 90.15 | 90.20 | 90.21 | 90.07 | 90.07 | 90.15 | 90.07 | 90.06 | 90.07 | 90.16 | 90.07 |
| RoBERTa | 86.41 | 86.40 | 86.43 | 86.41 | 86.41 | 86.41 | 86.42 | 86.45 | 86.48 | 86.44 | 86.41 | 86.37 | 86.41 | 86.41 | 86.41 | 86.38 | 86.41 |
| RoBERTa-large | 88.89 | 88.90 | 88.92 | 88.89 | 88.89 | 88.90 | 88.96 | 88.98 | 89.02 | 88.90 | 88.89 | 88.95 | 88.89 | 88.86 | 88.89 | 88.92 | 88.89 |

(h) Yahoo Answers

| Model | Uncal. | IRM | I-Max | TS | TS$_{\text{TvA}}$ | VS | VS$_{\text{reg\_TvA}}$ | DC | DC$_{\text{reg\_TvA}}$ | Beta | Beta$_{\text{TvA}}$ | Iso | Iso$_{\text{TvA}}$ | BBQ | BBQ$_{\text{TvA}}$ | HB | HB$_{\text{TvA}}$ |
|---|---|---|---|---|---|---|---|---|---|---|---|---|---|---|---|---|---|
| T5 | 75.35 | 75.35 | 75.26 | 75.35 | 75.35 | 75.34 | 75.34 | 75.36 | 75.35 | 75.24 | 75.35 | 75.30 | 75.35 | 75.22 | 75.35 | 75.25 | 75.35 |
| T5-large | 75.57 | 75.60 | 75.56 | 75.57 | 75.57 | 75.59 | 75.55 | 75.69 | 75.66 | 75.53 | 75.57 | 75.70 | 75.57 | 75.62 | 75.57 | 75.67 | 75.57 |
| RoBERTa | 72.38 | 72.37 | 72.35 | 72.38 | 72.38 | 72.39 | 72.43 | 72.74 | 73.05 | 72.72 | 72.38 | 73.00 | 72.38 | 72.87 | 72.38 | 72.91 | 72.38 |
| RoBERTa-large | 72.84 | 72.83 | 72.81 | 72.84 | 72.84 | 72.83 | 72.88 | 73.01 | 73.40 | 72.98 | 72.84 | 73.48 | 72.84 | 73.29 | 72.84 | 73.38 | 72.84 |

Table 10: ECE with 15 equal mass bins in % (lower is better). Methods in purple impact the model prediction, potentially degrading accuracy; methods in teal do not.

### (a) CIFAR-10

| Model | Uncal. | IRM | I-Max | TS | TS$_{\text{TvA}}$ | VS | VS$_{\text{reg\_TvA}}$ | DC | DC$_{\text{reg\_TvA}}$ | Beta | Beta$_{\text{TvA}}$ | Iso | Iso$_{\text{TvA}}$ | BBQ | BBQ$_{\text{TvA}}$ | HB | HB$_{\text{TvA}}$ |
|---|---|---|---|---|---|---|---|---|---|---|---|---|---|---|---|---|---|
| ResNet-50 | 1.81 | 0.71 | 0.62 | 1.33 | 1.24 | 1.28 | 1.29 | 1.28 | 1.29 | 2.03 | 1.55 | 0.93 | 0.70 | 2.45 | 0.47 | 2.33 | **0.46** |
| ResNet-110 | 2.58 | 0.48 | 0.51 | 1.78 | 1.72 | 1.74 | 1.74 | 1.72 | 1.75 | 2.91 | 1.56 | 0.99 | 0.43 | 2.59 | 0.34 | 2.71 | **0.23** |
| WRN | 1.80 | 0.58 | 0.51 | 1.74 | 1.75 | 1.48 | 1.52 | 1.49 | 1.52 | 1.74 | 2.09 | 0.98 | 0.62 | 2.07 | 0.54 | 2.15 | **0.46** |
| DenseNet | 2.04 | 0.56 | 0.46 | 2.00 | 2.07 | 1.48 | 1.70 | 1.47 | 1.70 | 2.03 | 2.45 | 0.78 | 0.56 | 2.71 | 0.69 | 2.75 | **0.29** |
| CLIP (ViT-B/32) | 4.73 | 1.33 | 1.30 | 0.93 | 0.91 | 2.76 | 1.79 | 2.80 | 1.80 | 1.66 | 1.34 | 1.06 | 1.05 | 2.15 | 1.47 | 2.47 | **0.90** |
| CLIP (ViT-B/16) | 5.38 | 1.12 | 1.01 | 0.59 | **0.48** | 2.91 | 1.91 | 2.87 | 1.87 | 1.21 | 1.80 | 1.05 | 0.87 | 1.63 | 0.91 | 2.41 | 0.63 |
| CLIP (ViT-L/14) | 4.90 | 0.54 | 0.50 | 0.56 | 0.48 | 1.98 | 1.74 | 1.93 | 1.72 | 0.96 | 0.94 | 0.55 | 0.46 | 0.98 | 0.51 | 0.96 | **0.37** |

### (b) CIFAR-100

| Model | Uncal. | IRM | I-Max | TS | TS$_{\text{TvA}}$ | VS | VS$_{\text{reg\_TvA}}$ | DC | DC$_{\text{reg\_TvA}}$ | Beta | Beta$_{\text{TvA}}$ | Iso | Iso$_{\text{TvA}}$ | BBQ | BBQ$_{\text{TvA}}$ | HB | HB$_{\text{TvA}}$ |
|---|---|---|---|---|---|---|---|---|---|---|---|---|---|---|---|---|---|
| ResNet-50 | 6.53 | 1.55 | 1.26 | 5.00 | 3.54 | 5.15 | 2.31 | 5.14 | 2.30 | 5.54 | 3.90 | 5.60 | 1.44 | 10.10 | 2.08 | 10.97 | **1.19** |
| ResNet-110 | 7.83 | 1.27 | 1.18 | 5.31 | 4.24 | 5.11 | 2.86 | 5.07 | 2.90 | 5.97 | 4.95 | 6.49 | 1.30 | 9.72 | 2.11 | 10.89 | **1.13** |
| WRN | 4.33 | 1.07 | 0.95 | 4.36 | 2.79 | 4.46 | 2.38 | 4.45 | 2.37 | 4.40 | 2.85 | 4.28 | 1.22 | 10.29 | 0.94 | 9.99 | **0.83** |
| DenseNet | 5.16 | 1.12 | **0.86** | 4.25 | 2.30 | 4.47 | 2.26 | 4.48 | 2.25 | 4.83 | 2.94 | 4.50 | 1.37 | 10.46 | 1.27 | 10.60 | 1.07 |
| CLIP (ViT-B/32) | 9.51 | 2.10 | 1.74 | 1.86 | 1.78 | 8.72 | 3.49 | 7.97 | 1.99 | 6.51 | 2.75 | 2.32 | **1.25** | 7.95 | 2.13 | 7.99 | 1.35 |
| CLIP (ViT-B/16) | 10.64 | 3.35 | 3.04 | 2.66 | 2.72 | 8.68 | 3.15 | 8.20 | 1.72 | 7.03 | 2.55 | 2.68 | 1.77 | 7.14 | 2.12 | 7.13 | **1.52** |
| CLIP (ViT-L/14) | 10.96 | 3.14 | 2.94 | 2.47 | 2.53 | 6.01 | 2.05 | 6.57 | 1.74 | 6.52 | 1.93 | 2.39 | 1.64 | 6.65 | 1.66 | 6.48 | **1.44** |

### (c) ImageNet

| Model | Uncal. | IRM | I-Max | TS | TS$_{\text{TvA}}$ | VS | VS$_{\text{reg\_TvA}}$ | DC | DC$_{\text{reg\_TvA}}$ | Beta | Beta$_{\text{TvA}}$ | Iso | Iso$_{\text{TvA}}$ | BBQ | BBQ$_{\text{TvA}}$ | HB | HB$_{\text{TvA}}$ |
|---|---|---|---|---|---|---|---|---|---|---|---|---|---|---|---|---|---|
| ResNet-18 | 2.59 | 0.78 | **0.55** | 1.86 | 1.82 | 1.71 | 2.07 | 3.36 | 3.35 | 4.36 | 1.39 | 3.85 | 0.75 | 9.84 | 0.65 | 9.45 | 0.56 |
| ResNet-34 | 3.61 | 0.74 | **0.52** | 1.75 | 1.75 | 1.81 | 2.02 | 3.40 | 2.91 | 4.86 | 1.11 | 4.04 | 0.71 | 9.17 | 0.68 | 9.29 | 0.64 |
| ResNet-50 | 41.15 | 2.75 | 2.60 | 3.19 | 1.76 | 3.23 | 1.10 | 3.22 | 1.13 | 11.30 | 2.22 | 1.29 | 0.76 | 8.24 | 0.98 | 5.45 | **0.52** |
| ResNet-101 | 13.55 | 0.79 | 0.56 | 3.69 | 2.34 | 4.21 | 1.60 | 4.18 | 1.56 | 9.21 | 2.10 | 3.01 | 0.63 | 7.84 | 1.00 | 6.68 | **0.46** |
| EffNet-B7 | 12.60 | 0.51 | 0.48 | 3.84 | 2.94 | 3.82 | 1.58 | 3.81 | 1.56 | 9.31 | 2.32 | 2.93 | 0.56 | 6.91 | 0.71 | 6.06 | **0.39** |
| EffNetV2-S | 16.92 | 0.63 | **0.40** | 4.04 | 3.32 | 3.91 | 1.69 | 3.89 | 1.67 | 7.98 | 2.52 | 2.96 | 0.65 | 7.58 | 0.92 | 6.50 | 0.58 |
| EffNetV2-M | 24.88 | 0.90 | 0.67 | 3.78 | 2.66 | 3.83 | 1.35 | 3.82 | 1.34 | 8.31 | 1.83 | 2.88 | 0.73 | 6.68 | 1.08 | 5.38 | **0.54** |
| EffNetV2-L | 8.48 | 0.60 | 0.44 | 2.84 | 1.49 | 3.06 | 0.94 | 3.05 | 0.90 | 9.45 | 1.18 | 2.51 | 0.62 | 6.03 | 0.78 | 5.30 | **0.37** |
| ConvNeXt-T | 16.95 | 1.17 | 0.87 | 3.08 | 1.67 | 3.47 | 1.21 | 3.46 | 1.17 | 8.95 | 1.83 | 2.55 | 0.82 | 7.64 | 0.99 | 6.01 | **0.67** |
| ConvNeXt-S | 17.60 | 0.76 | 0.56 | 3.80 | 2.56 | 4.19 | 1.44 | 4.18 | 1.41 | 8.77 | 2.02 | 3.06 | 0.71 | 7.36 | 0.79 | 6.07 | **0.51** |
| ConvNeXt-B | 18.77 | 0.68 | **0.44** | 3.81 | 2.67 | 4.09 | 1.44 | 4.07 | 1.44 | 9.44 | 2.18 | 3.03 | 0.73 | 7.48 | 1.04 | 6.04 | 0.58 |
| ConvNeXt-L | 12.51 | 0.65 | 0.43 | 4.02 | 2.90 | 4.42 | 1.82 | 4.41 | 1.77 | 7.89 | 1.59 | 3.26 | 0.63 | 7.05 | 0.78 | 6.27 | **0.41** |
| ViT-B/32 | 6.37 | 0.71 | 0.61 | 4.10 | 2.49 | 4.64 | 1.90 | 4.62 | 1.84 | 6.53 | 1.75 | 3.58 | 0.71 | 9.24 | 0.74 | 8.30 | **0.58** |
| ViT-B/16 | 5.56 | 0.75 | **0.53** | 4.19 | 3.18 | 4.27 | 2.09 | 4.25 | 2.07 | 7.24 | 2.39 | 3.38 | 0.70 | 7.68 | 0.93 | 7.12 | 0.57 |
| ViT-L/32 | 4.13 | 0.85 | 0.75 | 5.30 | 4.20 | 5.37 | 2.67 | 5.37 | 2.64 | 6.19 | 2.80 | 4.42 | 0.72 | 9.18 | 1.03 | 8.64 | **0.63** |
| ViT-L/16 | 5.17 | 1.02 | **0.59** | 5.92 | 5.20 | 5.28 | 2.74 | 5.26 | 2.69 | 7.36 | 3.59 | 4.10 | 0.76 | 7.82 | 1.29 | 8.32 | 0.60 |
| ViT-H/14 | 0.61 | 0.56 | **0.35** | 1.75 | 0.83 | 1.88 | 1.08 | 1.92 | 1.05 | 8.06 | 0.70 | 2.46 | 0.60 | 3.96 | 0.53 | 4.64 | 0.41 |
| Swin-T | 6.82 | 0.77 | 0.49 | 3.10 | 1.82 | 3.43 | 1.33 | 3.43 | 1.27 | 7.70 | 1.67 | 2.94 | 0.71 | 7.52 | 0.90 | 6.87 | **0.48** |
| Swin-S | 3.57 | 0.59 | 0.52 | 3.92 | 2.98 | 4.17 | 1.84 | 4.17 | 1.82 | 7.88 | 2.33 | 3.29 | 0.59 | 6.78 | 0.83 | 6.97 | **0.50** |
| Swin-B | 4.65 | 0.60 | **0.35** | 4.36 | 3.71 | 4.22 | 2.04 | 4.21 | 2.04 | 7.89 | 2.81 | 3.33 | 0.63 | 6.50 | 0.78 | 6.64 | 0.41 |
| SwinV2-T | 8.31 | 0.72 | **0.46** | 3.62 | 2.21 | 3.92 | 1.60 | 3.90 | 1.59 | 8.66 | 1.83 | 3.07 | 0.71 | 7.97 | 0.67 | 6.84 | 0.52 |
| SwinV2-S | 6.06 | 0.64 | **0.45** | 4.18 | 3.32 | 4.24 | 1.88 | 4.23 | 1.83 | 8.46 | 2.34 | 3.15 | 0.56 | 7.04 | 0.79 | 6.75 | 0.50 |
| SwinV2-B | 5.27 | 0.61 | 0.46 | 4.42 | 3.68 | 4.25 | 1.95 | 4.22 | 1.88 | 7.47 | 2.74 | 3.33 | 0.57 | 6.56 | 0.75 | 6.53 | **0.42** |
| CLIP (ViT-B/32) | 1.51 | 1.00 | **0.73** | 1.62 | 1.56 | 1.42 | 0.81 | 36.01 | 70.52 | 3.59 | 0.84 | 2.25 | 0.98 | 7.88 | 1.04 | 6.66 | 0.79 |
| CLIP (ViT-B/16) | 1.78 | 1.25 | 0.74 | 1.90 | 1.88 | 1.60 | 0.80 | 34.02 | 66.04 | 4.51 | 1.07 | 2.31 | 0.95 | 8.17 | 0.83 | 6.96 | **0.71** |
| CLIP (ViT-L/14) | 2.54 | 1.17 | **0.64** | 2.03 | 2.04 | 1.79 | 1.32 | 26.06 | 66.39 | 5.93 | 1.50 | 2.39 | 1.10 | 9.03 | 0.88 | 7.87 | 0.84 |

### (d) ImageNet-21K

| Model | Uncal. | IRM | I-Max | TS | TS$_{\text{TvA}}$ | VS | VS$_{\text{reg\_TvA}}$ | DC | DC$_{\text{reg\_TvA}}$ | Beta | Beta$_{\text{TvA}}$ | Iso | Iso$_{\text{TvA}}$ | BBQ | BBQ$_{\text{TvA}}$ | HB | HB$_{\text{TvA}}$ |
|---|---|---|---|---|---|---|---|---|---|---|---|---|---|---|---|---|---|
| MN3 | 12.34 | err. | err. | 8.69 | 4.43 | 2.51 | 2.38 | 58.84 | 81.16 | err. | 1.14 | 1.93 | 0.28 | err. | 0.31 | 5.53 | **0.19** |
| ViT-B/16 | 6.54 | err. | err. | 9.03 | 6.86 | 2.34 | 1.55 | 8.18 | 3.20 | err. | 3.72 | 2.14 | 0.21 | err. | 0.43 | 7.90 | **0.17** |

### (e) Amazon Fine Foods

| Model | Uncal. | IRM | I-Max | TS | TS$_{\text{TvA}}$ | VS | VS$_{\text{reg\_TvA}}$ | DC | DC$_{\text{reg\_TvA}}$ | Beta | Beta$_{\text{TvA}}$ | Iso | Iso$_{\text{TvA}}$ | BBQ | BBQ$_{\text{TvA}}$ | HB | HB$_{\text{TvA}}$ |
|---|---|---|---|---|---|---|---|---|---|---|---|---|---|---|---|---|---|
| T5 | 5.17 | 0.28 | 0.22 | 1.35 | 1.36 | 1.35 | 1.43 | 0.99 | 1.30 | 5.44 | 0.87 | 0.47 | 0.29 | 1.14 | **0.21** | 2.51 | **0.21** |
| T5-large | 5.76 | 0.27 | 0.22 | 1.82 | 1.84 | 1.71 | 1.67 | 1.37 | 1.35 | 5.71 | 1.81 | 0.75 | 0.26 | 1.97 | **0.17** | 3.18 | **0.17** |
| RoBERTa | 7.89 | 0.33 | 0.23 | 2.27 | 2.21 | 1.46 | 2.11 | 1.60 | 1.90 | 7.47 | 4.29 | 0.45 | 0.32 | 2.73 | 0.22 | 3.76 | **0.21** |
| RoBERTa-large | 6.82 | 0.34 | 0.18 | 2.53 | 2.45 | 1.47 | 2.13 | 1.29 | 1.93 | 6.35 | 4.21 | 0.56 | 0.28 | 2.81 | 0.24 | 3.68 | **0.16** |

### (f) DynaSent

| Model | Uncal. | IRM | I-Max | TS | TS$_{\text{TvA}}$ | VS | VS$_{\text{reg\_TvA}}$ | DC | DC$_{\text{reg\_TvA}}$ | Beta | Beta$_{\text{TvA}}$ | Iso | Iso$_{\text{TvA}}$ | BBQ | BBQ$_{\text{TvA}}$ | HB | HB$_{\text{TvA}}$ |
|---|---|---|---|---|---|---|---|---|---|---|---|---|---|---|---|---|---|
| T5 | 7.92 | 1.76 | 1.33 | 1.60 | 1.60 | 4.78 | 2.12 | 4.58 | 2.20 | 10.85 | 2.83 | 1.46 | 1.89 | 2.00 | 1.51 | 1.58 | **1.21** |
| T5-large | 9.62 | 1.56 | **1.02** | 3.34 | 3.33 | 7.26 | 2.00 | 6.98 | 2.07 | 11.71 | 4.67 | 1.79 | 1.64 | 1.50 | 1.36 | 1.75 | 1.09 |
| RoBERTa | 17.34 | 2.38 | 1.59 | 13.14 | 13.14 | 15.80 | 7.68 | 14.96 | 6.35 | 18.34 | 10.36 | 1.69 | 2.36 | 2.45 | 1.27 | 2.15 | **1.08** |
| RoBERTa-large | 14.80 | 1.51 | 1.14 | 10.88 | 10.87 | 13.43 | 5.53 | 12.71 | 4.53 | 15.63 | 9.09 | 1.93 | 1.41 | 2.78 | 1.09 | 2.90 | **0.61** |

### (g) MNLI

| Model | Uncal. | IRM | I-Max | TS | TS$_{\text{TvA}}$ | VS | VS$_{\text{reg\_TvA}}$ | DC | DC$_{\text{reg\_TvA}}$ | Beta | Beta$_{\text{TvA}}$ | Iso | Iso$_{\text{TvA}}$ | BBQ | BBQ$_{\text{TvA}}$ | HB | HB$_{\text{TvA}}$ |
|---|---|---|---|---|---|---|---|---|---|---|---|---|---|---|---|---|---|
| T5 | 6.46 | 0.90 | 0.64 | 1.22 | 1.20 | 3.21 | 1.91 | 3.20 | 1.98 | 7.73 | 2.11 | 0.86 | 0.90 | 1.55 | 0.58 | 1.83 | **0.45** |
| T5-large | 7.58 | 0.77 | 0.51 | 4.40 | 4.39 | 5.63 | 1.74 | 5.31 | 1.69 | 8.20 | 4.43 | 1.11 | 0.79 | 1.77 | 0.40 | 2.18 | **0.35** |
| RoBERTa | 10.25 | 0.98 | 0.76 | 6.47 | 6.47 | 7.80 | 2.34 | 7.33 | 2.40 | 11.02 | 6.07 | 1.00 | 1.02 | 2.56 | 0.77 | 3.07 | **0.64** |
| RoBERTa-large | 8.17 | 1.00 | 0.56 | 4.91 | 4.90 | 6.13 | 1.85 | 5.74 | 1.84 | 8.80 | 5.26 | 1.24 | 0.95 | 2.02 | **0.48** | 2.50 | 0.49 |

### (h) Yahoo Answers

| Model | Uncal. | IRM | I-Max | TS | TS$_{\text{TvA}}$ | VS | VS$_{\text{reg\_TvA}}$ | DC | DC$_{\text{reg\_TvA}}$ | Beta | Beta$_{\text{TvA}}$ | Iso | Iso$_{\text{TvA}}$ | BBQ | BBQ$_{\text{TvA}}$ | HB | HB$_{\text{TvA}}$ |
|---|---|---|---|---|---|---|---|---|---|---|---|---|---|---|---|---|---|
| T5 | 6.64 | 0.83 | 0.90 | **0.71** | 0.98 | 2.66 | 0.92 | 2.64 | 0.88 | 7.68 | 1.68 | 1.47 | 0.86 | 3.52 | 0.86 | 4.78 | 0.72 |
| T5-large | 9.04 | 0.88 | 0.68 | 1.46 | 1.71 | 4.67 | 1.26 | 4.82 | 1.27 | 10.32 | 2.36 | 1.81 | 0.86 | 3.72 | 0.89 | 4.89 | **0.65** |
| RoBERTa | 19.53 | 1.06 | 0.79 | 12.02 | 12.00 | 16.26 | 2.18 | 15.85 | 1.70 | 20.13 | 9.36 | 1.94 | 0.99 | 5.15 | 0.72 | 6.48 | **0.62** |
| RoBERTa-large | 19.65 | 1.00 | 0.92 | 12.76 | 12.75 | 16.67 | 2.74 | 16.29 | 2.68 | 20.18 | 10.11 | 1.87 | 0.94 | 6.53 | 0.74 | 6.70 | **0.60** |

Table 11: Brier score of the predicted class in $10^{-2}$ (lower is better). Methods in purple impact the model prediction, potentially degrading accuracy; methods in teal do not.

(a) CIFAR-10

| Model | Uncal. | IRM | I-Max | TS | TS$_\text{TvA}$ | VS | VS$_\text{reg\_TvA}$ | DC | DC$_\text{reg\_TvA}$ | Beta | Beta$_\text{TvA}$ | Iso | Iso$_\text{TvA}$ | BBQ | BBQ$_\text{TvA}$ | HB | HB$_\text{TvA}$ |
|---|---|---|---|---|---|---|---|---|---|---|---|---|---|---|---|---|---|
| ResNet-50 | 3.75 | 3.64 | 3.64 | 3.67 | 3.66 | 3.67 | 3.65 | 3.66 | 3.65 | 3.82 | 3.69 | 3.67 | **3.63** | 4.05 | 3.76 | 4.00 | 3.78 |
| ResNet-110 | 3.95 | 3.64 | 3.65 | 3.75 | 3.72 | 3.72 | 3.73 | 3.73 | 3.73 | 4.02 | 3.73 | **3.62** | 3.64 | 4.21 | 3.76 | 4.25 | 3.80 |
| WRN | 3.06 | **3.02** | 3.02 | 3.11 | 3.08 | 3.11 | 3.07 | 3.11 | 3.07 | 3.17 | 3.10 | **3.02** | 3.03 | 3.37 | 3.09 | 3.43 | 3.13 |
| DenseNet | 3.68 | 3.54 | 3.56 | 3.69 | 3.67 | 3.67 | 3.62 | 3.65 | 3.62 | 3.78 | 3.70 | 3.56 | **3.53** | 4.01 | 3.58 | 4.01 | 3.67 |
| CLIP (ViT-B/32) | 7.63 | 7.37 | 7.33 | 7.27 | 7.26 | 6.31 | 6.19 | 6.24 | **6.16** | 6.43 | 7.33 | 6.35 | 7.35 | 6.42 | 7.37 | 6.62 | 7.36 |
| CLIP (ViT-B/16) | 6.48 | 6.08 | 6.07 | 6.02 | 6.01 | 5.15 | 5.05 | 5.12 | **5.03** | 5.21 | 6.15 | 5.20 | 6.09 | 5.24 | 6.08 | 5.39 | 6.13 |
| CLIP (ViT-L/14) | 3.62 | 3.20 | 3.22 | 3.15 | 3.15 | 2.65 | **2.61** | 2.63 | **2.61** | 2.74 | 3.20 | 2.64 | 3.19 | 2.76 | 3.18 | 2.74 | 3.19 |

(b) CIFAR-100

| Model | Uncal. | IRM | I-Max | TS | TS$_\text{TvA}$ | VS | VS$_\text{reg\_TvA}$ | DC | DC$_\text{reg\_TvA}$ | Beta | Beta$_\text{TvA}$ | Iso | Iso$_\text{TvA}$ | BBQ | BBQ$_\text{TvA}$ | HB | HB$_\text{TvA}$ |
|---|---|---|---|---|---|---|---|---|---|---|---|---|---|---|---|---|---|
| ResNet-50 | 12.75 | 11.96 | 12.02 | 12.44 | 12.14 | 12.39 | 11.93 | 12.39 | **11.92** | 12.50 | 12.16 | 12.08 | 11.99 | 14.26 | 12.12 | 13.65 | 12.01 |
| ResNet-110 | 13.86 | **12.69** | 12.74 | 13.29 | 12.94 | 13.27 | 12.82 | 13.26 | 12.82 | 13.37 | 13.11 | 13.07 | 12.70 | 14.88 | 12.80 | 14.24 | 12.75 |
| WRN | 11.05 | 10.72 | 10.74 | 11.05 | 10.85 | 11.03 | **10.71** | 11.03 | 10.72 | 11.08 | 10.82 | 10.77 | 10.75 | 12.65 | 10.82 | 12.07 | 10.86 |
| DenseNet | 12.49 | 12.09 | 12.10 | 12.33 | 12.16 | 12.30 | **12.04** | 12.30 | **12.04** | 12.47 | 12.16 | 12.11 | 12.07 | 14.06 | 12.13 | 13.41 | 12.11 |
| CLIP (ViT-B/32) | 17.25 | 16.20 | 16.12 | 15.88 | 15.90 | 15.98 | **14.72** | 16.03 | 15.09 | 15.92 | 16.28 | 14.97 | 16.23 | 15.55 | 16.31 | 15.36 | 16.25 |
| CLIP (ViT-B/16) | 17.21 | 16.01 | 15.94 | 15.42 | 15.41 | 15.37 | **13.98** | 15.44 | 14.06 | 15.49 | 15.87 | 14.39 | 15.85 | 14.69 | 15.92 | 14.52 | 15.88 |
| CLIP (ViT-L/14) | 14.78 | 13.33 | 13.27 | 12.70 | 12.71 | 11.53 | **10.67** | 11.69 | 10.90 | 12.59 | 13.17 | 11.00 | 13.16 | 11.56 | 13.18 | 11.38 | 13.15 |

(c) ImageNet

| Model | Uncal. | IRM | I-Max | TS | TS$_\text{TvA}$ | VS | VS$_\text{reg\_TvA}$ | DC | DC$_\text{reg\_TvA}$ | Beta | Beta$_\text{TvA}$ | Iso | Iso$_\text{TvA}$ | BBQ | BBQ$_\text{TvA}$ | HB | HB$_\text{TvA}$ |
|---|---|---|---|---|---|---|---|---|---|---|---|---|---|---|---|---|---|
| ResNet-18 | 13.93 | 13.86 | **13.85** | 13.94 | 13.93 | 13.92 | 13.92 | 14.48 | 14.53 | 14.50 | 13.87 | 13.89 | 13.87 | 15.56 | 13.90 | 15.02 | 13.93 |
| ResNet-34 | 13.15 | **12.97** | 12.99 | 13.04 | 13.05 | 12.99 | 13.05 | 13.54 | 13.53 | 13.70 | 12.99 | 13.13 | 12.99 | 14.92 | 13.01 | 14.43 | 13.08 |
| ResNet-50 | 29.79 | 12.25 | 12.33 | 10.95 | 10.88 | 10.98 | 10.92 | 10.98 | 10.92 | 13.14 | 12.07 | **10.71** | 12.02 | 11.90 | 12.08 | 11.08 | 12.12 |
| ResNet-101 | 12.65 | 10.72 | 10.79 | 10.65 | 10.51 | 10.70 | 10.51 | 10.70 | 10.51 | 11.97 | 10.77 | **10.35** | 10.71 | 11.72 | 10.74 | 11.40 | 10.75 |
| EffNet-B7 | 11.28 | 9.59 | 9.69 | 9.71 | 9.55 | 9.72 | 9.51 | 9.72 | 9.50 | 11.02 | 9.70 | **9.41** | 9.60 | 10.71 | 9.62 | 10.35 | 9.65 |
| EffNetV2-S | 12.39 | 9.40 | 9.45 | 9.66 | 9.48 | 9.71 | 9.50 | 9.71 | 9.50 | 10.61 | 9.55 | **9.37** | 9.43 | 10.64 | 9.46 | 10.37 | 9.50 |
| EffNetV2-M | 16.05 | 9.72 | 9.83 | 9.59 | 9.44 | 9.55 | 9.32 | 9.54 | 9.31 | 10.64 | 9.78 | **9.18** | 9.72 | 10.40 | 9.76 | 10.02 | 9.78 |
| EffNetV2-L | 9.80 | 8.99 | 9.08 | 8.90 | 8.82 | 8.93 | 8.83 | 8.93 | 8.84 | 10.28 | 9.01 | **8.81** | 9.00 | 9.96 | 9.01 | 9.70 | 9.03 |
| ConvNeXt-T | 14.02 | 10.87 | 10.96 | 10.39 | 10.33 | 10.40 | 10.33 | 10.40 | 10.32 | 11.79 | 10.89 | **10.13** | 10.87 | 11.61 | 10.90 | 11.14 | 10.91 |
| ConvNeXt-S | 13.62 | 10.30 | 10.35 | 10.16 | 10.02 | 10.16 | 9.95 | 10.15 | 9.95 | 11.32 | 10.36 | **9.78** | 10.32 | 11.27 | 10.35 | 10.83 | 10.37 |
| ConvNeXt-B | 13.86 | 10.14 | 10.20 | 10.05 | 9.89 | 10.01 | 9.79 | 10.01 | 9.79 | 11.36 | 10.20 | **9.60** | 10.14 | 11.11 | 10.18 | 10.63 | 10.19 |
| ConvNeXt-L | 11.58 | 9.88 | 9.99 | 9.92 | 9.76 | 9.99 | 9.70 | 9.99 | 9.71 | 10.97 | 9.92 | **9.46** | 9.88 | 10.96 | 9.90 | 10.46 | 9.94 |
| ViT-B/32 | 12.68 | 12.22 | 12.26 | 12.35 | 12.17 | 12.53 | 12.33 | 12.54 | 12.32 | 13.28 | 12.26 | **12.11** | 12.24 | 13.72 | 12.26 | 13.40 | 12.30 |
| ViT-B/16 | 11.02 | 10.65 | 10.71 | 10.89 | 10.72 | 11.01 | 10.84 | 11.01 | 10.83 | 11.98 | 10.73 | **10.59** | 10.66 | 12.09 | 10.67 | 11.84 | 10.72 |
| ViT-L/32 | 12.15 | **11.92** | 12.02 | 12.35 | 12.11 | 12.49 | 12.21 | 12.49 | 12.21 | 13.20 | 12.02 | **11.92** | 11.93 | 13.87 | 11.96 | 13.45 | 12.04 |
| ViT-L/16 | 11.39 | 11.13 | 11.22 | 11.73 | 11.46 | 11.88 | 11.57 | 11.88 | 11.57 | 12.82 | 11.28 | 11.13 | **11.11** | 13.21 | 11.15 | 12.71 | 11.22 |
| ViT-H/14 | **7.46** | 7.47 | 7.51 | 7.49 | **7.46** | 7.57 | 7.59 | 7.57 | 7.59 | 8.79 | **7.46** | 7.58 | 7.47 | 8.60 | 7.48 | 8.46 | 7.52 |
| Swin-T | 11.09 | 10.59 | 10.64 | 10.64 | 10.53 | 10.71 | 10.64 | 10.71 | 10.63 | 11.68 | 10.64 | **10.51** | 10.61 | 11.89 | 10.63 | 11.66 | 10.66 |
| Swin-S | 10.14 | 9.98 | 10.04 | 10.23 | 10.06 | 10.32 | 10.12 | 10.32 | 10.13 | 11.40 | 10.06 | **9.95** | 9.98 | 11.48 | 10.00 | 11.14 | 10.06 |
| Swin-B | 10.18 | 9.90 | 10.00 | 10.21 | 10.05 | 10.22 | 10.10 | 10.23 | 10.10 | 11.42 | 10.01 | **9.82** | 9.91 | 11.46 | 9.93 | 11.06 | 10.00 |
| SwinV2-T | 11.06 | 10.33 | 10.39 | 10.45 | 10.29 | 10.54 | 10.38 | 10.54 | 10.38 | 11.65 | 10.38 | **10.25** | 10.34 | 11.69 | 10.36 | 11.43 | 10.37 |
| SwinV2-S | 10.05 | 9.61 | 9.67 | 9.91 | 9.72 | 10.01 | 9.78 | 10.01 | 9.77 | 11.13 | 9.70 | **9.57** | 9.62 | 11.13 | 9.64 | 10.70 | 9.69 |
| SwinV2-B | 10.00 | 9.64 | 9.70 | 9.97 | 9.79 | 10.04 | 9.82 | 10.03 | 9.83 | 11.05 | 9.76 | **9.59** | 9.64 | 11.21 | 9.66 | 10.78 | 9.73 |
| CLIP (ViT-B/32) | 17.76 | 17.75 | 17.72 | 17.75 | 17.76 | 17.05 | **16.37** | 31.17 | 50.78 | 17.52 | 17.74 | 17.24 | 17.76 | 18.05 | 17.79 | 17.38 | 17.81 |
| CLIP (ViT-B/16) | 16.99 | 16.99 | 16.89 | 16.98 | 16.98 | 16.12 | **15.54** | 30.53 | 44.73 | 16.74 | 16.97 | 16.22 | 16.98 | 17.14 | 16.99 | 16.54 | 17.03 |
| CLIP (ViT-L/14) | 14.96 | 14.91 | 14.93 | 14.97 | 14.99 | 14.17 | **13.67** | 26.10 | 47.35 | 14.95 | 14.91 | 14.32 | 14.92 | 15.39 | 14.93 | 14.87 | 14.99 |

(d) ImageNet-21K

| Model | Uncal. | IRM | I-Max | TS | TS$_\text{TvA}$ | VS | VS$_\text{reg\_TvA}$ | DC | DC$_\text{reg\_TvA}$ | Beta | Beta$_\text{TvA}$ | Iso | Iso$_\text{TvA}$ | BBQ | BBQ$_\text{TvA}$ | HB | HB$_\text{TvA}$ |
|---|---|---|---|---|---|---|---|---|---|---|---|---|---|---|---|---|---|
| MN3 | 14.08 | err. | err. | 13.35 | 12.71 | 17.17 | 16.26 | 49.94 | 69.66 | err. | 11.95 | 17.06 | 11.93 | err. | 11.93 | **10.70** | 11.95 |
| ViT-B/16 | 13.51 | err. | err. | 13.54 | 13.49 | 18.14 | 17.11 | 20.10 | 18.22 | err. | 12.96 | 18.12 | **12.76** | err. | 12.77 | 12.91 | 12.96 |

(e) Amazon Fine Foods

| Model | Uncal. | IRM | I-Max | TS | TS$_\text{TvA}$ | VS | VS$_\text{reg\_TvA}$ | DC | DC$_\text{reg\_TvA}$ | Beta | Beta$_\text{TvA}$ | Iso | Iso$_\text{TvA}$ | BBQ | BBQ$_\text{TvA}$ | HB | HB$_\text{TvA}$ |
|---|---|---|---|---|---|---|---|---|---|---|---|---|---|---|---|---|---|
| T5 | 7.78 | 7.27 | 7.30 | 7.27 | 7.27 | 6.93 | 6.85 | 6.73 | 6.75 | 7.32 | 7.30 | **6.69** | 7.28 | 6.77 | 7.29 | 7.19 | 7.33 |
| T5-large | 7.02 | 6.34 | 6.36 | 6.40 | 6.40 | 6.11 | 5.94 | **5.76** | 5.82 | 6.57 | 6.46 | 5.80 | 6.34 | 5.92 | 6.38 | 6.42 | 6.41 |
| RoBERTa | 8.66 | 7.30 | 7.35 | 7.42 | 7.42 | 7.22 | 7.02 | 6.87 | 6.99 | 8.12 | 7.85 | **6.86** | 7.30 | 7.11 | 7.48 | 7.70 | 7.37 |
| RoBERTa-large | 7.43 | 6.15 | 6.23 | 6.30 | 6.29 | 6.18 | 6.01 | 5.89 | 6.00 | 6.96 | 6.85 | **5.86** | 6.14 | 6.21 | 6.49 | 6.74 | 6.22 |

(f) DynaSent

| Model | Uncal. | IRM | I-Max | TS | TS$_\text{TvA}$ | VS | VS$_\text{reg\_TvA}$ | DC | DC$_\text{reg\_TvA}$ | Beta | Beta$_\text{TvA}$ | Iso | Iso$_\text{TvA}$ | BBQ | BBQ$_\text{TvA}$ | HB | HB$_\text{TvA}$ |
|---|---|---|---|---|---|---|---|---|---|---|---|---|---|---|---|---|---|
| T5 | 14.68 | 13.96 | 13.96 | 13.89 | 13.89 | 14.15 | 13.88 | 14.09 | 13.87 | 15.40 | 14.02 | **13.84** | 13.98 | 14.00 | 14.05 | 14.21 | 13.96 |
| T5-large | 13.67 | 12.48 | 12.52 | 12.52 | 12.52 | 13.08 | 12.21 | 12.82 | 12.16 | 14.16 | 12.80 | **12.14** | 12.50 | 12.43 | 12.62 | 12.78 | 12.54 |
| RoBERTa | 18.98 | 14.96 | 15.09 | 17.03 | 17.03 | 18.16 | 15.53 | 17.66 | 15.15 | 19.39 | 16.62 | **14.67** | 14.97 | 15.60 | 15.64 | 15.96 | 15.01 |
| RoBERTa-large | 16.14 | 13.21 | 13.35 | 14.65 | 14.65 | 15.54 | 13.51 | 15.29 | 13.37 | 16.66 | 14.70 | **13.09** | 13.20 | 13.91 | 13.83 | 14.20 | 13.25 |

(g) MNLI

| Model | Uncal. | IRM | I-Max | TS | TS$_\text{TvA}$ | VS | VS$_\text{reg\_TvA}$ | DC | DC$_\text{reg\_TvA}$ | Beta | Beta$_\text{TvA}$ | Iso | Iso$_\text{TvA}$ | BBQ | BBQ$_\text{TvA}$ | HB | HB$_\text{TvA}$ |
|---|---|---|---|---|---|---|---|---|---|---|---|---|---|---|---|---|---|
| T5 | 8.95 | 8.21 | 8.25 | 8.21 | 8.21 | 8.35 | 8.19 | 8.29 | 8.17 | 9.33 | 8.28 | **8.14** | 8.22 | 8.28 | 8.29 | 8.65 | 8.24 |
| T5-large | 8.56 | 7.49 | 7.52 | 7.76 | 7.76 | 7.98 | 7.46 | 7.88 | 7.41 | 8.80 | 7.98 | **7.40** | 7.49 | 7.80 | 7.78 | 8.07 | 7.54 |
| RoBERTa | 11.50 | 9.60 | 9.67 | 10.24 | 10.23 | 10.57 | 9.57 | 10.36 | 9.54 | 11.80 | 10.48 | **9.44** | 9.60 | 10.06 | 10.03 | 10.48 | 9.60 |
| RoBERTa-large | 9.30 | 8.04 | 8.12 | 8.40 | 8.40 | 8.65 | 8.04 | 8.56 | 8.03 | 9.57 | 8.72 | **7.98** | 8.05 | 8.40 | 8.39 | 8.71 | 8.12 |

(h) Yahoo Answers

| Model | Uncal. | IRM | I-Max | TS | TS$_\text{TvA}$ | VS | VS$_\text{reg\_TvA}$ | DC | DC$_\text{reg\_TvA}$ | Beta | Beta$_\text{TvA}$ | Iso | Iso$_\text{TvA}$ | BBQ | BBQ$_\text{TvA}$ | HB | HB$_\text{TvA}$ |
|---|---|---|---|---|---|---|---|---|---|---|---|---|---|---|---|---|---|
| T5 | 14.70 | 14.18 | 14.21 | 14.14 | 14.15 | 14.22 | 14.14 | 14.21 | **14.13** | 14.98 | 14.20 | 14.14 | 14.18 | 14.41 | 14.24 | 14.75 | 14.24 |
| T5-large | 15.23 | 14.25 | 14.29 | 14.23 | 14.25 | 14.44 | 14.21 | 14.41 | **14.12** | 15.66 | 14.28 | 14.18 | 14.24 | 14.49 | 14.33 | 14.83 | 14.29 |
| RoBERTa | 20.95 | 15.94 | 16.12 | 17.57 | 17.56 | 19.20 | 15.92 | 18.84 | **15.54** | 21.23 | 17.15 | 15.64 | 15.94 | 16.75 | 16.55 | 17.20 | 15.99 |
| RoBERTa-large | 20.94 | 15.68 | 15.86 | 17.57 | 17.57 | 19.23 | 15.59 | 18.96 | **15.30** | 21.21 | 17.14 | 15.38 | 15.68 | 16.71 | 16.42 | 17.15 | 15.72 |

Table 12: Calibration methods applied to in-context learning of LLMs. Accuracy and ECE are in %.

| Model | Shots | Dataset Method | TREC Acc. (↑) | TREC ECE (↓) | SST-5 Acc. (↑) | SST-5 ECE (↓) | DBpedia Acc. (↑) | DBpedia ECE (↓) |
|---|---|---|---|---|---|---|---|---|
| GPT-J 6B | 0 | Uncalibrated | 24.7 | 29.7 | 33.7 | 22.5 | 19.7 | 27.4 |
|  |  | ConC | 40.0 | 14.0 | 40.7 | 10.3 | 47.7 | 24.6 |
|  |  | LinC | **58.9** | 26.4 | **46.3** | 11.0 | **62.2** | 12.8 |
|  |  | LinC+HB$_{\text{TvA}}$ | **58.9** | **6.5** | **46.3** | **7.0** | **62.2** | **5.7** |
|  | 1 | Uncalibrated | 43.7 | 12.1 | 36.3 | 30.9 | 58.7 | 14.2 |
|  |  | ConC | 41.7 | 13.6 | **50.7** | 14.2 | 82.7 | 6.9 |
|  |  | LinC | **59.9** | 9.1 | 50.1 | 12.3 | **84.4** | 6.6 |
|  |  | LinC+HB$_{\text{TvA}}$ | **59.9** | **3.9** | 50.1 | **7.3** | **84.4** | **5.1** |
|  | 4 | Uncalibrated | 26.0 | 41.6 | 51.3 | 28.2 | 89.0 | 15.7 |
|  |  | ConC | 40.3 | 14.4 | **54.3** | 8.8 | 94.0 | 6.9 |
|  |  | LinC | **57.9** | 9.7 | 53.6 | 10.6 | **94.3** | 5.7 |
|  |  | LinC+HB$_{\text{TvA}}$ | **57.9** | **5.2** | 53.6 | **7.1** | **94.3** | **4.8** |
|  | 8 | Uncalibrated | 36.0 | 26.0 | 48.3 | 9.7 | 92.3 | 9.2 |
|  |  | ConC | 46.7 | 15.5 | 43.7 | 11.7 | 92.0 | 6.8 |
|  |  | LinC | **60.7** | **6.3** | **51.7** | 9.5 | **93.9** | 5.8 |
|  |  | LinC+HB$_{\text{TvA}}$ | **60.7** | 6.6 | **51.7** | **7.6** | **93.9** | **2.6** |
| Llama-2 13B | 0 | Uncalibrated | 48.7 | 21.4 | 34.0 | 17.6 | 54.3 | 19.7 |
|  |  | ConC | 71.7 | 18.7 | 33.3 | 17.2 | 75.3 | 17.2 |
|  |  | LinC | **73.3** | 11.4 | **47.6** | 11.3 | **84.4** | 16.2 |
|  |  | LinC+HB$_{\text{TvA}}$ | **73.3** | **9.3** | **47.6** | **6.7** | **84.4** | **4.1** |
|  | 1 | Uncalibrated | 63.0 | 8.6 | 41.3 | 29.4 | 90.7 | 11.4 |
|  |  | ConC | 76.0 | **5.9** | 41.0 | 12.6 | 92.3 | 5.2 |
|  |  | LinC | **79.7** | 6.3 | **48.7** | 12.1 | **93.1** | 4.4 |
|  |  | LinC+HB$_{\text{TvA}}$ | **79.7** | 6.0 | **48.7** | 9.6 | **93.1** | **3.0** |
|  | 4 | Uncalibrated | 60.0 | 12.0 | 50.7 | 37.6 | 94.0 | 9.9 |
|  |  | ConC | 71.3 | 6.9 | 51.3 | 18.6 | **95.3** | 3.8 |
|  |  | LinC | **75.6** | 8.0 | **52.9** | 15.1 | **95.3** | 3.8 |
|  |  | LinC+HB$_{\text{TvA}}$ | **75.6** | **4.0** | **52.9** | **7.6** | **95.3** | **2.2** |
|  | 8 | Uncalibrated | 70.0 | **5.2** | **55.0** | 7.0 | 94.7 | 5.9 |
|  |  | ConC | **73.7** | 12.6 | 44.0 | 22.8 | 94.3 | 3.9 |
|  |  | LinC | 73.5 | 9.4 | 50.2 | 14.4 | **95.3** | 3.9 |
|  |  | LinC+HB$_{\text{TvA}}$ | 73.5 | 7.0 | 50.2 | **4.2** | **95.3** | **2.1** |

Large Language Models (LLMs) exhibit an in-context learning (ICL) capability, meaning they can learn from just a few examples in the context. It works by constructing a prompt that includes input-output pairs demonstrating the considered task, followed by a query for a new input. See [8] for a survey. Recent works develop calibration methods whose main goal is to improve the performance of ICL for LLMs, without requiring a complicated model fine-tuning. [77] uses a customized variant of Platt scaling (more specifically, Vector Scaling). Their method infers good values of the vector scaling parameters in a data-free procedure. The idea is that for a "content-free" input, e.g., "N/A", the calibrated probability has a 50% chance (for a binary classification task) of removing a bias toward the positive or negative class. In our paper, we denote this method as *ConC*. [1] builds on top of this work but uses a calibration set to learn the scaling parameters by minimizing the cross-entropy loss. This can be considered as Matrix Scaling. We denote this method as *LinC*. [78] proposes a per-class normalization of the probabilities on a given batch. [25] estimates the in-context model label marginal $p(y)$ from limited data and uses it to calibrate the model probabilities. Paper [20] uses a Gaussian mixture model.

In our experiments, we have tested a two-step calibration. First, we use the state-of-the-art method LinC to maximize the accuracy by learning scaling parameters on a calibration set. Then, we apply HB$_{\text{TvA}}$ to scale the confidences to lower the calibration error ECE, while preserving the accuracy gains. We use the same calibration set for the two methods. LinC performance depends on hyperparameter values, but to keep the experiments simple, we fixed the following values: 100 epochs, a learning rate of 0.001, and 300 calibration samples. It means that the reported performance of LinC is suboptimal and could be enhanced even more. We used the same experimental setting as [1]. We used the models GPT-J with 6B parameters [62] and Llama-2 with 13B parameters [59]. The text classification datasets are TREC [61] for question classification with 6 classes, SST-5 [55] for sentiment analysis with 5 classes, and DBpedia [74] for topic classification with 14 classes. The 0-shot, 1-shot, 4-shot, and 8-shot learning settings were tested. Five different sets of 300 test samples were randomly selected, and results are averaged over 5 seeds. We evaluated the accuracy and ECE for each configuration. Please see Table 12 for the results. In most cases, LinC+HB$_{\text{TvA}}$ achieves the best accuracy and ECE.

