# OpenReview forum: "Confidence Calibration of Classifiers with Many Classes"
_NeurIPS.cc/2024/Conference — NeurIPS 2024 poster_

### Official Review · Reviewer_QaKm · 2024-06-13

**Soundness:** 2
**Presentation:** 2
**Contribution:** 2
**Rating:** 4
**Confidence:** 4

**Summary:**

The paper proposes a confidence calibration method.  It reformulates a multi-class problem as the binary task "is the prediction correct?".  Then a given calibration method  (e.g. Temperature Scaling) is applied to that binary task.

**Strengths:**

The proposed method is very simple and can be efficiently applied to tasks with a large number of classes,

**Weaknesses:**

The main justification of a calibration model is its improved results (usually measured by ECE).
In the case of TS the paper reports a minor improvement. However,  in the case of HB, the paper reports a huge calibration improvement.  It is not clear what is the reason for the improvement difference.

As far as I understand the HB method, we first split the validation samples  to bins according to the confidence.


Although the proposed idea is very very simple,  the  method presentation is overcomplicated and there is  an over-seling of the method.

**Questions:**

ggg

**Limitations:**

ggg

---

> ### Author Rebuttal · Authors · 2024-08-05
>
> Thank you for the review. We are glad that you find our approach simple and efficient. Please find our rebuttal below.
>
> **Weaknesses**
>
> > The main justification of a calibration model is its improved results (usually measured by ECE).
> In the case of TS the paper reports a minor improvement. However, in the case of HB, the paper reports a huge calibration improvement. It is not clear what is the reason for the improvement difference.
>
> The reason that improvements for HB are more significant than for TS is simply that standard HB (and other binary methods) does not perform well when the number of classes is high. This is what we describe as "Issue 3" in the paper. For experimental results, you can see in Table 1 that improvements for HB are moderate for C10 (10 classes), higher for C100 (100 classes), even higher for IN (1k classes), and even higher for IN21K (10k classes). Concerning TS, TvA solves "Issue 1" (the inefficientness of the loss) which is not as serious as "Issue 3".
> We do not see how this different behavior is a weakness, but we will include the explanation in the final version.
>
>
> > As far as I understand the HB method, we first split the validation samples to bins according to the confidence.
>
> This is correct but we do not really understand how this describes a weakness. Could you please clarify this point?
>
> > Although the proposed idea is very very simple, the method presentation is overcomplicated and there is an over-seling of the method.
>
> We agree that the idea is simple, this is what we consider to be one of its main strengths. Easy and quick to implement, it can improve existing calibration methods. We chose to explain the limitations of existing approaches before explaining how our approach applies to scaling and binary calibration methods. Algorithm 1, and Algorithms 2 and 3 of the Appendix provide, in our opinion, simple presentations of the approach. Also, other Reviewers praise the paper's clarity: "writing is clear, and easy to understand" from Reviewer S8uZ; "well-written and pleasant to read" from Reviewer eq8Z; "nicely presents" from Reviewer yKjq. As for the "over-seling", could you be more specific?

---

> > ### Comment · Reviewer_QaKm · 2024-08-10
> >
> > TS is the most standard calibration method and the fact that applying the proposed method to TS does not result in a clear improvement is a weakness. Hence, my score remains 4.

---

> > > ### Author Response · Authors · 2024-08-10
> > > **Answer to Reviewer QaKm**
> > >
> > > Thank you for clarifying your point. For your curiosity, here is what we believe to be the reason. TS is already data efficient as only a single parameter is learned. It only needs a few calibration data to converge to its optimal value but does not benefit from having more data (which is seen in Figure 3 of the paper). By applying our approach to TS, we improve its performance. Our approach changes how calibration methods are applied but does not change the methods themselves. The performance of TS is still limited by the fact that only a single parameter is learned.

---

### Official Review · Reviewer_oe96 · 2024-07-06

**Soundness:** 1
**Presentation:** 1
**Contribution:** 1
**Rating:** 3
**Confidence:** 5

**Summary:**

The paper addresses the issue of miscalibrated confidence scores in neural network classifiers, particularly for problems with many classes. Traditional methods often fail in these scenarios. The authors propose transforming the multiclass calibration problem into a single binary classification problem, termed Top-versus-All (TvA), allowing the use of standard calibration methods more efficiently. This approach improves the performance of calibration methods like Temperature Scaling and Vector Scaling by better utilizing the calibration data and reducing overfitting. Experiments on various datasets and models demonstrate the scalability and effectiveness of this method.

**Strengths:**

Simple Method: The Top-versus-All (TvA) approach is straightforward and easy to implement, requiring minimal changes to existing calibration methods.

Good Performance: The TvA method significantly enhances the performance of standard calibration techniques, consistently improving Expected Calibration Error (ECE) across various datasets and models.

Scalability: The method scales well to handle classifiers with a large number of classes, addressing a common limitation in traditional calibration methods.

**Weaknesses:**

1. Lack of novelty. I read a paper very similar to the idea previously, make top vs all, but I am sorry that I cannot find the link. And as I remember, the Multiclass to Binary idea has been used in this literature multiple times.

2. Lack of the comparison with some SOTA post hoc methods such as [1] , the performance may not succeed.

3. Lack of comparison with some SOTA train time methods, such as [2].

4. Overall, the method lack the analysis and comparison with other SOTA methods.

[1] Proximity-Informed Calibration for Deep Neural Networks
[2] Dual focal loss for calibration

**Questions:**

see Weaknesses

**Limitations:**

see Weaknesses

---

> ### Author Rebuttal · Authors · 2024-08-05
>
> Thank you for the review. We appreciate that you find that our method is simple, scalable, and has good performance. We hope our response can address your concerns.
>
> **Weaknesses**
>
> > 1. Lack of novelty. I read a paper very similar to the idea previously, make top vs all, but I am sorry that I cannot find the link. And as I remember, the Multiclass to Binary idea has been used in this literature multiple times.
>
> We are sorry we were not able to find such a reference. It would be helpful if you could provide it to us. Also, the Related Work includes a paragraph on Multiclass to Binary, and Appendix C includes a comparison with competing approaches. We thus already discussed in the paper how our approach differs from other work.
>
> > 2. Lack of the comparison with some SOTA post hoc methods such as [1] , the performance may not succeed.
>
> Indeed, we have missed this nice reference; thank you for pointing it out. We have conducted preliminary experiments following the experimental setting of paper [1]'s Table 4. Results are available in the PDF attached to the global response. They show that our approach performs better. We plan to add more comparisons with their method in the final version of our paper.
>
>
> > 3. Lack of comparison with some SOTA train time methods, such as [2].
>
> As we have stated in a paragraph starting line 90, we focus on calibrating already-trained models. Train time methods require a high development time, often compromise accuracy, and are not adapted to pre-trained models. Because of the computation time required by train-time methods, paper [2]'s experiments are much more limited than ours. Their biggest dataset is Tiny ImageNet, a subset of downscaled images from ImageNet. In our paper, we not only consider ImageNet but also the even more complex ImageNet-21K dataset. We have compared 26 models for ImageNet, each one calibrated with 5 different seeds, while [2] only studies a single model for ImageNet. A fair comparison with [2] would require huge computing resources. To us, fast development is one of the main strengths of post-hoc calibration methods, the main scope of the paper.
>
> Furthermore, many train time methods actually benefit from post-hoc methods, as can be seen in Table 1 of [2]. This means that our work could also be applied on top of train time methods to further improve calibration.
>
> We will add the reference to [2] in the "Training calibrated networks" paragraph of the Related work for the final version.
>
>
> > 4. Overall, the method lack the analysis and comparison with other SOTA methods.
>
> We already have included the SOTA methods IRM and I-Max, and besides the preliminary results in the attached PDF, we plan to add more comparisons with [1] in the final version, as you suggested.

---

> > ### Comment · Reviewer_oe96 · 2024-08-10
> >
> > I have read the comments from other reviewers, the author has addressed my concern on performance by providing the comparison with other works. But I think the novelty is still trival. I will raise my score to Reject instead of strong reject.

---

> > > ### Author Response · Authors · 2024-08-10
> > > **Answer to Reviewer oe96**
> > >
> > > We are glad that we successfully addressed your concern about performance. However, we regret you find the novelty trivial. The main intuition is indeed trivial, but we believe our approach's motivation and justification are clear. Also, we consider this triviality as a strength because it allows the approach to be straightforward and easy to implement, as you mentioned.
> > >
> > > We are sincerely grateful that you engaged in the discussion and raised your score.

---

### Official Review · Reviewer_yKjq · 2024-07-08

**Soundness:** 3
**Presentation:** 3
**Contribution:** 3
**Rating:** 6
**Confidence:** 4

**Summary:**

This paper presents a top-versus-all (TvA) approach for the confidence calibration of classifiers with many classes. The authors categorizes the post-processing calibration methods into two groups: scaling and binary methods, and lists several problems with these approaches. The authors reformulate the problem of calibrating a multiclass classifier into a single binary classifier, where the confidence score represents the maximum class probability of the binary problem. They show the effectiveness of the proposed method on numerous neural networks used for image or text classification.

**Strengths:**

1) This paper nicely presents the shortcomings of the existing methods and describes how the proposed method addresses these challenges, the proposed method is innovative and provides a simple and efficient solution to calibrating classifiers with many classes.
2) This paper presents extensive experimental results on multiple datasets and network architectures.
3) This paper provides a well-structured review of relevant literature and clearly explains the motivation and significance of the proposed approach.

**Weaknesses:**

1) The theoretical contribution of the TvA approach and its impact on calibration could be further strengthened.
2) The propose method may exhibit limited novelty, as it builds upon established approaches rather than introducing entirely novel elements.
2) The calibration of CLIP appears to be better than other models on ImageNet, can you explain why? Additionally, it seems like most calibration methods don't work on CLIP, which confuses me.

**Questions:**

Please see my comments in Weaknesses.

**Limitations:**

The authors have addressed the limitations in the main text.

---

> ### Author Rebuttal · Authors · 2024-08-05
>
> Thank you for the constructive feedback. We value that you praise our approach, its motivation, and significance and acknowledge the extensiveness of our experiments. We hope our response can address your concerns.
>
> **Weaknesses**
>
> > 1. The theoretical contribution of the TvA approach and its impact on calibration could be further strengthened.
>
> Please see our global answer which addresses this weakness.
>
> > 2. The propose method may exhibit limited novelty, as it builds upon established approaches rather than introducing entirely novel elements.
>
> The novelty of our approach is to calibrate a surrogate binary classifier. This is a new reformulation of the confidence calibration problem. It does not build upon established approaches but rather applies on top of existing methods such as HB.
>
> > 3. The calibration of CLIP appears to be better than other models on ImageNet, can you explain why? Additionally, it seems like most calibration methods don't work on CLIP, which confuses me.
>
> Indeed, CLIP has a different behavior from other models, which is consistent with the results of (Galil et al., 2023). There could be several reasons to explain the good "out of the box" calibration of CLIP and its particular behavior. The multimodal training regime and zero-shot adaptation as a classifier are quite different from standard image classifiers. CLIP's "logits" are based on the cosine similarity of images with textual prompts representing the classes: they are not like standard image classifier logits trained with a fixed number of classes. The pre-training dataset is also different from ImageNet (much larger and containing images with a textual legend, not the class label).
>
> The fact that many calibration methods do not improve CLIP's calibration is probably also due to the standard methods having difficulties improving already well-calibrated models. This also happens with ViT-H/14.
>
> We will clarify this in the final version.

---

> > ### Comment · Reviewer_yKjq · 2024-08-13
> >
> > Thanks for your response, I'm keeping my score in view of the limited novelty of the method.

---

### Official Review · Reviewer_eq8Z · 2024-07-09

**Soundness:** 3
**Presentation:** 4
**Contribution:** 3
**Rating:** 7
**Confidence:** 2

**Summary:**

The paper relates to post-hoc confidence calibration i.e. the calibration of
the top-prediction of a classifier such that ``when the class l is predicted
with confidence q, the probability of the actual class being l is also q''.
The calibration does not update the classifier.

The paper tackles two issues of past calibration methods:
- The one vs all approaches that that do not scale well with the number of
  classes (there are as many binary calibrators as number of classes)
- Each binary calibrator is tuned on datasets which imbalance increases with
  the number of classes: the positives are the instances of class $k$ and the
  negatives are all the other instances
- The overfitting of scaling methods due to their parametric nature (the number
  of parameter increases with the number of classes)

To do so, it proposes an alternative formulation of the calibration problem:
instead of calibrating the top-prediction directly, the paper calibrates an
intermediate binary classifier that estimates whether the top-prediction is
correct or not.

This addresses the two previous issues as:
- There is only one calibration optimization that calibrates the single binary
  classifier described above, which is independent of the number of classes
- The imbalance remains but is reduced and linked to the classifier's accuracy
- The calibration of the binary classifier with scaling methods requires less
  parameters and is hence less prone to overfitting

The proposed calibration can be optimized with off-the-shelf binary calibrators
and the experiments compare the proposed calibration against the traditional
calibrations, i.e., calibrating the classifier's top prediction directly.

The experiments compare the proposed formulation that reduce the ECE compared
to the traditional calibrations.
The proposed formulation usually reduce the ECE compared to the traditional
calibration and improves over IRM and I-MAX.
The improvement is more notable on datasets with a higher number of classes.

**Strengths:**

S1. The paper is well-written and pleasant to read.

S2. The proposed calibration formulation is simple yet efficient: it calibrates
   a single intermediate binary problem that estimates whether the classifier's
   prediction is correct. Thus, it can leverage the strengths of existing
   methods on a simpler optimization problem.

S3. By design, the proposed method preserves the classification accuracy since
   it does not edit the network.

S4. The experiments are exhaustive in terms of datasets and calibration methods:
- the experiments are run on image and text classification on multiple datasets
  with a wide range of number of classes: 10 on cifar-10 to 10K on ImageNet-21K.
- scaling methods (Temperature and Vector Scaling, Dirichlet Calibration) and
  binary methods (Isotonic regression, BBQ and Histogram Binning); and other
  methods tackling the one-vs-all limitations (IRM and I-MAX).

**Weaknesses:**

W1. Not really a weakness but a consequence of the design choice: the proposed method keeps the classifier untouched so that the calibration does not impact the accuracy.
However, improving the confidence of incorrect predictions could help improving the classification accuracy by re-distributing the probabilities.
In practice though (Tab.8), the accuracy varies only by a few points.

**Questions:**

Q1. One of the claimed advantage of the proposed method (TvA) is the "stronger
  gradient" of TvA compared to that of the default scaling methods (understand
  the gradient's magnitude is higher).
  However, it is not clear how this is an advantage on its own as i) the impact of the
  gradient's magnitude is mitigated by the choice of learning rate; ii) a
  higher gradient magnitude does not imply a better optimization (this is
  specific to each optimization problem).
  Can more details be given on why `stronger gradients' are an advantage?

**Limitations:**

- The advantage of 'preserving' the accuracy is limited by the fact that existing
  calibration method that affect the accuracy usually induce a variation of ~1%
  only, and can even affect positively the accuracy i.e. increase it.
  This leads to the question of whether 'preserving of the accuracy' is really a
  limitation for other methods.

- Issue 1 describes the limitation of using cross entropy loss during
  calibration as it increases the probability of the true class which only
  **indirectly** impacts the probability of the top-prediction i.e. the
  confidence.
  L: 173 describes this optimization as 'inefficient' compared to the proposed
  one, however it is hard to quantify the 'efficiency' of an optimization so a
  better word might be 'indirect'.

Misc.:
- Eq6: v is undefined, does it refer to the logit probability vector?
- lack of legends for the blue and red bars (which is accuracy, which is
  confidence) even though one can understand
- Legend for Global acc. and avg. confidence not very visible

---

> ### Author Rebuttal · Authors · 2024-08-05
>
> Thank you for spending the time and effort to write this thorough review, which demonstrates a deep understanding of our work. We are grateful that you praise our writing and recognize our approach's efficiency and the exhaustivity of our experiments. The points you raised are relevant, and we will update our paper accordingly. We believe your confidence score could be higher. Please see our responses below.
>
> **Weaknesses**
>
> > W1. Not really a weakness but a consequence of the design choice: the proposed method keeps the classifier untouched so that the calibration does not impact the accuracy. However, improving the confidence of incorrect predictions could help improving the classification accuracy by re-distributing the probabilities. In practice though (Tab.8), the accuracy varies only by a few points.
>
> Indeed, TvA applied to binary methods keeps the classifier accuracy untouched which means that it cannot improve accuracy. However, we consider that an advantage because in practice binary methods otherwise usually degrade accuracy (Table 8).
>
>
> **Questions**
>
> > Q1. One of the claimed advantage of the proposed method (TvA) is the "stronger gradient" of TvA compared to that of the default scaling methods (understand the gradient's magnitude is higher). However, it is not clear how this is an advantage on its own as i) the impact of the gradient's magnitude is mitigated by the choice of learning rate; ii) a higher gradient magnitude does not imply a better optimization (this is specific to each optimization problem). Can more details be given on why `stronger gradients' are an advantage?
>
> The reasoning about "stronger gradient" was inspired by two articles: [1] and [2] (see page 5 for both).
>
> More specifically in our case, when predictions are incorrect, the gradient with TvA is proportional to $\frac{s}{s-1}$ with $s$ the confidence. This term means that as the confidence for wrong predictions gets higher, so does the gradient to reduce the confidence. This effect is not mitigated by the choice of learning rate (which does not vary with $s$) and we believe it allows a better optimization. We will clarify this point in the final version.
>
> [1] *Zhao, Z., Liu, Z., & Larson, M. (2021). On Success and Simplicity: A Second Look at Transferable Targeted Attacks. Advances in Neural Information Processing Systems (Vol. 34, pp. 6115-6128)*
>
> [2] *Naseer, M. M., Khan, S. H., Khan, M. H., Shahbaz Khan, F., & Porikli, F. (2019). Cross-Domain Transferability of Adversarial Perturbations. Advances in Neural Information Processing Systems (Vol. 32)*
>
> **Limitations**
>
> > The advantage of 'preserving' the accuracy is limited by the fact that existing calibration method that affect the accuracy usually induce a variation of ~1% only, and can even affect positively the accuracy i.e. increase it. This leads to the question of whether 'preserving of the accuracy' is really a limitation for other methods.
>
> Given the effort required to gain 1% accuracy on ImageNet, losing it during calibration is not negligible in our opinion. In principle, we believe that decoupling accuracy optimization (during model training) and calibration (during post-hoc calibration) is a more efficient approach than train time calibration (optimizing both accuracy and calibration during model training) or having calibration methods impacting the accuracy. Indeed, it avoids to manage compromises between two objectives during development. Preserving accuracy is a characteristic of our approach, which we consider an advantage in most cases, but which other people might view differently.
>
> > Issue 1 describes the limitation of using cross entropy loss during calibration as it increases the probability of the true class which only indirectly impacts the probability of the top-prediction i.e. the confidence. L: 173 describes this optimization as 'inefficient' compared to the proposed one, however it is hard to quantify the 'efficiency' of an optimization so a better word might be 'indirect'.
>
> Corrected, thank you.
>
> > Misc.:
>
> $v$ is defined line 238, this is the parameters vector that scales the logits values. While Temperature Scaling uses a single coefficient to scale the logits, Vector Scaling scales all class logits independantly (1 coefficient per class).
>
> We added the signification of the blue and red bars and improved the legend for global accuracy and average confidence.

---

> > ### Comment · Reviewer_eq8Z · 2024-08-12
> >
> > The rebuttal addressed all the comments in the review, thank you.
> >
> > The comment on the confidence of the review is noted but the score will remain as is as it reflects that I am not an expert in the area so it is hard to assess the contribution against previous works (except for what is described in the related work of course).

---

### Official Review · Reviewer_RhL1 · 2024-07-11

**Soundness:** 3
**Presentation:** 3
**Contribution:** 3
**Rating:** 6
**Confidence:** 4

**Summary:**

The paper proposes a method to improve confidence calibration in neural network-based classification models. It transforms the multiclass calibration problem into a binary classification surrogate, demonstrating enhanced performance across various neural network applications in image and text classification.

**Strengths:**

1. The paper's discussion on the standard approach to confidence calibration provides a comprehensive overview of existing methods and their limitations, highlighting the need for more effective strategies.

2. The Top-versus-All approach presented in the paper offers a straightforward yet powerful solution to the multiclass calibration challenge. By transforming the problem into a binary classification task, it simplifies the calibration process while maintaining effectiveness in improving prediction confidence.

3. The scalability and generality of the proposed method to LLMs are particularly noteworthy. By demonstrating its applicability across diverse neural network architectures used in image and text classification tasks, the paper underscores its potential impact on enhancing calibration methods for a wide range of applications.

**Weaknesses:**

1. The paper lacks rigorous theoretical evidence to substantiate its claims, which weakens the strength of the proposed solutions. Without detailed proofs, the effectiveness of the Top-versus-All approach and other methods in improving confidence calibration remains somewhat speculative.

2. While the paper claims scalability to LLMs, the experimental evaluation primarily focuses on smaller-scale models. This limited scope raises questions about the method's performance and applicability to state-of-the-art LLMs like LLaMA or Gemma. Extending experiments to these latest models would provide more convincing evidence of scalability and effectiveness across a broader range of neural architectures.

3. Figure 3 in the paper indicates a notable variance in the ECE test results. This variability suggests inconsistent performance across different settings or datasets, potentially indicating limitations in the method's stability and reliability under varying conditions.

**Questions:**

1. how does the method apply to CLIP? On the contrastive softmax loss? What if the number of classes is varying during training CLIP? I.e., the batch size is varying.

**Limitations:**

Limitations are well discussed in the paper

---

> ### Author Rebuttal · Authors · 2024-08-05
>
> Thank you for reviewing our paper. We appreciate that you value our discussion on the limitations of existing approaches, the straightforwardness of our approach, and its scalability and generality. We have addressed each of your concerns below.
>
> **Weaknesses**
>
> > 1. The paper lacks rigorous theoretical evidence to substantiate its claims, which weakens the strength of the proposed solutions. Without detailed proofs, the effectiveness of the Top-versus-All approach and other methods in improving confidence calibration remains somewhat speculative.
>
> Please see our global answer which addresses this weakness.
>
> > 2. While the paper claims scalability to LLMs, the experimental evaluation primarily focuses on smaller-scale models. This limited scope raises questions about the method's performance and applicability to state-of-the-art LLMs like LLaMA or Gemma. Extending experiments to these latest models would provide more convincing evidence of scalability and effectiveness across a broader range of neural architectures.
>
> This might be a misunderstanding. Experiments in Table 1 show results with T5 and RoBERTa, which we call Pre-trained Language Models, and which we believe you call smaller-scale models. What we call LLMs are GPT-J 6B and LLaMA-2 13B (as you suggested), whose results are included in the Appendix, Table 11. This is mentioned in the Experiments section, lines 318-322. Does that resolve the issue?
>
> > 3. Figure 3 in the paper indicates a notable variance in the ECE test results. This variability suggests inconsistent performance across different settings or datasets, potentially indicating limitations in the method's stability and reliability under varying conditions.
>
> There is indeed some variance for scaling methods (figure at the bottom). We believe this is because these methods involve a learning process, which might not be perfect each time as we did not optimize each run due to the scale of our experiments. However, this variability is not increased when our TvA approach is applied: it comes from the underlying methods.
> Also, please note that for binary methods (figure at the top), especially when TvA is applied, the variance is very small.
> Additionally, Table 6 of the Appendix includes the standard deviations for ECE. In most cases, compared to the underlying methods, applying our approach TvA either does not impact the variance or reduces it.
>
> **Questions**
>
> > 1. how does the method apply to CLIP? On the contrastive softmax loss? What if the number of classes is varying during training CLIP? I.e., the batch size is varying.
>
> In this paper, we use a pre-trained CLIP model as a zero-shot classifier with the standard method, without retraining. We compute the cosine similarities between the images and the text "a photo of a {c}" with c taking the values of all the labels for the considered task, e.g., 1000 different class names for ImageNet. The cosine similarities are converted to "logits" by multiplying them by a coefficient.
> For more details, you can check the provided code full_code/utils.py, lines 137-171, where the CLIPClassifier class is defined. We hope we answered your question and will clarify this point in the final version.

---

### Official Review · Reviewer_S8uZ · 2024-07-14

**Soundness:** 1
**Presentation:** 2
**Contribution:** 2
**Rating:** 3
**Confidence:** 4

**Summary:**

This paper proposes a new learning objective for model calibration in multi-class classification tasks. The authors analyze the issues with the current softmax-based scaling method and argue that it only considers the confidence (the probability of the predicted class) without accounting for other remaining classes during calibration. The proposed Top-versus-All method divides the calibration data into class-wise datasets. Experiments on various models, tasks, and calibration baseline methods have shown some effectiveness of the proposed approach.

**Strengths:**

1. The experiment is conducted on many models and test sets.

2. The motivation is clear and convincing.

3. The writing is clear, and easy to understand.

**Weaknesses:**

1. This paper calibrates a surrogate binary classifier that predicts whether the class prediction is correct. However, according to the results in Table 3, in the majority of experimental results, the AUROC is worse when using the proposed method. The AUROC is not improved, meaning that the calibrator has not improved its ability to effectively distinguish between correct and incorrect model predictions, thus failing to learn to predict whether the class prediction is correct. This is contradictory to the paper's main objective and narrative.

2. The authors answered YES for Theory Assumptions and Proofs in the checklist. However, no proofs are shown. Some mathematical calculations are presented in Appendix D, but those are not proofs. They overclaim the theoretical contribution.

3. The evaluation metrics are limited. Only quantitative results on ECE and AUROC are reported (even though the AUROC is not improved). Other popular metrics such as ACE, MCE, or PIECE are not provided.

4. Authors report the results on CLIP models but do not discuss or compare them to recent calibration methods for CLIP models. For example:

   [1] Wang, S., Wang, J., Wang, G., Zhang, B., Zhou, K. and Wei, H., 2024. Open-Vocabulary Calibration for Vision-Language Models. International Conference on Machine Learning (ICML), 2024.


5. There are several existing model calibration papers in the literature that focus on directly estimating the probability of correctness for each sample/prediction.

     [2] Xiong, M., Deng, A., Koh, P.W.W., Wu, J., Li, S., Xu, J. and Hooi, B., 2023. Proximity-informed calibration for deep neural networks. Advances in Neural Information Processing Systems, 36, pp.68511-68538.

     [3] Liu, Y., Wang, L., Zou, Y., Zou, J. and Zheng, L., 2024. Optimizing Calibration by Gaining Aware of Prediction Correctness. arXiv preprint arXiv:2404.13016.

    For example, in paper [3], the idea is almost the same with minor differences. It also learns to predict the probability of correctness of the model's predictions as confidence. However, it can distinguish between correct and incorrect predictions based on their AUROC results.

**Questions:**

1. The results in Table 3 are not aligned with the paper's narrative. Why not discuss the mechanism or reason behind this? I think it is a vital issue.

2. Can the authors please provide strong reasons to convince me that your method is superior to similar works [2] and [3]?

**Limitations:**

Yes. They have a limitation section.

---

> ### Author Rebuttal · Authors · 2024-08-05
>
> We appreciate your careful consideration of our paper. We are glad that you appreciated our experiments and found our motivation and writing clear. We answer your questions below, and the weaknesses in an additional Comment.
>
> **Questions**
>
> > 1. The results in Table 3 are not aligned with the paper's narrative. [...]
>
> Our paper addresses confidence calibration, usually measured by ECE. AUROC is a global rank-based metric for selective classification: it relies on the relative values of the scores, not their absolute values. Even though calibration and selective classification may be related, improvement in calibration does not directly translate to better selective classification. This has been clearly demonstrated experimentally by [A].
>
> A good example of that difference is the behavior of $HB_{TvA}$: it is the best calibration method overall but actually degrades the AUROC in most cases. Such a difference can be explained by the fact that selective classification benefits from a continuous score able to discriminate between certain and uncertain examples finely, but HB quantizes the confidences into, e.g., 10 different values.
>
> We included experiments for selective classification by computing the AUROC to see if it could benefit from improvements in calibration. Our experiments confirmed the findings of [A].
>
> [A] *Ido Galil, Mohammed Dabbah, and Ran El-Yaniv. What Can we Learn From The Selective Prediction And Uncertainty Estimation Performance Of 523 Imagenet Classifiers? ICLR 2023*
>
> > 2. Can the authors please provide strong reasons to convince me that your method is superior to similar works [2] and [3]?
>
> We are glad that you open the discussion, and we hope the following arguments can convince you.
>
> The primary objective in [2] differs from ours: it "focuses on the problem of proximity bias in model calibration, a phenomenon wherein deep models tend to be more overconfident on data of low proximity,". The goal in [2] is to lower the difference in the confidence score values between regions of low and high density, i.e., to make the confidence score independent of a local density indicator called "proximity." There is no theoretical guarantee, however, that minimizing the proximity bias improves the confidence calibration, the focus of our work. Theorem 4.2 about the PIECE metric is a direct consequence of Jensen's inequality and is true for any random variable $D$, not necessarily a proximity score. Theorem 5.1 is an interesting bias/variance decomposition of the Brier score. However, as this type of decomposition usually states, the error may come from bias (here, a wrong initial calibration) or high estimation variance (which can be related to low density but is not expressed as such in the decomposition). We experimentally compare our approach to the ProCal algorithm using the code provided by [2] and observe that our approach gives much better ECE confidence calibration and, for half of the models, also better PIECE values (see attached PDF file).
> As you mentioned, both works however share similarities. [2] proposes a calibration method to achieve three goals: mitigate proximity bias, improve confidence calibration, and provide a plug-and-play method. We share the last two goals. Concerning improving confidence calibration, our approach has better results, as shown in the attached PDF. Both approaches are plug-and-play, but they apply very differently. The method by [2] is applied *after* existing calibration methods to further improve calibration. It thus does not solve any of the four issues we identified (e.g., cross-entropy loss is still inefficient, and One-versus-All still leads to highly imbalanced problems). Our Top-versus-All approach is a reformulation of the calibration problem that uses a surrogate binary classifier. Existing approaches are applied to this surrogate classifier, which is how the four issues are solved. We do not propose a new method but a new way of applying existing methods. Our approach does not introduce new hyperparameters (except in the particular case of regularizing scaling methods). [2] introduces several new hyperparameters, such as the choice of the distance, the number of KNN neighbors, or a shrinkage coefficient.
>
> Concerning paper [3], according to the NeurIPS 2024 FAQ for Authors, "Authors are not expected to compare to work that appeared only a month or two before the deadline.", which is the case for this work. However, we can still discuss it. In our understanding, the main intuition is similar indeed: binarize the calibration problem. However, what they do with this intuition differs vastly from our approach. [3] derives a loss (Eq. 7), which is almost the standard binary cross-entropy loss we use for scaling methods but without a logarithm. They use this loss to learn a separate model that predicts a sample-wise temperature coefficient. This is a new calibration method, which is not straightforward to implement due to the numerous hyperparameters (network architecture, image transformations...). It also requires multiple inferences at test-time, which can be problematic in some production models. Our approach is, again, not a calibration method but a general reformulation of the calibration problem that enhances existing methods. By looking at their Table 1, they get an ECE of 2.22 on ImageNet (in-distribution), while our approach achieves values around 0.5 for most models in our paper's Table 1. In our understanding of their results, the AUROC improvements seem mostly due to the use of image transformations, not from their proposed loss. Their method seems to work best in out-of-distribution scenarios, which is not the main objective of our paper.
>
> We will include the discussions on these relevant papers and the AUROC results in the final version of the paper.
>
> We hope that our rebuttal has addressed your concerns to your satisfaction. We would be grateful if you could consider revising your score in light of our response.

---

> ### Author Response · Authors · 2024-08-05
> **Answer to the weaknesses**
>
> **Weaknesses**
>
> > 1. [...] The AUROC is not improved [...]
>
> We address this point in our response to the related question; please see the rebuttal above.
>
> > 2. The authors answered YES for Theory Assumptions and Proofs in the checklist. [...]
>
> Please see our global answer, which addresses this weakness.
> We do not believe that we are making a significant theoretical contribution to the field. If there is any way that our work can be interpreted otherwise, we would appreciate being informed. We have misinterpreted the checklist and will answer NA for Theory Assumptions and Proofs to address potential confusion.
>
> > 3. The evaluation metrics are limited. [...]
>
> Besides ECE and AUROC we also include ECE with equal-mass bins (also sometimes called ACE), Brier score, and display qualitative results with reliability diagrams. Also, please note that what we call ECE with equal-mass bins is actually called ACE in some works. ACE uses equal-mass bins and is computed either classwise (as in the original definition) or not (as in [2] by looking at their code). To remove ambiguity, we call it ECE with equal-mass bins. As for the classwise version of ACE, it is not estimated well when the number of classes is high. We discussed this issue in Appendix E. Concerning MCE and PIECE, we will add the results in the Appendix of the final version of the paper. In the meantime, they are included in the attached PDF.
>
> > 4. [...] recent calibration methods for CLIP models.
>
> We were not aware of [1], which is not yet published; we thank you for providing the reference. We used CLIP as a zero-shot classifier and considered it as any other classifier. Our method is built with generality in mind, while [1] applies specifically to Vision-Language Models. We plan to discuss this approach and compare it to ours for CLIP in the final version, given that the code is available.
>
> > 5. [...] existing model calibration papers [2] [3] [...]
>
> We address this concern in our answer to the related question in the rebuttal above.

---

### Author Rebuttal · Authors · 2024-08-05

We thank all the Reviewers for the time spent reviewing our paper and for the constructive comments. In this global response, we group, summarize, and discuss the main identified strengths and weaknesses.

**Strengths**

The Reviewers mostly agree on three strengths:
- *The method is simple and efficient* (5 Reviewers: RhL1, eq8Z, yKjq, oe96, and QaKm).
- *The experiments are comprehensive* (5 Reviewers: S8uZ, RhL1, eq8Z, yKjq, and oe96).
- *The writing is clear* (4 Reviewers: S8uZ, RhL1, eq8Z, and yKjq).

These are indeed what we consider to be some of the main strengths of our paper, and we are glad that most Reviewers successfully identified them.
We also would like to emphasize the reproducibility and practical applicability of our work. Our approach is "plug-and-play" and easily integrates with most existing calibration methods with minimal code modification. We have made our complete codebase available (currently hosted on Anonymous GitHub), allowing practitioners to utilize our method and replicate our results.

**Weaknesses**

Two weaknesses are shared by several Reviewers:
- *The theoretical contribution could be further strengthened* (3 Reviewers: S8uZ, RhL1, and yKjq).
- *Some references are missing* (2 Reviewers: S8uZ, and oe96).

Concerning theoretical contribution, the primary focus of our paper is on experimental research, with an emphasis on practicality. To compensate for the lack of theoretical evidence, we conducted experiments on complex real-world problems, such as natural image and text classification. Given the extensiveness of our experiments, which five Reviewers highlighted, we consider the effectiveness of Top-versus-All to be clearly demonstrated. We also aimed to clarify our intuition by highlighting four shortcomings of standard approaches and discussing how our method addresses them. Moreover, our paper is not devoid of any theory: we developed a theoretical justification for the specific case of Temperature Scaling on lines 227-235 that derives from calculations included in Appendix D. When predictions are incorrect, the gradient with TvA is proportional to $\frac{s}{s-1}$ with $s$ being the confidence. This term means that as the confidence for wrong predictions gets higher, so does the gradient to reduce the confidence. Despite the limited scope of our theoretical justification, we believe it is a valuable outcome that partly explains our good experimental results.

Concerning the missing references, we thank the Reviewers for providing them to us. In particular, two Reviewers cited [1]. We thus conducted preliminary experiments to compare it to our approach. Results are available in the attached PDF. We used [1]'s public code to reproduce the experimental setting closest to ours, corresponding to the results of Table 4 in [1]. The comparison shows that our approach is always better for confidence calibration (the main focus of our work) and competitive for reducing the proximity bias (the main focus of their work), with our approach being the best for 2 of the 4 models studied. In addition to these better experimental results, we argue how our approach is superior to [1] in our response to Reviewer S8uZ.

Other weaknesses were mentioned by only one Reviewer each. To us, this means that, while valid, they are not major concerns. We address all weaknesses and questions in the rebuttals addressed to each review below.


[1] Xiong, M., Deng, A., Koh, P.W.W., Wu, J., Li, S., Xu, J. and Hooi, B., 2023. Proximity-informed calibration for deep neural networks. Advances in Neural Information Processing Systems, 36, pp.68511-68538.

---

### Decision · Program_Chairs · 2024-09-25

**Decision:**

Accept (poster)

**Comment:**

The paper lacks strong theoretical contributions and there is some concern about novelty.  However, its simplicity, practicality, and  experimental validation could make it a valuable contribution to the field of confidence calibration, particularly when dealing with a large number of classes.  The paper had many reviews (with some of them conflicting) and I have to discard some of the negative reviews on novelty due to the fact that there was no concrete justification by the reviewers.  I find that the paper is borderline accept as a poster.